# WHEN DOES PRECONDITIONING HELP OR HURT GENERALIZATION?

*Shun-ichi Amari[1], Jimmy Ba[2,3], Roger Grosse[2,3], Xuechen Li[4], Atsushi Nitanda[5,6], Taiji Suzuki[5,6], Denny Wu[2,3], Ji Xu[7]

[1]RIKEN CBS, [2]University of Toronto, [3]Vector Institute, [4]Google Research, Brain Team, [5]University of Tokyo, [6]RIKEN AIP, [7]Columbia University

amari@brain.riken.jp, {jba,rgrosse,lxuechen,dennywu}@cs.toronto.edu, {nitanda,taiji}@mist.i.u-tokyo.ac.jp, jixu@cs.columbia.edu

## ABSTRACT

While second order optimizers such as natural gradient descent (NGD) often speed up optimization, their effect on generalization has been called into question. This work presents a more nuanced view on how the *implicit bias* of optimizers affects the comparison of generalization properties. We provide an exact asymptotic bias-variance decomposition of the generalization error of preconditioned ridgeless regression in the overparameterized regime, and consider the inverse population Fisher information matrix (used in NGD) as a particular example. We determine the optimal preconditioner $P$ for both the bias and variance, and find that the relative generalization performance of different optimizers depends on label noise and "shape" of the signal (true parameters): when the labels are noisy, the model is misspecified, or the signal is misaligned with the features, NGD can achieve lower risk; conversely, GD generalizes better under clean labels, a well-specified model, or aligned signal. Based on this analysis, we discuss several approaches to manage the bias-variance tradeoff, and the potential benefit of interpolating between first- and second-order updates. We then extend our analysis to regression in the reproducing kernel Hilbert space and demonstrate that preconditioning can lead to more efficient decrease in the population risk. Lastly, we empirically compare the generalization error of first- and second-order optimizers in neural network experiments, and observe robust trends matching our theoretical analysis.

## 1 INTRODUCTION

We study the generalization property of an estimator $\hat{\theta}$ obtained by minimizing the empirical risk (or the training error) $L(f_\theta)$ via a preconditioned gradient update with preconditioner $P$:

$$\theta_{t+1} = \theta_t - \eta P(t)\nabla_{\theta_t} L(f_{\theta_t}), \quad t = 0, 1, \ldots \quad (1.1)$$

Setting $P = I$ recovers gradient descent (GD). Choices of $P$ which exploit second-order information include the inverse Fisher information matrix, which gives the natural gradient descent (NGD) (Amari, 1998); the inverse Hessian, which leads to Newton's method (LeCun et al., 2012); and diagonal matrices estimated from past gradients, which include various adaptive gradient methods (Duchi et al., 2011; Kingma & Ba, 2014). These preconditioners often alleviate the effect of pathological curvature and speed up *optimization*, but their impact on *generalization* has been under debate: Wilson et al. (2017) reported that in neural network optimization, adaptive or second-order methods generalize worse compared to gradient descent (GD), whereas other empirical studies showed that second-order methods achieve comparable, if not better generalization (Xu et al., 2020).

The generalization property of optimizers relates to the discussion of *implicit bias* (Gunasekar et al., 2018a), i.e. preconditioning may lead to a different converged solution (with potentially the same training loss), as illustrated in Figure 1. While many explanations have been proposed, our starting point is the well-known observation that GD often implicitly regularizes the parameter $\ell_2$ norm. For instance in overparameterized least squares regression, GD and many first-order methods find the minimum $\ell_2$ norm solution from zero initialization (without explicit regularization), but preconditioned updates may not. This being said, while the minimum $\ell_2$ norm solution can generalize well

---

*Alphabetical ordering. Correspondence to: Denny Wu (dennywu@cs.toronto.edu).

in the overparameterized regime (Bartlett et al., 2019), it is unclear whether preconditioning leads to inferior solutions – even in the simple setting of overparameterized linear regression, *quantitative* understanding of how preconditioning affects generalization is largely lacking.

Motivated by the observations above, in Section 3 we start with overparameterized least squares regression (unregularized) and analyze the stationary solution ($t \to \infty$) of update (1.1) under time-invariant preconditioner. Extending previous analysis in the proportional limit (Hastie et al., 2019), we consider a more general random design setting and derive the exact population risk in its *bias-variance decomposition*. We characterize the optimal $\boldsymbol{P}$ within a general class of preconditioners for both the bias and variance, and focus on the comparison between GD, for which $\boldsymbol{P}$ is identity, and NGD, for which $\boldsymbol{P}$ is the inverse population Fisher information matrix[1]. We find that the comparison of generalization is affected by the following factors:

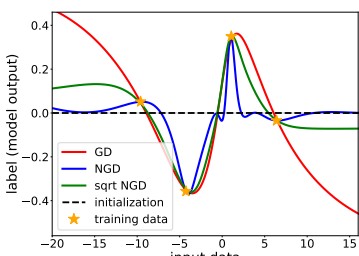

Figure 1: 1D illustration of different implicit biases: two-layer sigmoid network trained with preconditioned GD.

1. **Label Noise:** Additive noise in the labels leads to the *variance* term in the risk. We prove that NGD achieves the optimal variance among a general class of preconditioned updates.

2. **Model Misspecification:** Under misspecification, there does not exist a perfect $f_{\boldsymbol{\theta}}$ that recovers the true function (target). We argue that this factor is similar to additional label noise, and thus NGD may also be beneficial when the model is misspecified.

3. **Data-Signal-Alignment:** Alignment describes how the target signal distributes among the input features. We show that GD achieves lower *bias* when signal is isotropic, whereas NGD is preferred under "misalignment" — when the target function focuses on small feature directions.

Beyond the decomposition of stationary risk, our findings in Section 4 and 5 are summarized as:

- In Section 4.1 and 4.2 we discuss how the bias-variance tradeoff can be realized by different choices of preconditioner $\boldsymbol{P}$ (e.g. interpolating between GD and NGD) or early stopping.

- In Section 4.3 we extend our analysis to regression in the RKHS and show that under early stopping, a preconditioned update interpolating between GD and NGD achieves minimax optimal convergence rate in much fewer steps, and thus reduces the population risk faster than GD.

- In Section 5 we empirically test how well our findings in linear model carry over to neural networks: under a student-teacher setup, we compare the generalization of GD with preconditioned updates and illustrate the influence of all aforementioned factors. The performance of neural networks under a variety of manipulations results in trends that align with our theoretical analysis.

## 2 BACKGROUND AND RELATED WORKS

**Natural Gradient Descent.** NGD is a second-order method originally proposed in Amari (1997). Consider a data distribution $p(\boldsymbol{x})$ on the space $\mathcal{X}$, a function $f_{\boldsymbol{\theta}} : \mathcal{X} \to \mathcal{Z}$ parameterized by $\boldsymbol{\theta}$, and a loss function $L(\boldsymbol{X}, f_{\boldsymbol{\theta}}) = \frac{1}{n} \sum_{i=1}^{n} l(y_i, f_{\boldsymbol{\theta}}(\boldsymbol{x}_i))$, where $l : \mathcal{Y} \times \mathcal{Z} \to \mathbb{R}$. Also suppose a probability distribution $p(y|\boldsymbol{z}) = p(y|f_{\boldsymbol{\theta}}(\boldsymbol{x}))$ is defined on the space of labels. Then, the natural gradient is defined as: $\tilde{\nabla}_{\theta} L(\boldsymbol{X}, f_{\boldsymbol{\theta}}) = \boldsymbol{F}^{-1} \nabla_{\theta} L(\boldsymbol{X}, f_{\boldsymbol{\theta}})$, where $\boldsymbol{F} = \mathbb{E}[\nabla_{\boldsymbol{\theta}} \log p(\boldsymbol{x}, y|\boldsymbol{\theta}) \nabla_{\boldsymbol{\theta}} \log p(\boldsymbol{x}, y|\boldsymbol{\theta})^{\top}]$ is the *Fisher information matrix*, or simply the (population) Fisher. Note that expectations in the Fisher are under the joint distribution of the model $p(\boldsymbol{x}, y|\boldsymbol{\theta}) = p(\boldsymbol{x})p(y|f_{\boldsymbol{\theta}}(\boldsymbol{x}))$. In the literature, the Fisher is sometimes defined under the empirical data distribution $\{\boldsymbol{x}_i\}_{i=1}^{n}$ (Amari et al., 2000). We instead refer to this quantity as the *sample Fisher*, the properties of which influence optimization and have been studied in Karakida et al. (2018); Kunstner et al. (2019); Thomas et al. (2020). We remark that in linear and kernel regression under squared loss, sample Fisher-based updates give the same stationary solution as GD (see Section 3), whereas population Fisher-based update may not.

While the population Fisher is typically difficult to obtain, extra unlabeled data can be used in its estimation, which empirically improves generalization (Pascanu & Bengio, 2013). Moreover, under structural assumptions, parametric approaches to estimate $\boldsymbol{F}$ can be more sample-efficient (Martens & Grosse, 2015; Ollivier, 2015), and thus closing the gap between sample and population Fisher.

---

[1]From now on we use NGD to denote the *population* Fisher-based update, and we write "sample NGD" when $\boldsymbol{P}$ is the inverse or pseudo-inverse of the sample Fisher; see Section 2 for discussion.

When the per-instance loss is the negative log-probability of an exponential family, the sample Fisher coincides with the *generalized Gauss-Newton matrix* (Martens, 2014). In least squares regression, which is the focus of this work, the quantity also coincides with the Hessian. We thus take NGD as a representative example of preconditioned update, and we expect our findings to also translate to other second-order methods (not including adaptive gradient methods) in regression problems.

**Analysis of Preconditioned Gradient Descent.** While Wilson et al. (2017) outlined one example under fixed training data where GD generalizes better than adaptive methods, in the online learning setting, for which optimization speed relates to generalization, several works have shown the advantage of preconditioning (Levy & Duchi, 2019; Zhang et al., 2019a). In addition, Zhang et al. (2019b); Cai et al. (2019) established convergence and generalization guarantees of sample Fisher-based updates for neural networks in the kernel regime. Lastly, the generalization of different optimizers relates to the notion of "sharpness" (Keskar et al., 2016; Dinh et al., 2017), and it has been argued that second-order updates tend to find sharper minima (Wu et al., 2018).

We note that two concurrent works also discussed the generalization performance of preconditioned updates. Wadia et al. (2020) connected second-order methods with data whitening in linear models, and qualitatively showed that whitening (thus second-order update) harms generalization in certain cases. Vaswani et al. (2020) analyzed the complexity of the maximum $P$-margin solution in linear classification problems. We emphasize that instead of *upper bounding* the risk (e.g. Rademacher complexity), which may not decide the optimal $P$ for generalization error, we compute the *exact risk* for least squares regression, which allows us to precisely compare different preconditioners.

## 3 ASYMPTOTIC RISK OF RIDGELESS INTERPOLANTS

In this section we consider the following setup: given $n$ training samples $\{\boldsymbol{x}_i\}_{i=1}^n$ labeled by a teacher model (target function) $f^* : \mathbb{R}^d \to \mathbb{R}$ with additive noise: $y_i = f^*(\boldsymbol{x}_i) + \varepsilon_i$, we learn a linear student model $f_{\boldsymbol{\theta}}$ by minimizing the squared loss: $L(\boldsymbol{X}, f_{\boldsymbol{\theta}}) = \sum_{i=1}^n (y_i - \boldsymbol{x}_i^\top \boldsymbol{\theta})^2$. We assume a random design: $\boldsymbol{x}_i = \boldsymbol{\Sigma}_{\boldsymbol{X}}^{1/2} \boldsymbol{z}_i$, where $\boldsymbol{z}_i \in \mathbb{R}^d$ is an i.i.d. vector with zero-mean, unit-variance, and finite 12th moment, and $\varepsilon$ is i.i.d. noise independent to $\boldsymbol{z}$ with mean 0 and variance $\sigma^2$. Our goal is to compute the population risk $R(f) = \mathbb{E}_{\boldsymbol{x}}[(f^*(\boldsymbol{x}) - f(\boldsymbol{x}))^2]$ in the proportional asymptotic limit:

- **(A1) Overparameterized Proportional Limit:** $n, d \to \infty$, $d/n \to \gamma \in (1, \infty)$.

(A1) entails that the number of features (or parameters) is larger than the number of samples, and there exist multiple empirical risk minimizers with potentially different generalization properties.

Denote $\boldsymbol{X} = [\boldsymbol{x}_1^\top, ..., \boldsymbol{x}_n^\top]^\top \in \mathbb{R}^{n \times d}$ the data matrix and $\boldsymbol{y} \in \mathbb{R}^n$ the corresponding label vector. We optimize the parameters $\boldsymbol{\theta}$ via a preconditioned gradient flow with preconditioner $\boldsymbol{P}(t) \in \mathbb{R}^{d \times d}$,

$$\frac{\partial \boldsymbol{\theta}(t)}{\partial t} = -\boldsymbol{P}(t) \frac{\partial L(\boldsymbol{\theta}(t))}{\partial \boldsymbol{\theta}(t)} = \frac{1}{n} \boldsymbol{P}(t) \boldsymbol{X}^\top (\boldsymbol{y} - \boldsymbol{X}\boldsymbol{\theta}(t)), \quad \boldsymbol{\theta}(0) = 0. \quad (3.1)$$

In this linear setup, many common choices of preconditioner do not change through time: under Gaussian likelihood, the sample Fisher (and also Hessian) corresponds to the sample covariance $\boldsymbol{X}^\top \boldsymbol{X}/n$ up to variance scaling, whereas the population Fisher corresponds to the population covariance $\boldsymbol{F} = \boldsymbol{\Sigma}_{\boldsymbol{X}}$. We thus limit our analysis to fixed preconditioner of the form $\boldsymbol{P}(t) =: \boldsymbol{P}$.

Write parameters at time $t$ under update (3.1) with fixed $\boldsymbol{P}$ as $\boldsymbol{\theta}_{\boldsymbol{P}}(t)$. For positive definite $\boldsymbol{P}$, the stationary solution is given as: $\hat{\boldsymbol{\theta}}_{\boldsymbol{P}} := \lim_{t \to \infty} \boldsymbol{\theta}_{\boldsymbol{P}}(t) = \boldsymbol{P}\boldsymbol{X}^\top (\boldsymbol{X}\boldsymbol{P}\boldsymbol{X}^\top)^{-1}\boldsymbol{y}$. One may check that discrete time gradient descent update (with appropriate step size) and other variants that do not alter the span of gradient (e.g. stochastic gradient or momentum) converge to the same solution as well.

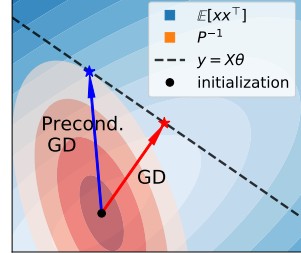

Figure 2: Geometric illustration (2D) of how the interpolating $\boldsymbol{\theta}_{\boldsymbol{P}}$ depends on the preconditioner.

Intuitively speaking, if the data distribution (blue contour in Figure 2) is not isotropic, then certain directions will be more "important" than others. In this case uniform $\ell_2$ shrinkage (which GD implicitly provides) may not be most desirable, and certain $\boldsymbol{P}$ that takes data geometry into account may lead to better generalization instead. The above intuition will be made rigorous in this section.

**Remark.** $\hat{\boldsymbol{\theta}}_{\boldsymbol{P}}$ *is the minimum* $\|\boldsymbol{\theta}\|_{\boldsymbol{P}^{-1}}$ *norm interpolant:* $\hat{\boldsymbol{\theta}}_{\boldsymbol{P}} = \arg\min_{\boldsymbol{\theta}}\|\boldsymbol{\theta}\|_{\boldsymbol{P}^{-1}}$, s.t. $\boldsymbol{X}\boldsymbol{\theta} = \boldsymbol{y}$ *for positive definite* $\boldsymbol{P}$*. For GD this translates to the parameter* $\ell_2$ *norm, whereas for NGD* $(\boldsymbol{P} = \boldsymbol{F}^{-1})$*, the implicit bias is the* $\|\boldsymbol{\theta}\|_{\boldsymbol{\Sigma}_{\boldsymbol{X}}}$ *norm. Since* $\mathbb{E}[f(\boldsymbol{x})^2] = \|\boldsymbol{\theta}\|_{\boldsymbol{\Sigma}_{\boldsymbol{X}}}^2$*, NGD finds an interpolating function with smallest norm under the data distribution. We empirically observe this divide between small parameter norm and function norm in neural networks as well (see Figure 1 and Appendix A.1).*

We highlight the following choices of $\boldsymbol{P}$ and the corresponding stationary solution $\hat{\boldsymbol{\theta}}_{\boldsymbol{P}}$ as $t \to \infty$.

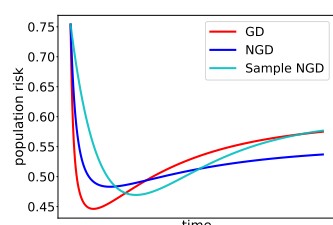

- **Identity:** $\boldsymbol{P} = \boldsymbol{I}_d$ recovers GD that converges to the min $\ell_2$ norm interpolant (also true for momentum GD and SGD), which we write as $\hat{\boldsymbol{\theta}}_{\boldsymbol{I}} := \boldsymbol{X}^{\top}(\boldsymbol{X}\boldsymbol{X}^{\top})^{-1}\boldsymbol{y}$ and refer to as the *GD solution*.
- **Population Fisher:** $\boldsymbol{P} = \boldsymbol{F}^{-1} = \boldsymbol{\Sigma}_{\boldsymbol{X}}^{-1}$ leads to the estimator $\hat{\boldsymbol{\theta}}_{\boldsymbol{F}^{-1}}$, which we refer to as the *NGD solution*.
- **Sample Fisher:** since the sample Fisher is rank-deficient, we may add a damping term $\boldsymbol{P} = (\boldsymbol{X}^{\top}\boldsymbol{X} + \lambda\boldsymbol{I}_d)^{-1}$ or take the pseudo-inverse $\boldsymbol{P} = (\boldsymbol{X}^{\top}\boldsymbol{X})^{\dagger}$. In both cases, the gradient is still spanned by $\boldsymbol{X}$, and thus the update finds the same min $\ell_2$-norm solution $\hat{\boldsymbol{\theta}}_{\boldsymbol{I}}$ (also true for full-matrix Adagrad (Agarwal et al., 2018)), although the trajectory differs (see Figure 3).

Figure 3: Population risk of preconditioned linear regression vs. time with the following $\boldsymbol{P}$: $\boldsymbol{I}$ (red), $\boldsymbol{\Sigma}_{\boldsymbol{X}}^{-1}$ (blue) and $(\boldsymbol{X}^{\top}\boldsymbol{X})^{\dagger}$ (cyan). Time is rescaled differently for each curve (convergence speed is not comparable). Observe that GD and sample NGD give the same stationary risk.

**Remark.** *The above choices reveal a gap between sample- and population-based* $\boldsymbol{P}$*: while the sample Fisher accelerates optimization (Zhang et al., 2019b), the following sections demonstrate generalization properties only possessed by the population Fisher.*

We compare the population risk of the GD solution $\hat{\boldsymbol{\theta}}_{\boldsymbol{I}}$ and NGD solution $\hat{\boldsymbol{\theta}}_{\boldsymbol{F}^{-1}}$ in its bias-variance decomposition w.r.t. label noise (Hastie et al., 2019) and discuss the two components separately,

$$R(\boldsymbol{\theta}) = \underbrace{\mathbb{E}_{\boldsymbol{x}}[(f^*(\boldsymbol{x}) - \langle \boldsymbol{x}, \mathbb{E}_{\varepsilon}[\boldsymbol{\theta}]\rangle)^2]}_{B(\boldsymbol{\theta}),\ \text{bias}} + \underbrace{\text{tr}(\text{Cov}(\boldsymbol{\theta})\boldsymbol{\Sigma}_{\boldsymbol{X}})}_{V(\boldsymbol{\theta}),\ \text{variance}}.$$

Note that the *bias* does not depend on label noise $\varepsilon$, and the *variance* does not depend on the teacher model $f^*$. Additionally, given that $f^*$ can be independently decomposed into a linear component on features $\boldsymbol{x}$ and a residual: $f^*(\boldsymbol{x}) = \langle \boldsymbol{x}, \boldsymbol{\theta}^*\rangle + f_c^*(\boldsymbol{x})$, we can separate the bias term into a *well-specified* component $\|\boldsymbol{\theta}^* - \mathbb{E}\boldsymbol{\theta}\|_{\boldsymbol{\Sigma}_{\boldsymbol{X}}}^2$, which captures the difficulty in learning $\boldsymbol{\theta}^*$, and a *misspecified* component, which corresponds to the error due to fitting $f_c^*$ (beyond what the student can represent).

### 3.1 THE VARIANCE TERM: NGD IS OPTIMAL

We first characterize the stationary variance which is independent to the teacher model $f^*$. We restrict ourselves to preconditioners satisfying the following assumption on the spectral distribution:

- **(A2) Converging Eigenvalues:** $\boldsymbol{P}$ is positive definite and as $n, d \to \infty$, the spectral distribution of $\boldsymbol{\Sigma}_{\boldsymbol{X}\boldsymbol{P}} := \boldsymbol{P}^{1/2}\boldsymbol{\Sigma}_{\boldsymbol{X}}\boldsymbol{P}^{1/2}$ converges weakly to $\boldsymbol{H}_{\boldsymbol{X}\boldsymbol{P}}$ supported on $[c, C]$ for $c, C > 0$.

The following theorem characterizes the asymptotic variance and the corresponding optimal $\boldsymbol{P}$.

**Theorem 1.** *Given (A1-2), the asymptotic variance is given as*

$$V(\hat{\boldsymbol{\theta}}_{\boldsymbol{P}}) \xrightarrow{p} \sigma^2 \Big( \lim_{\lambda \to 0_+} m'(-\lambda)m^{-2}(-\lambda) - 1 \Big), \tag{3.2}$$

*where* $m(z) > 0$ *is the Stieltjes transform of the limiting distribution of eigenvalues of* $\frac{1}{n}\boldsymbol{X}\boldsymbol{P}\boldsymbol{X}^{\top}$ *satisfying the self-consistent equation* $m^{-1}(z) = -z + \gamma \int \tau(1 + \tau m(z))^{-1}\mathrm{d}\boldsymbol{H}_{\boldsymbol{X}\boldsymbol{P}}(\tau)$.

*Furthermore, under (A1-2),* $V(\hat{\boldsymbol{\theta}}_{\boldsymbol{P}}) \geq \sigma^2(\gamma - 1)^{-1}$*, and the equality is achieved by* $\boldsymbol{P} = \boldsymbol{F}^{-1}$.

Formula (3.2) is a direct extension of (Hastie et al., 2019, Thorem 4), which can be obtained from the general results of Ledoit & Péché (2011); Dobriban et al. (2018). Theorem 1 implies that preconditioning with the inverse population Fisher $\boldsymbol{F}$ results in the optimal stationary variance, which is supported by Figure 5(a). In other words, when the labels are noisy so that the risk is dominated by

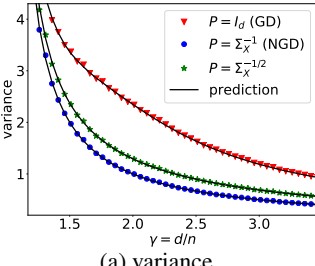 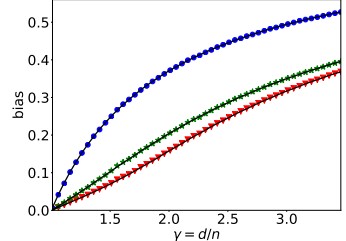 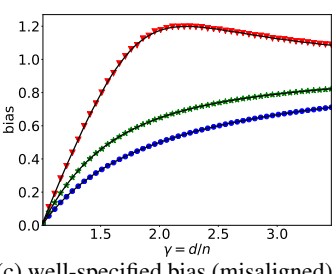

(a) variance.  (b) well-specified bias (isotropic).  (c) well-specified bias (misaligned).

Figure 5: We set eigenvalues of $\mathbf{\Sigma_X}$ as two point masses with $\kappa_X = 20$ and $\|\mathbf{\Sigma_X}\|_F^2 = d$; empirical values (dots) are computed with $n = 300$. (a) NGD (blue) achieves minimum variance. (b) GD (red) achieves lower bias under isotropic signal: $\mathbf{\Sigma_\theta} = \mathbf{I}_d$. (c) NGD achieves lower bias under "misalignment": $\mathbf{\Sigma_X} = \mathbf{\Sigma_\theta}^{-1}$.

the variance, we expect NGD to generalize better upon convergence. We emphasize that this advantage is only present when the population Fisher is used, but not its sample-based counterpart (which gives $\hat{\boldsymbol{\theta}}_I$). In Appendix A.3 we discuss the sample complexity of estimating $\mathbf{F}$ from unlabeled data.

**Misspecification $\approx$ Label Noise.** Under model misspecification, there does not exist a linear student that perfectly recovers the teacher $f^*$, which we may decompose as: $f^*(\boldsymbol{x}) = \boldsymbol{x}^\top \boldsymbol{\theta}^* + f_c^*(\boldsymbol{x})$. In the simple case where $f_c^*$ is an independent linear function on unobserved features (Hastie et al., 2019, Section 5): $y_i = \boldsymbol{x}_i^\top \boldsymbol{\theta}^* + \boldsymbol{x}_{c,i}^\top \boldsymbol{\theta}^c + \varepsilon_i$, where $\boldsymbol{x}_{c,i} \in \mathbb{R}^{d_c}$ is independent to $\boldsymbol{x}_i$, we can show that the additional error in the *bias* term due to misspecification is analogous to the *variance* term above:

**Corollary 2.** *Under (A1)(A2), for the above unobserved features model with $\mathbb{E}[\boldsymbol{x}^c \boldsymbol{x}^{c\top}] = \mathbf{\Sigma_X^c}$ and $\mathbb{E}[\boldsymbol{\theta}^c \boldsymbol{\theta}^{c\top}] = d_c^{-1} \mathbf{\Sigma_\theta^c}$, the additional error due to misspecification can be written as $B_c(\hat{\boldsymbol{\theta}}_P) = d_c^{-1} \mathrm{tr}(\mathbf{\Sigma_X^c} \mathbf{\Sigma_\theta^c})(V(\hat{\boldsymbol{\theta}}_P) + 1)$, where $V(\hat{\boldsymbol{\theta}}_P)$ is the variance in (3.2).*

In this case, misspecification can be interpreted as additional label noise, for which NGD is optimal by Theorem 1. While Corollary 2 describes one specific example of misspecification, we may expect similar outcome under broader settings. In particular, (Mei & Montanari, 2019, Remark 5) indicates that for many nonlinear $f_c^*$, the misspecified bias is same as variance due to label noise. We empirically observe similar findings under general covariance in Figure 4, in which $f_c^*$ is a quadratic function. Observe that NGD leads to lower bias compared to GD as we further misspecify the teacher model.

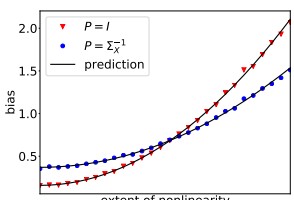

Figure 4: Misspecified bias with $\mathbf{\Sigma_\theta} = \mathbf{I}_d$ (favors GD) and $f_c^*(\boldsymbol{x}) = \alpha(\boldsymbol{x}^\top \boldsymbol{x} - \mathrm{tr}(\mathbf{\Sigma_X}))$, where $\alpha$ controls the *extent of nonlinearity*. Predictions are generated by matching $\sigma^2$ with second moment of $f_c^*$.

### 3.2 THE BIAS TERM: ALIGNMENT AND "DIFFICULTY" OF LEARNING

We now analyze the bias term when the teacher model is linear on the input features (hence well-specified): $f^*(\boldsymbol{x}) = \boldsymbol{x}^\top \boldsymbol{\theta}^*$. Extending the random effects hypothesis in Dobriban et al. (2018), we consider a more general prior on $\boldsymbol{\theta}^*$: $\mathbb{E}[\boldsymbol{\theta}^* \boldsymbol{\theta}^{*\top}] = d^{-1} \mathbf{\Sigma_\theta}$, and assume the following joint relations[2]:

- **(A3) Joint Convergence:** $\mathbf{\Sigma_X}$ and $\boldsymbol{P}$ share the same eigenvector matrix $\boldsymbol{U}$. The empirical distributions of elements of $(\boldsymbol{e}_x, \boldsymbol{e}_\theta, \boldsymbol{e}_{xp})$ jointly converge to random variables $(\upsilon_x, \upsilon_\theta, \upsilon_{xp})$ supported on $[c', C']$ for $c', C' > 0$, where $\boldsymbol{e}_x, \boldsymbol{e}_{xp}$ are eigenvalues of $\mathbf{\Sigma_X}, \mathbf{\Sigma_{XP}}$, and $\boldsymbol{e}_\theta = \mathrm{diag}\left(\boldsymbol{U}^\top \mathbf{\Sigma_\theta} \boldsymbol{U}\right)$.

We remark that when $\boldsymbol{P} = \boldsymbol{I}_d$, previous works (Hastie et al., 2019; Xu & Hsu, 2019) considered the special case of isotropic prior $\mathbf{\Sigma_\theta} = \boldsymbol{I}_d$. Our assumption thus allows for analysis of the bias term under much more general $\mathbf{\Sigma_\theta}$[3], which gives rise to interesting phenomena that are not captured by simplified settings, such as non-monotonic bias and variance for $\gamma > 1$ (see Figure 15), and the epoch-wise double descent phenomenon (see Appendix A.5). Under this general setup, we have the following characterization of the asymptotic bias and the corresponding optimal preconditioner:

**Theorem 3.** *Under (A1)(A3), the expected bias $B(\hat{\boldsymbol{\theta}}_P) := \mathbb{E}_{\boldsymbol{\theta}^*}[B(\hat{\boldsymbol{\theta}}_P)]$ is given as*

$$B(\hat{\boldsymbol{\theta}}_P) \xrightarrow{p} \lim_{\lambda \to 0_+} m'(-\lambda) m^{-2}(-\lambda) \mathbb{E}\left[\upsilon_x \upsilon_\theta (1 + \upsilon_{xp} m(-\lambda))^{-2}\right], \qquad (3.3)$$

---

[2]Note that (A2)(A3) covers many common choices of preconditioner, such as the population Fisher and variants of the sample Fisher (which is degenerate but leads to the same minimum $\ell_2$ norm solution as GD).

[3]Two concurrent works (Wu & Xu, 2020; Richards et al., 2020) also considered similar relaxation of $\mathbf{\Sigma_\theta}$ in the context of ridge regression.

*where expectation is taken over $\upsilon$ and $m(z)$ is the Stieltjes transform defined in Theorem 1.*

*Furthermore, among all $\boldsymbol{P}$ satisfying (A3), the optimal bias is achieved by $\boldsymbol{P} = \boldsymbol{U} \operatorname{diag}(\boldsymbol{e_\theta}) \boldsymbol{U}^\top$.*

Note that the optimal $\boldsymbol{P}$ depends on the "orientation" of the teacher model $\boldsymbol{\Sigma_\theta}$, which is usually not known in practice. This result can thus be interpreted as a *no-free-lunch* characterization in choosing an optimal preconditioner for the bias term *a priori*. As a consequence of the theorem, when the true parameters $\boldsymbol{\theta}^*$ have roughly equal magnitude (isotropic), GD achieves lower bias (see Figure 5(b) where $\boldsymbol{\Sigma_\theta} = \boldsymbol{I}_d$). On the other hand, NGD leads to lower bias when $\boldsymbol{\Sigma_X}$ is "misaligned" with $\boldsymbol{\Sigma_\theta}$, i.e. when $\boldsymbol{\theta}^*$ focus on the least varying directions of input features (see Figure 5(c) where $\boldsymbol{\Sigma_\theta} = \boldsymbol{\Sigma_X^{-1}}$), in which case learning is intuitively difficult since the features are not useful.

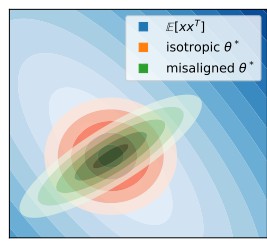

Figure 6: Illustration of isotropic and misaligned $\boldsymbol{\theta}^*$.

**Connection to Source Condition.** The "difficulty" of learning above relates to the *source condition* in RKHS literature (Cucker & Smale, 2002) (i.e., $\mathbb{E}[\boldsymbol{\Sigma_X^{r/2}} \boldsymbol{\theta}^*] < \infty$, see (A4) in Section 4.3), in which the coefficient $r$ can be interpreted as a measure of "misalignment". To elaborate this connection, we consider the setting of $\boldsymbol{\Sigma_\theta} = \boldsymbol{\Sigma_X^r}$: note that as $r$ decreases, the teacher $\boldsymbol{\theta}^*$ focuses more on input features with small magnitude, thus the learning problem becomes harder, and vice versa. In this case we can show a clear transition in $r$ for the comparison between GD and NGD.

**Proposition 4** (Informal). *When $\boldsymbol{\Sigma_\theta} = \boldsymbol{\Sigma_X^r}$, there exists a transition point $r^* \in (-1, 0)$ such that GD achieves lower (higher) stationary bias than NGD when $r > (<) r^*$.*

The above proposition confirms that for the stationary bias (well-specified), NGD outperforms GD in the misaligned setting (i.e., small $r$), whereas GD has an advantage when the signal is aligned (large $r$). For formal statement and more discussion on the transition point $r^*$ see Appendix A.2.

## 4 BIAS-VARIANCE TRADEOFF

Our characterization of stationary risk suggests that preconditioners that achieve the optimal bias and variance are generally different. This section discusses how bias-variance tradeoff can be realized by interpolating between preconditioners or by early stopping. Additionally, we analyze the non-parametric least squares setting and show that by balancing the bias and variance, a preconditioned update that interpolates between GD and NGD decreases the population risk faster than GD.

### 4.1 INTERPOLATING BETWEEN PRECONDITIONERS

Depending on the orientation of the teacher model, we may expect a bias-variance tradeoff in choosing $\boldsymbol{P}$. Intuitively, given $\boldsymbol{P}_1$ that minimizes the bias and $\boldsymbol{P}_2$ that minimizes the variance, it is possible that a preconditioner interpolating between $\boldsymbol{P}_1$ and $\boldsymbol{P}_2$ can balance the bias and variance and thus generalize better. The following proposition confirms this intuition in a setup of general $\boldsymbol{\Sigma_\theta}$ [4] and isotropic $\boldsymbol{\Sigma_\theta}$, for which GD ($\boldsymbol{P} = \boldsymbol{I}_d$) achieves optimal stationary bias and NGD ($\boldsymbol{P} = \boldsymbol{F}^{-1}$) achieves optimal variance.

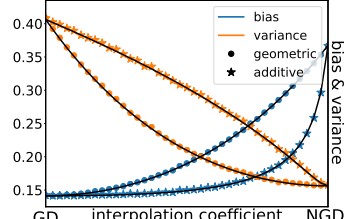

**Proposition 5** (Informal). *Let $\boldsymbol{\Sigma_X} \neq \boldsymbol{I}_d$ and $\boldsymbol{\Sigma_\theta} = \boldsymbol{I}_d$. Consider the following choices of interpolation scheme: (i) $\boldsymbol{P}_\alpha = \alpha \boldsymbol{\Sigma_X^{-1}} + (1-\alpha)\boldsymbol{I}_d$, (ii) $\boldsymbol{P}_\alpha = (\alpha \boldsymbol{\Sigma_X} + (1-\alpha)\boldsymbol{I}_d)^{-1}$, (iii) $\boldsymbol{P}_\alpha = \boldsymbol{\Sigma_X^{-\alpha}}$. The stationary variance monotonically decreases with $\alpha \in [0, 1]$ for all three choices. For (i), the stationary bias monotonically increases with $\alpha \in [0, 1]$, whereas for (ii) and (iii), the bias monotonically increases with $\alpha$ in certain range that depends on $\boldsymbol{\Sigma_X}$.*

Figure 7: Bias-variance tradeoff with $\kappa_X = 25$, $\boldsymbol{\Sigma_\theta} = \boldsymbol{I}_d$ and SNR=32/5. As we additively (ii) or geometrically (iii) interpolate from GD to NGD (left to right), the stationary bias (blue) increases and the stationary variance (orange) decreases.

In other words, as the signal-to-noise ratio (SNR) decreases, one can increase $\alpha$, which makes the update closer to NGD, to improve generalization, and vice versa. Indeed as shown in Figure 7 and 16(c), at a certain SNR, interpolating between $\boldsymbol{\Sigma_X^{-1}}$ and $\boldsymbol{\Sigma_\theta}$ can improve the stationary risk.

---

[4]Note that this reduces to the random effects model studied in Dobriban et al. (2018).

**Remark.** *Two of the aforementioned interpolation schemes reflect common practical choices: additive interpolation (ii) corresponds to the damping term to stably invert the Fisher, whereas geometric interpolation (iii) resembles the "conservative" square-root scaling in adaptive gradient methods.*

## 4.2 THE ROLE OF EARLY STOPPING

Thus far we considered the stationary solution of the unregularized objective. It is known that the bias-variance tradeoff can also be controlled by either explicit or algorithmic regularization. We briefly comment on the effect of early stopping, starting from the monotonicity of the variance term.

**Proposition 6.** *For all $\boldsymbol{P}$ satisfying (A2), the variance $V(\boldsymbol{\theta}_{\boldsymbol{P}}(t))$ monotonically increases with time.*

The proposition confirms the intuition that early stopping reduces overfitting. Variance reduction can benefit GD in its comparison to NGD, which achieves lowest stationary variance: indeed, Figure 3 and 19 show that under early stopping, GD may be favored even if NGD has lower stationary risk.

On the other hand, early stopping may not improve the bias term. While a complete analysis is difficult partially due to the potential non-monotonicity of the bias term (see Appendix A.5), we speculate that previous findings for the stationary bias also translate to early stopping. As a concrete example, we consider well-specified settings in which either GD or NGD achieves the optimal stationary bias, and demonstrate that such optimality is also preserved under early stopping:

**Proposition 7.** *Given (A1) and denote the optimal early stopping bias as $B^{\mathrm{opt}}(\boldsymbol{\theta}) = \inf_{t \geq 0} B(\boldsymbol{\theta}(t))$. When $\boldsymbol{\Sigma_\theta} = \boldsymbol{\Sigma_X^{-1}}$, we have $B^{\mathrm{opt}}(\boldsymbol{\theta_P}) \geq B^{\mathrm{opt}}(\boldsymbol{\theta_{F^{-1}}})$ for all $\boldsymbol{P}$ satisfying (A3). Whereas when $\boldsymbol{\Sigma_\theta} = \boldsymbol{I}_d$, we have $B^{\mathrm{opt}}(\boldsymbol{\theta_{F^{-1}}}) \geq B^{\mathrm{opt}}(\boldsymbol{\theta_I})$.*

Figure 19 illustrates that the observed trend in the stationary bias (well-specified) is indeed preserved under optimal early stopping: GD or NGD achieves lower early stopping bias under isotropic or misaligned teacher model, respectively. We leave a more precise characterization as future work.

## 4.3 FAST DECAY OF POPULATION RISK

Our previous analysis suggests that certain preconditioners can achieve lower population risk, but does not address which method decreases the risk more efficiently. Knowing that preconditioned updates may accelerate optimization, one natural question to ask is, is this speedup also present for generalization under fixed dataset? We answer this question in the affirmative in a slightly different model: we study least squares regression in the RKHS, and show that a preconditioned update that interpolates between GD and NGD achieves the minimax optimal rate in much fewer steps than GD.

We provide a brief outline and defer the details to Appendix D.9.1. Let $\mathcal{H}$ be an RKHS included in $L_2(P_X)$ equipped with a bounded kernel function $k$, and $K_{\boldsymbol{x}} \in \mathcal{H}$ be the Riesz representation of the kernel function. Define $S$ as the canonical operator from $\mathcal{H}$ to $L_2(P_X)$, and write $\Sigma = S^*S$ and $L = SS^*$. We aim to learn the teacher model $f^*$, under the following standard regularity conditions:

- **(A4) Source Condition:** $\exists r \in (0, \infty)$, $M > 0$ s.t. $f^* = L^r h^*$ for $h^* \in L_2(P_X)$ and $\|f^*\|_\infty \leq M$.

- **(A5) Capacity Condition:** $\exists s > 1$ s.t. $\mathrm{tr}(\Sigma^{1/s}) < \infty$ and $2r + s^{-1} > 1$.

- **(A6) Regularity of RKHS:** $\exists \mu \in [s^{-1}, 1]$, $C_\mu > 0$ s.t. $\sup_{\boldsymbol{x} \in \mathrm{supp}(P_X)} \|\Sigma^{1/2 - 1/\mu} K_{\boldsymbol{x}}\|_{\mathcal{H}} \leq C_\mu$.

Note that in the source condition (A4), the coefficient $r$ controls the complexity of the teacher model and relates to the notions of model misalignment in Section 3.2: large $r$ indicates a smoother teacher model which is "easier" to learn, and vice versa[5] (Steinwart et al., 2009). Given training points $\{(\boldsymbol{x}_i, y_i)\}_{i=1}^n$, we consider the following preconditioned update on the student model $f_t \in \mathcal{H}$:

$$f_t = f_{t-1} - \eta(\Sigma + \alpha I)^{-1}(\hat{\Sigma} f_{t-1} - \hat{S}^* Y), \quad f_0 = 0, \tag{4.1}$$

where $\hat{\Sigma} = \frac{1}{n}\sum_{i=1}^n K_{\boldsymbol{x}_i} \otimes K_{\boldsymbol{x}_i}$, $\hat{S}^* Y = \frac{1}{n}\sum_{i=1}^n y_i K_{\boldsymbol{x}_i}$. In this setup, the population Fisher corresponds to the covariance operator $\Sigma$, and thus (4.1) can be interpreted as *additive* interpolation between GD and NGD: update with large $\alpha$ behaves like GD, and small $\alpha$ like NGD. Related to our update is the FALKON algorithm (Rudi et al., 2017) – a preconditioned gradient method for kernel ridge regression. The key difference is that we consider optimizing the original objective (instead of

---

[5] We remark that most previous works considered the case where $r \geq 1/2$ which implies $f^* \in \mathcal{H}$.

a regularized version as in FALKON) under early stopping. Importantly, since we aim to understand *how preconditioning affects generalization*, explicit regularization should not be taken into account.

The following theorem shows that with appropriately chosen $\alpha$, the preconditioned update (4.1) leads to more efficient decrease in the population risk, due to faster decay of the bias term.

**Theorem 8** (Informal). *Under (A4-6), the population risk of $f_t$ can be written as $R(f_t) = \|Sf_t - f^*\|^2_{L_2(P_X)} \le B(t) + V(t)$ defined in Appendix D.9. Given $r \ge 1/2$ or $\mu \le 2r$, preconditioned update (4.1) with $\alpha = n^{-\frac{2s}{2rs+1}}$ achieves minimax optimal convergence rate $R(f_t) = \tilde{O}\left(n^{-\frac{2rs}{2rs+1}}\right)$ in $t = \Theta(\log n)$ steps, whereas ordinary gradient descent requires $t = \Theta\left(n^{\frac{2rs}{2rs+1}}\right)$ steps.*

We comment that the optimal interpolation coefficient $\alpha$ and stopping time $t$ are chosen to balance the bias $B(t)$ and variance $V(t)$. Note that $\alpha$ depends on the teacher model in the following way: for $n > 1$, $\alpha$ decreases as $r$ becomes smaller, which corresponds to non-smooth and "difficult" $f^*$, and vice versa. This agrees with our previous observation that NGD is advantageous when the teacher model is difficult to learn. We defer empirical verification of this result to Appendix C.

## 5 NEURAL NETWORK EXPERIMENTS

**Protocol.** We compare the generalization performance of GD and NGD in neural network settings and illustrate the influence of the following three factors: $(i)$ label noise; $(ii)$ misspecification; $(iii)$ signal misalignment. We also verify the potential advantage of interpolating between GD and NGD.

We consider the MNIST and CIFAR-10 datasets. To create a student-teacher setup, we split the training set into two halves, one of which (*pretrain* split) along with the original labels is used to pretrain the teacher, and the other (*distill* split) along with the teacher's labels is used to distill (Hinton et al., 2015) the student. We take the teacher to be either a two-layer fully-connected ReLU network or a ResNet (He et al., 2016), and the student is a two-layer ReLU network. We normalize the teacher's labels logits following Ba & Caruana (2014) before potentially adding label noise, and fit the student by minimizing the L2 loss. Student models are trained on a subset of the distill split with full-batch updates. We implement NGD using Hessian-free optimization (Martens, 2010). We use 100k unlabeled data (possibly applying data augmentation) to estimate the population Fisher. We report the test error when the training error is below $0.2\%$ of its initial value as a proxy for the stationary risk. We defer detailed setup to Appendix E and additional results to Appendix C.

### 5.1 EMPIRICAL FINDINGS

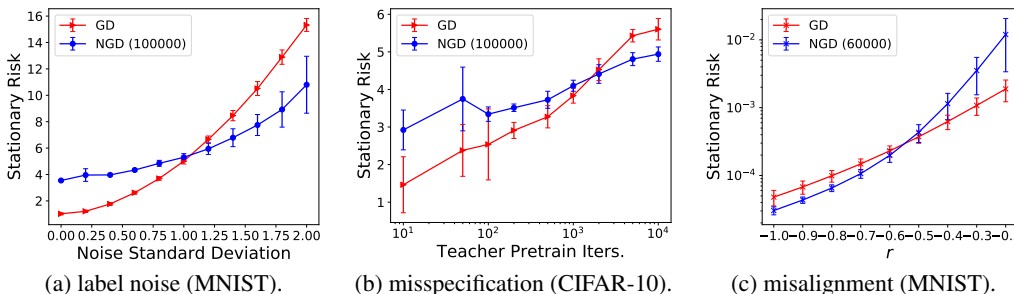

(a) label noise (MNIST).  (b) misspecification (CIFAR-10).  (c) misalignment (MNIST).

Figure 8: Comparison between NGD and GD. Error bar is one standard deviation away from mean over five independent runs. Numbers in parentheses denote amount of unlabeled examples for estimating the Fisher.

**Label Noise.** We pretrain the teacher with the full pretrain split and use 1024 examples from the distill split to fit the student. For both the student and teacher, we use a two-layer ReLU net with 80 hidden units. We corrupt the labels with isotropic Gaussian noise of varying magnitude. Figure 8(a) shows that as the noise level increases (the variance term begins to dominate), the stationary risk of both NGD and GD worsen, with GD worsening faster, which aligns with our observation in Figure 5.

**Misspecification.** We use a ResNet-20 teacher and the same two-layer ReLU student from the label noise experiment. We control the misspecification level by varying amount of pretraining

of the teacher. Intuitively, large teacher models that are trained longer should be more complex and thus likely to be outside of functions that a small two-layer student can represent (hence the problem becomes misspecified). Indeed, Figure 8(b) shows that NGD eventually achieves better generalization as the number of training steps for the teacher increases. In Appendix A.4 we report a heuristic measure of model misspecification that relates to the NTK (Jacot et al., 2018), and confirm that the quantity increases as more label noise is added or as the teacher model is trained longer.

**Misalignment.** We set the student and teacher to be the same two-layer ReLU network. We construct the teacher model by perturbing the student's initialization, the direction of which is given by $\boldsymbol{F}^r$, where $\boldsymbol{F}$ is the Fisher of the student model and $r \in [-1, 0]$. Intuitively, as $r$ decreases, the important parameters of the teacher (i.e. larger update directions) becomes misaligned with the student's gradient, and thus learning becomes more "difficult". While this analogy is rather superficial due to the non-convexity of neural network optimization, Figure 8(c) shows that as $r$ becomes smaller (setup is more misaligned), NGD begins to generalize better than GD (in terms of stationary risk).

**Interpolating between Preconditioners.** We validate our observations in Section 3 and 4 on the difference between the sample Fisher and population Fisher, and the potential benefit of interpolating between GD and NGD, in neural network experiments. Figure 9(a) shows that as we decrease the number of unlabeled data in estimating the Fisher, which renders the preconditioner closer to the sample Fisher, the stationary risk becomes more akin to that of GD, especially in the large noise setting. This agrees with our remark on sample vs. population Fisher in Section 3 and Appendix A.1.

Figure 9(b)(c) supports the bias-variance tradeoff discussed in Section 4.1 in neural network settings. In particular, we interpret the left end of the figure to correspond to the bias-dominant regime (due to the same architecture for the student and teacher), and the right end to be the variance-dominant regime (due to large label noise). Observe that at certain SNR, a preconditioner that interpolates (additively or geometrically) between GD and NGD can achieve lower stationary risk.

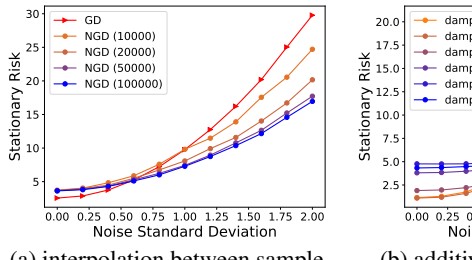
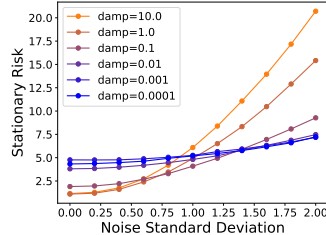
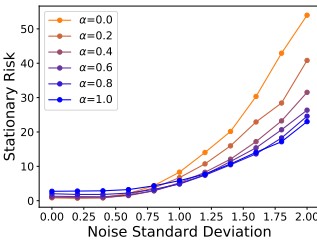

|(a) interpolation between sample and population Fisher (CIFAR-10).|(b) additive interpolation (ii) between GD and NGD (MNIST).|(c) geometric interpolation (iii) between GD and NGD (MNIST).|

Figure 9: (a) numbers in parentheses indicate the amount of unlabeled data used in estimating the Fisher $\boldsymbol{F}$; we expect the estimated Fisher to be closer to the sample Fisher when the number of unlabeled data is small. (a) additive interpolation $\boldsymbol{P} = (\boldsymbol{F} + \alpha \boldsymbol{I}_d)^{-1}$; larger damping parameter yields update closer to GD. (b) geometric interpolation $\boldsymbol{P} = \boldsymbol{F}^{-\alpha}$; larger $\alpha$ parameter yields update closer to that of NGD (blue).

## 6 DISCUSSION AND CONCLUSION

We analyzed the generalization properties of a general class of preconditioned gradient descent in overparameterized least squares regression, with particular emphasis on natural gradient descent. We identified three factors that affect the comparison of generalization performance of different optimizers, the influence of which we also empirically observed in neural network[6]. We then determined the optimal preconditioner for each factor. While the optimal $\boldsymbol{P}$ is usually not known in practice, we provided justification for common algorithmic choices by discussing the bias-variance tradeoff. Note that our current theoretical setup is limited to fixed preconditioners or those that do not alter the span of gradient, and thus does not cover many adaptive gradient methods; understanding these optimizers in similar setting would be an interesting future direction. Another important problem is to further characterize the interplay between preconditioning and explicit (e.g. weight decay[7]) or algorithmic regularization (e.g. large step size and gradient noise).

---

[6]We however note that observations in linear model may not translate to neural network: many works have illustrated such a gap (e.g., Ghorbani et al. (2019); Allen-Zhu & Li (2019); Suzuki (2020); Yang & Hu (2020)).

[7]In a companion work (Wu & Xu, 2020) we characterized the impact of $\ell_2$ regularization in similar settings.

ACKNOWLEDGEMENT

The authors would like to thank Murat A. Erdogdu, Fartash Faghri, Ryo Karakida, Yiping Lu, Jiaxin Shi, Shengyang Sun, Yusuke Tsuzuku, Guodong Zhang, Michael Zhang, Tianzong Zhang, and anonymous ICLR reviewers for helpful feedback. The authors are also grateful to Tomoya Murata for his contribution to preliminary studies on the nonparametric least squares setting.

JB and RG were supported by the CIFAR AI Chairs program. JB and DW were partially supported by LG Electronics and NSERC. AN was partially supported by JSPS Kakenhi (19K20337) and JST-PRESTO. TS was partially supported by JSPS Kakenhi (26280009, 15H05707 and 18H03201), Japan Digital Design and JST-CREST. JX was supported by a Cheung-Kong Graduate School of Business Fellowship. Resources used in preparing this research were provided, in part, by the Province of Ontario, the Government of Canada through CIFAR, and companies sponsoring the Vector Institute.

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

T­ABLE OF C­ONTENTS

## A    DISCUSSION ON ADDITIONAL RESULTS

### A.1    IMPLICIT BIAS OF GD VS. NGD

It is known that gradient descent is the steepest descent with respect to the $\ell_2$ norm, i.e. the update direction is constructed to decrease the loss under small changes in the parameters measured by the $\ell_2$ norm (Gunasekar et al., 2018a). Following this analogy, NGD is the steepest descent with respect to the KL divergence on the predictive distributions (Martens, 2014); this can be interpreted as a proximal update which penalizes how much the predictions change on the data distribution.

Intuitively, the above discussion suggests GD tend to find solution that is close to the initialization in the Euclidean distance between parameters, whereas NGD prefers solution close to the initialization in terms of function outputs on the data distribution. This observation turns out to be exact in the case of ridgeless interpolant under the squared loss, as remarked in Section 3. Moreover, Figure 1 and 10 confirms the same trend in the optimization of overparameterized neural network. In particular,

- GD results in small changes in parameters, whereas NGD results in small changes in the function.

- Preconditioning with the pseudo-inverse of the sample Fisher, i.e., $\boldsymbol{P} = (\boldsymbol{J}^\top \boldsymbol{J})^\dagger$, leads to implicit bias similar to that of GD (also noted in (Zhang et al., 2019b)), but different than NGD with the population Fisher.

- "Interpolating" between GD and NGD (green) results in properties in between GD and NGD.

**Remark.** *Qualitatively speaking, the small change in the function output is the essential reason that NGD performs well under noisy labels in the interpolation setting: NGD seeks to interpolate the training data by changing the function only "locally", so that memorizing the noisy labels has small impact on the "global" shape of the learned function (see Figure 1).*

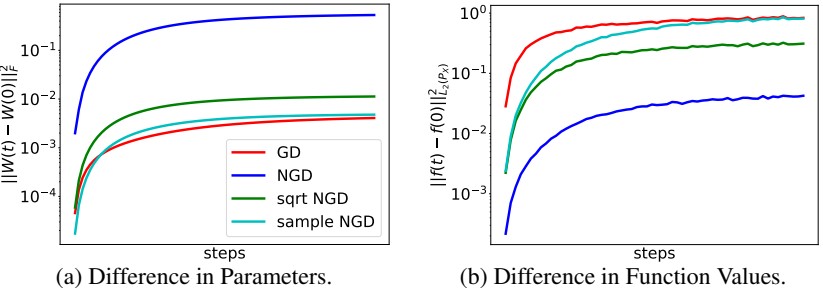

(a) Difference in Parameters.          (b) Difference in Function Values.

Figure 10: Illustration of implicit bias of GD and NGD. We set $n = 100$, $d = 50$, and regress a two-layer ReLU network with 50 hidden units towards a teacher model of the same architecture on Gaussian input. The x-axis is rescaled for each optimizer such that the final training error is below $10^{-3}$. GD finds solution with small changes in the parameters, whereas NGD finds solution with small changes in the function. Note that the sample Fisher (cyan) has implicit bias similar to GD and does not resemble NGD (population Fisher).

We note that the above observation also implies that wide neural networks trained with NGD (population Fisher) is less likely to stay in the kernel regime: the distance traveled from initialization can be large (see Figure 10(a)) and thus the Taylor expansion around the initialization is no longer accurate. In other words, the analogy between wide neural net and its linearized kernel model (which we partially employed in Section 5) may not be valid in models trained by NGD[8].

**Implicit Bias of Interpolating Preconditioners.**    We also expect that as we interpolate from GD to NGD, the distance traveled in the parameter space would gradually increase, and distance traveled in the function space would decrease. Figure 11 demonstrate that this is indeed the case for neural networks: we use the same two-layer MLP setup on MNIST as in Section 5. Observe that updates closer to GD result in smaller change in the parameters, and ones close to NGD lead to smaller change in the function outputs.

---

[8]Note that this gap is only present when the population Fisher is used; previous works have shown NTK-type global convergence for sample Fisher-related update (Zhang et al., 2019b; Cai et al., 2019).

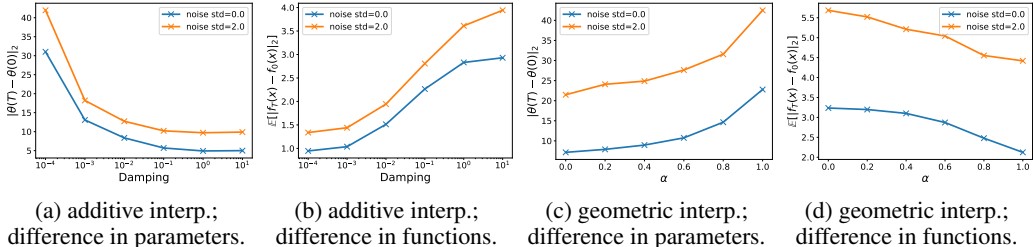

(a) additive interp.;
difference in parameters.

(b) additive interp.;
difference in functions.

(c) geometric interp.;
difference in parameters.

(d) geometric interp.;
difference in functions.

Figure 11: Illustration of the implicit bias of preconditioned gradient descent that interpolates between GD and NGD on MNIST. Note that as the update becomes more similar to NGD (smaller damping or larger $\alpha$), the distance traveled in the parameter space increases, where as the distance traveled on the output space decreases.

**Two Kinds of Second-order Optimizer.** Note that our optimal preconditioner derived in Section 3 requires knowledge of *population* second-order statistics, which we empirically approximate using extra unlabeled data. Consequently, our results suggest that different "types" of second-order information (sample vs. population) may affect generalization differently. Broadly speaking, there are two types of practical approximate second-order optimizers for neural networks. Some algorithms, such as Hessian-free optimization (Martens, 2010; Martens & Sutskever, 2012; Desjardins et al., 2013), approximate second-order matrices (typically the Hessian or Fisher) using the exact matrix on finite training examples. In high-dimensional problems, this sample-based approximation can be very different from the population quantity (e.g. it is degenerate in the overparameterized regime). Other algorithms fit a parametric approximation to the Fisher, such as diagonal (Duchi et al., 2011; Kingma & Ba, 2014), quasi-diagonal (Ollivier, 2015), or Kronecker-factored (Martens & Grosse, 2015). If the parametric assumption is accurate, these approximations are more statistically efficient and thus may lead to better approximation to the population Fisher. Our analysis reveals a difference (in generalization properties) between sample- and population-based preconditioned updates, which may also suggest a separation between the two kinds of approximate second-order optimizers. As future work, we intend to investigate this discrepancy in real-world problems.

## A.2 BIAS TERM UNDER SPECIFIC SOURCE CONDITION

Motivated by the connection between the notion of "alignment" and the *source condition* in Section 3.2, we consider a specific case of $\boldsymbol{\theta}^*$: $\boldsymbol{\Sigma_\theta} = \boldsymbol{\Sigma_X^r}$, where $r$ controls the extent of misalignment, and Theorem 3 implies that the optimal preconditioner for the bias term (well-specified) is $\boldsymbol{P} = \boldsymbol{\Sigma_X^r}$. Note that smaller $r$ corresponds to more misaligned and thus "difficult" problem, and vice versa. In this setup we have the following comparison between GD and NGD.

**Proposition** (Formal Statement of Proposition 4). *Consider the setting of Theorem 3 and $\boldsymbol{\Sigma_\theta} = \boldsymbol{\Sigma_X^r}$, then for all $r \le -1$ we have $B(\hat{\boldsymbol{\theta}}_{\boldsymbol{F}^{-1}}) \le B(\hat{\boldsymbol{\theta}}_{\boldsymbol{I}})$, whereas for all $r \ge 0$, we have $B(\hat{\boldsymbol{\theta}}_{\boldsymbol{F}^{-1}}) \ge B(\hat{\boldsymbol{\theta}}_{\boldsymbol{I}})$; the equality is achieved when features are isotropic (i.e., $\boldsymbol{\Sigma_X} = c\boldsymbol{I}_d$).*

The proposition confirms the intuition that when parameters of the teacher model $\boldsymbol{\theta}^*$ are more "aligned" with features $\boldsymbol{x}$ than the isotropic setting ($r \ge 0$), then GD achieves lower bias than NGD; on the other hand, when $\boldsymbol{\Sigma_\theta}$ is more "misaligned" than the $\boldsymbol{\Sigma_X^{-1}}$ case ($r \le -1$), then NGD is guaranteed to be advantageous for the bias term. We therefore expect a transition from the NGD-dominated to GD-dominated regime for some $r \in (-1, 0)$. The exact value of $r$ for such transition depends on the spectral distribution of $\boldsymbol{\Sigma_X}$ and varies case-by-case (one would need to specifically evaluate the equality (D.9)). To give a concrete example, when $\boldsymbol{\Sigma_X}$ has a simple block structure, we can explicitly determine the the transition point $r^* \in (-1, 0)$, as shown by the following corollary.

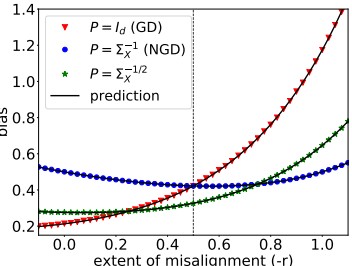

Figure 12: We set $\boldsymbol{\Sigma_\theta} = \boldsymbol{\Sigma_X^r}$, $\gamma = 2$, $\kappa = 20$, and plot the stationary bias (well-specified) under varying $r$.

**Corollary 9.** *Assume $\boldsymbol{\Sigma_\theta} = \boldsymbol{\Sigma_X^r}$, and eigenvalues of $\boldsymbol{\Sigma_X}$ come from two equally-weighted point masses with $\kappa \triangleq \lambda_{\max}(\boldsymbol{\Sigma_X})/\lambda_{\min}(\boldsymbol{\Sigma_X})$. WLOG we take $\operatorname{tr}(\boldsymbol{\Sigma_X})/d = 1$. Then given $r^* = -\ln c_{\kappa,\gamma}/\ln \kappa$ (see Appendix D.10 for definition), we have $B(\hat{\boldsymbol{\theta}}_{\boldsymbol{I}}) \gtreqless B(\hat{\boldsymbol{\theta}}_{\boldsymbol{F}^{-1}})$ if and only if $r \lesseqgtr r^*$.*

**Remark.** *When $\gamma = 2$, the transition happens at $r^* = -1/2$ which is independent of the condition number $\kappa$, as indicated by the dashed line in Figure 12. However for other $\gamma > 1$, $r^*$ generally relates to both $\gamma$ and $\kappa$.*

Our characterization above is supported by Figure 12, in which we plot the bias term under varying extent of misalignment (controlled by $r$) in the setting of Corollary 9. Observe that as we construct the teacher model to be more "misaligned" (and thus difficult to learn) by decreasing $r$, NGD (blue) achieves lower bias compared to GD (red), and vice versa.

### A.3  ESTIMATING THE POPULATION FISHER

Our analysis on linear model considers the idealized setup with access to the exact population Fisher, which can be estimated using additional unlabeled data. In this section we discuss how our result in Section 3 and Section 4 are affected when the population covariance is approximated from $N$ i.i.d. (unlabeled) samples $\boldsymbol{X}_u \in \mathbb{R}^{N \times d}$. For the ridgeless interpolant we have the following result on the substitution error in replacing the true population covariance with the sample estimate.

**Proposition 10.** *Given (A1)(A3) and $N/d \to \psi > 1$ as $d \to \infty$, let $\hat{\boldsymbol{\Sigma}}_{\boldsymbol{X}} = \boldsymbol{X}_u^\top \boldsymbol{X}_u / N$, we have*

*(a) $\|\boldsymbol{\Sigma}_{\boldsymbol{X}} - \hat{\boldsymbol{\Sigma}}_{\boldsymbol{X}}\|_2 = O(\psi^{-1/2})$ almost surely.*

*(b) Denote the stationary bias and variance of NGD (with the exact population Fisher) as $B^*$ and $V^*$, respectively, and the bias and variance of the preconditioned update using the approximate Fisher $\hat{\boldsymbol{F}} = \hat{\boldsymbol{\Sigma}}_{\boldsymbol{X}}$ as $\hat{B}$ and $\hat{V}$, respectively. Let $0 < \epsilon < 1$ be the desired accuracy. Then $\psi = \Theta(\epsilon^{-2})$ suffices to achieve $|B^* - \hat{B}| < \epsilon$ and $|V^* - \hat{V}| < \epsilon$.*

Proposition 10 entails that when the preconditioner is a sample estimate of the Fisher $\hat{\boldsymbol{F}}$ (based on unlabeled data), we can approach the stationary bias and variance of the population Fisher at a rate of $\psi^{-1/2}$ as we increase the number of unlabeled data $N$ linearly in the dimensionality $d$. In other words, any non-vanishing accuracy $\epsilon$ can be achieved with finite $\psi$ (to push $\epsilon \to 0$, additional logarithmic factor is required, e.g. $N = O(d \log d)$, which is beyond the proportional limit).

On the other hand, for our result in Section 4.3, (Murata & Suzuki, 2020, Lemma A.5) directly implies that setting $N = \Theta(n^s \log n)$ is sufficient to approximate the population covariance operator (i.e., so that $\|\Sigma^{1/2}\Sigma_{N,\lambda}^{-1/2}\| = O(1)$). Finally, we remark that our analysis above does not impose any structural assumptions on the estimated matrix. When the Fisher exhibits certain structures (e.g. Kronecker factorization (Martens & Grosse, 2015)), then estimation can be more sample-efficient. For analysis on such structured approximations of the Fisher see Karakida & Osawa (2020).

### A.4  INTERPRETATION OF $\sqrt{\boldsymbol{y}^\top \boldsymbol{K}^{-1} \boldsymbol{y} / n}$

As a heuristic measure of model misspecification, in Figure 13 we report $\sqrt{\boldsymbol{y}^\top \boldsymbol{K}^{-1} \boldsymbol{y} / n}$ studied in Arora et al. (2019b), where $\boldsymbol{y}$ is the label vector and $\boldsymbol{K}$ is the NTK matrix (Jacot et al., 2018) of the student model. This quantity relates to kernel-based alignment measures (Cristianini et al., 2001), and in the context of neural network optimization, it can be interpreted as a proxy for measuring how much signal and noise are distributed along the eigendirections of the NTK (e.g., see Li et al. (2019); Dong et al. (2019); Su & Yang (2019)). Roughly speaking, large $\sqrt{\boldsymbol{y}^\top \boldsymbol{K}^{-1} \boldsymbol{y} / n}$ implies that the problem is "difficult" to learn by the student model via GD, and vice versa.

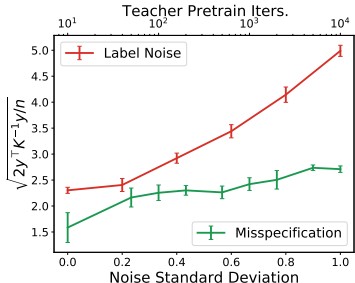

Figure 13: $\sqrt{\boldsymbol{y}^\top \boldsymbol{K}^{-1} \boldsymbol{y} / n}$ on two-layer neural network (CIFAR-10).

Here we give a heuristic argument on how this quantity relates to label noise and misspecification. For the ridgeless regression model considered in Section 3, write $y_i = f^*(\boldsymbol{x}_i) + f^c(\boldsymbol{x}_i) + \varepsilon_i$, where $f^*(\boldsymbol{x}) = \boldsymbol{x}^\top \boldsymbol{\theta}^*$, $f^c$ is the misspecified component, and $\varepsilon_i$ is the label noise, then we have

$$\mathbb{E}\big[\boldsymbol{y}^\top \boldsymbol{K}^{-1} \boldsymbol{y}\big] = \mathbb{E}\left[\left\|(\boldsymbol{X}\boldsymbol{X}^\top)^{-1/2}(f^*(\boldsymbol{X}) + f^c(\boldsymbol{X}) + \boldsymbol{\varepsilon})\right\|_2^2\right]$$

$$\stackrel{(i)}{\approx} \mathrm{tr}\big(\boldsymbol{\theta}^*\boldsymbol{\theta}^{*\top}\boldsymbol{X}^\top(\boldsymbol{X}\boldsymbol{X}^\top)^{-1}\boldsymbol{X}\big) + (\sigma^2 + \sigma_c^2)\mathrm{tr}\big((\boldsymbol{X}\boldsymbol{X}^\top)^{-1}\big), \qquad (A.1)$$

where we heuristically replaced the misspecified component with i.i.d. noise of the same variance $\sigma_c^2$. The first term of (A.1) resembles an RKHS norm of the target $\boldsymbol{\theta}^*$, whereas the second term is small when the feature matrix is well-conditioned or when the level of label noise $\sigma$ and misspecification $\sigma_c^2$ is small (note that these are conditions under which GD achieves good generalization). We may expect similar behavior for neural networks close to the kernel regime. This provides a non-rigorous explanation of the trend observed in Figure 13: as we increase the level of label noise or model misspecification (by pretraining the teacher for more steps), the quantity of interest becomes larger.

## A.5 NON-MONOTONICITY OF BIAS TERM W.R.T. TIME

Many previous works on the high-dimensional characterization of linear regression assumed a random effects model with an isotropic prior on the true parameters (Dobriban et al., 2018; Hastie et al., 2019), which may not hold in practice. As an example of the limitation of this assumption, note that when $\boldsymbol{\Sigma}_{\boldsymbol{\theta}} = \boldsymbol{I}_d$, it can be shown that the expected bias $B(\hat{\boldsymbol{\theta}}(t))$ monotonically decreases through time (see proof of Proposition 7). In contrast, when the target parameters do not follow an isotropic prior, the bias of GD can exhibit non-monotonicity, which gives rise to the "epoch-wise double descent" phenomenon also observed in deep learning (Nakkiran et al., 2019; Ishida et al., 2020).

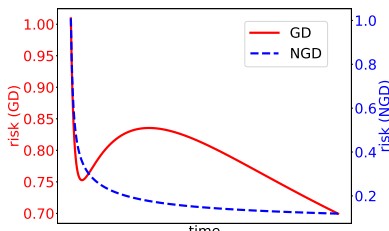

Figure 14: Epoch-wise double descent. Note that non-monotonicity of the bias w.r.t. time is present in GD but not NGD.

We empirically demonstrate this non-monotonicity when the model is close to the interpolation threshold in Figure 14. We set eigenvalues of $\boldsymbol{\Sigma}_{\boldsymbol{X}}$ to be two point masses with $\kappa_X = 32$, $\boldsymbol{\Sigma}_{\boldsymbol{\theta}} = \boldsymbol{\Sigma}_{\boldsymbol{X}}^{-1}$ and $\gamma = 16/15$. Note that the GD trajectory (red) exhibits non-monotonicity in the bias term, whereas for NGD the bias is monotonically decreasing through time (which we confirm in the proof of Proposition 7). We remark that this mechanism of epoch-wise double descent may not relate the empirical findings in deep neural networks (the robustness of which is also largely unknown), in which it is typically speculated that the variance term exhibits non-monotonicity.

## B ADDITIONAL RELATED WORKS

**Implicit Regularization in Optimization.** In overparameterized linear models, GD finds the minimum $\ell_2$ norm solution under many loss functions. For the more general mirror descent, the implicit bias is determined by the Bregman divergence of the update (Gunasekar et al., 2018b; Suggala et al., 2018; Azizan et al., 2019). Under the exponential or logistic loss, recent works showed that GD finds the max-margin direction in various models (Ji & Telgarsky, 2018; 2019; Soudry et al., 2018; Lyu & Li, 2019; Chizat & Bach, 2020). The implicit bias of Adagrad has been analyzed under similar setting (Qian & Qian, 2019). Implicit regularization also relates to the model architecture; examples include matrix factorization (Gunasekar et al., 2017; Saxe et al., 2013; Gidel et al., 2019; Arora et al., 2019a) and certain stylized neural networks (Li et al., 2017; Gunasekar et al., 2018b; Williams et al., 2019; Woodworth et al., 2020). For wide networks in the kernel regime (Jacot et al., 2018), the implicit bias of GD relates to properties of the neural tangent kernel (NTK) (Xie et al., 2016; Arora et al., 2019b; Bietti & Mairal, 2019). Finally, we note that the implicit bias of GD is not always explained by the minimum norm property (Razin & Cohen, 2020; Dauber et al., 2020).

**Asymptotics of Interpolating Estimators.** In Section 3 we analyzed overparameterized estimators that interpolate the training data. Recent works have shown that interpolation may not lead to overfitting (Liang & Rakhlin, 2018; Belkin et al., 2018c;b; Bartlett et al., 2019), and the optimal risk may be achieved under no regularization and extreme overparameterization (Belkin et al., 2018a; Xu & Hsu, 2019). The asymptotic risk of overparameterized models has been characterized in various settings, such as linear regression (Karoui, 2013; Dobriban et al., 2018; Hastie et al., 2019), random features regression (Mei & Montanari, 2019; Gerace et al., 2020; Dhifallah & Lu, 2020; Adlam & Pennington, 2020), max-margin classification (Montanari et al., 2019; Deng et al., 2019), and certain neural networks (Louart et al., 2018; Ba et al., 2020). Our analysis is based on random matrix theory results developed in Rubio & Mestre (2011); Ledoit & Péché (2011). Similar tools can also be used to study the gradient descent dynamics of linear regression (Liao & Couillet, 2018; Ali et al., 2019).

# C  ADDITIONAL FIGURES

## C.1  OVERPARAMETERIZED LINEAR REGRESSION

**Non-monotonicity of the Risk.**  Under our generalized (anisotropic) assumption on the covariance of the features and the target, both the bias and the variance term can exhibit non-monotonicity w.r.t. the overparameterization level $\gamma > 1$: in Figure 15 we observe two peaks in the bias term and three peaks in the variance term. In contrast, it is known that when $\mathbf{\Sigma_X} = \mathbf{I}_d$, both the bias and variance are *monotone* in the overparameterized regime (e.g., Hastie et al. (2019)).

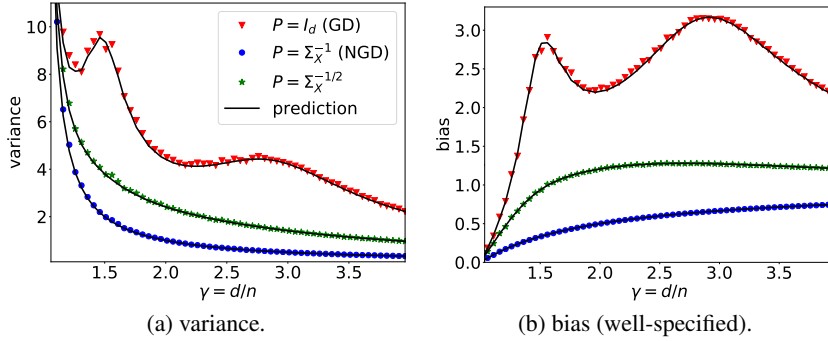

(a) variance.  (b) bias (well-specified).

Figure 15: Illustration of the "multiple-descent" curve of the risk for $\gamma > 1$. We take $n = 300$, eigenvalues of $\mathbf{\Sigma_X}$ as three equally-spaced point masses with $\kappa_X = 5000$ and $\|\mathbf{\Sigma_X}\|_F^2 = d$, and $\mathbf{\Sigma_\theta} = \mathbf{\Sigma_X^{-1}}$ (misaligned). Note that for GD, both the bias and the variance are highly non-monotonic for $\gamma > 1$.

**Additional Figures for Section 3 and 4.**  We include additional figures on (a) well-specified bias when $\mathbf{\Sigma_\theta} = \mathbf{I}_d$ (GD is optimal); (b) misspecified bias under unobserved features (predicted by Corollary 2); (c) bias-variance tradeoff by interpolating between preconditioners (SNR=5). As shown in Figure 16 and 17, in all cases the experimental values match the theoretical predictions.

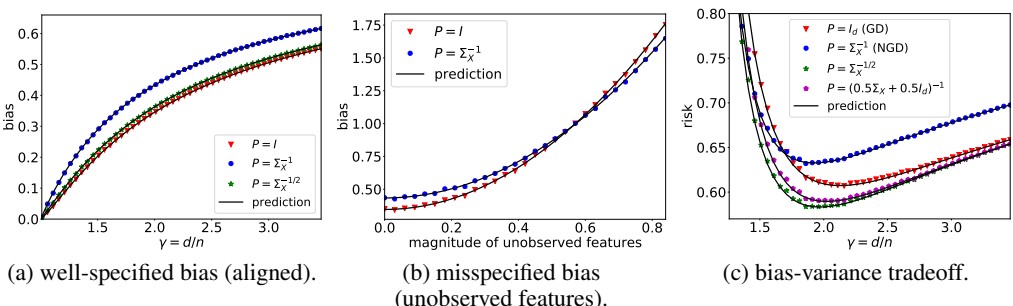

(a) well-specified bias (aligned).  (b) misspecified bias (unobserved features).  (c) bias-variance tradeoff.

Figure 16: We set eigenvalues of $\mathbf{\Sigma_X}$ as a uniform distribution with $\kappa_X = 20$ and $\|\mathbf{\Sigma_X}\|_F^2 = d$.

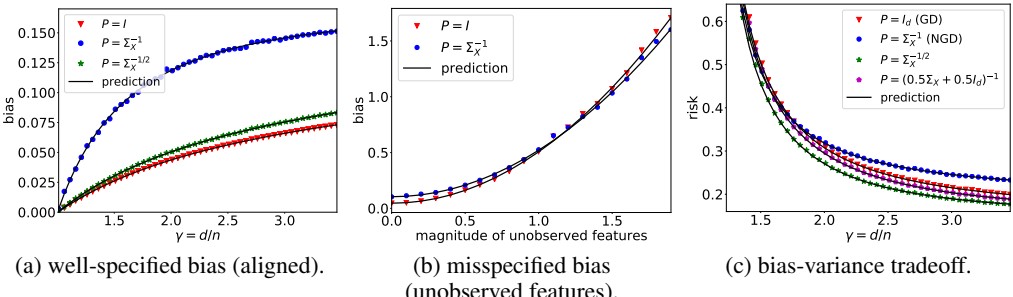

(a) well-specified bias (aligned).  (b) misspecified bias (unobserved features).  (c) bias-variance tradeoff.

Figure 17: We construct eigenvalues of $\mathbf{\Sigma_X}$ with a polynomial decay: $\lambda_i(\mathbf{\Sigma_X}) = i^{-1}$ and then rescale the eigenvalues such that $\kappa_X = 500$ and $\|\mathbf{\Sigma_X}\|_F^2 = d$.

**Early Stopping Risk.** Figure 18 compares the stationary risk with the optimal early stopping risk under varying misalignment level. We set $\Sigma_{\theta} = \Sigma_X^r$ and vary $r$ from 0 to -1. As discussed in Section 3.2 smaller $\alpha$ entails more "misaligned" teacher, and vice versa. Note that as the problem becomes more misaligned, NGD achieves lower stationary and early stopping risk.

Figure 19 reports the optimal early stopping risk under misspecification (same trend can be obtained when the x-axis is label noise). In contrast to the stationary risk (Figure 4), GD can be advantageous under early stopping even with large extent of misspecification (for isotropic teacher). This aligns with our finding in Section 4.2 that early stopping reduces the variance and the misspecified bias.

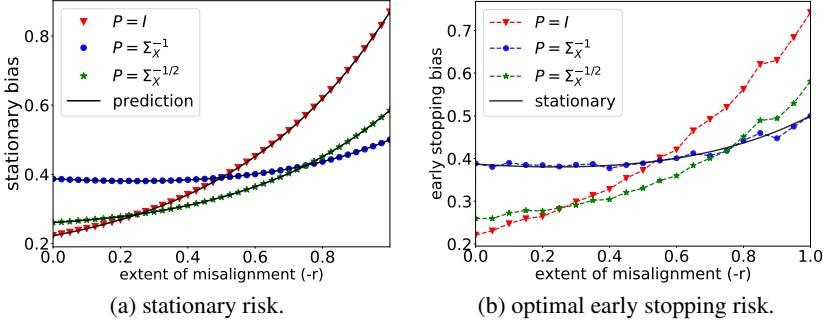

(a) stationary risk.                (b) optimal early stopping risk.

Figure 18: Well-specified bias against different extent of "alignment". We set $n = 300$, eigenvalues of $\Sigma_X$ as two point masses with $\kappa_X = 20$, and take $\Sigma_{\theta} = \Sigma_X^r$ and vary $r$ from -1 to 0. (a) GD achieves lower bias when $\Sigma_{\theta}$ is isotropic, NGD dominates when $\Sigma_X = \Sigma_{\theta}^{-1}$, whereas $P = \Sigma_X^{-1/2}$ (interpolates between GD and NGD) is advantageous in between. (b) optimal early stopping bias follows similar trend as stationary bias.

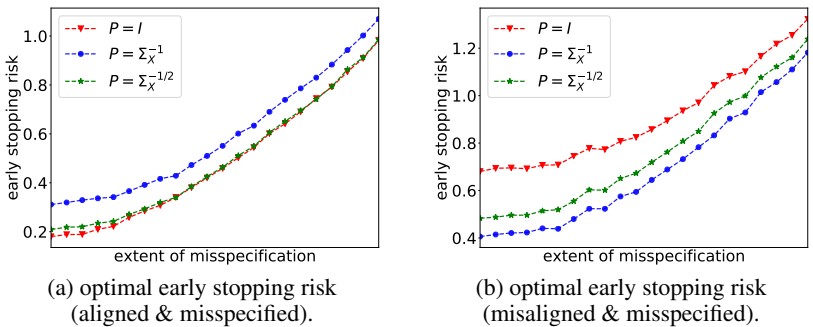

(a) optimal early stopping risk        (b) optimal early stopping risk
(aligned & misspecified).              (misaligned & misspecified).

Figure 19: Optimal early stopping risk vs. increasing model misspecification. We follow the same setup as Figure 5(c). (a) $\Sigma_{\theta} = I_d$ (favors GD); unlike Figure 5(c), GD has lower early stopping risk even under large extent of misspecification. (b) $\Sigma_{\theta} = \Sigma_X^{-1}$ (favors NGD); NGD is also advantageous under early stopping.

## C.2 RKHS SETTING

We simulate the optimization in the coordinates of RKHS via a finite-dimensional approximation (using extra unlabeled data). In particular, we consider the teacher model in the form of $f^*(\boldsymbol{x}) = \sum_{i=1}^N h_i \mu_i^r \phi_i(\boldsymbol{x})$ for square summable $\{h_i\}_{i=1}^N$, in which $r$ controls the "difficulty" of the learning problem. We find $\{\mu_i\}_{i=1}^N$ and $\{\phi_i\}_{i=1}^N$ by solving the eigenfunction problem for some kernel $k$. The student model takes the form of $f(\boldsymbol{x}) = \sum_{i=1}^N \frac{a_i}{\sqrt{\mu_i}} \phi_i(\boldsymbol{x})$ and we optimize the coefficients $\{a_i\}_{i=1}^N$ via the preconditioned update (4.1). We set $n = 1000$, $d = 5$, $N = 2500$ and consider the inverse multiquadratic (IMQ) kernel: $k(\boldsymbol{x}, \boldsymbol{y}) = \frac{1}{\sqrt{1+\|\boldsymbol{x}-\boldsymbol{y}\|_2^2}}$.

Recall that Theorem 8 suggests that for small $r$, i.e. "difficult" problem, the damping coefficient $\lambda$ would need to be small (which makes the update NGD-like), and vice versa. This result is (qualitatively) supported by Figure 20, from which we can see that small $\lambda$ is beneficial when $r$ is small, and vice versa. We remark that this observed trend is rather fragile and sensitive to various hyperparameters, and we leave a comprehensive characterization of this observation as future work.

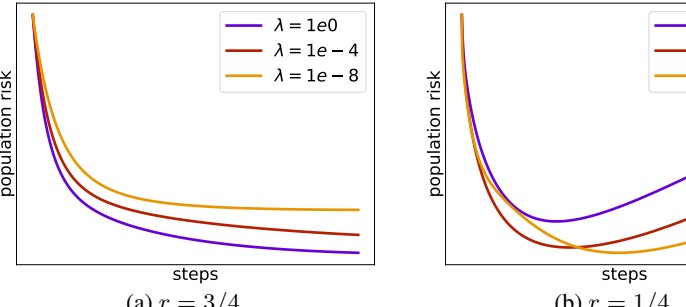

(a) $r = 3/4$.          (b) $r = 1/4$.

Figure 20: Population risk of the preconditioned update in RKHS that interpolates between GD and NGD. We use the IMQ kernel and set $n = 1000$, $d = 5$, $N = 2500$, $\sigma^2 = 5 \times 10^{-4}$. The x-axis has been rescaled for each curve and thus convergence speed is not directly comparable. Note that (a) large $\lambda$ (i.e., GD-like update) is beneficial when $r$ is large, and (b) small $\lambda$ (i.e., NGD-like update) is beneficial when $r$ is small.

### C.3 NEURAL NETWORK

**Label Noise.** In Figure 21, (a) we observe the same phenomenon on CIFAR-10 that NGD generalizes better as more label noise is added to the labels, and vice versa. Figure 21 (b) shows that in all cases with varying amounts of label noise, the early stopping risk is however worse than that of GD. This agrees with the observation in Section 4 and Figure 19(a) that early stopping can potentially favor GD due to the reduced variance.

**Misalignment.** In Figure 21(c)(d) we confirm the finding in Proposition 7 and Figure 18(b) in neural networks under synthetic data: we consider 50-dimensional Gaussian input, and both the teacher and the student model are two-layer ReLU networks with 50 hidden units. We construct the teacher by perturbing the initialization of the student as described in Section 5. As $r$ approaches -1 (problem more "misaligned"), NGD achieves lower early stopping risk (Figure 21(d)), whereas GD dominates the early stopping risk in less misaligned setting ( 21(c)). We note that this phenomenon is difficult to observe in practical neural network training on real-world data, which may be partially due to the fragility of the analogy between neural nets and linear models, especially under NGD (discussed in Appendix A.1).

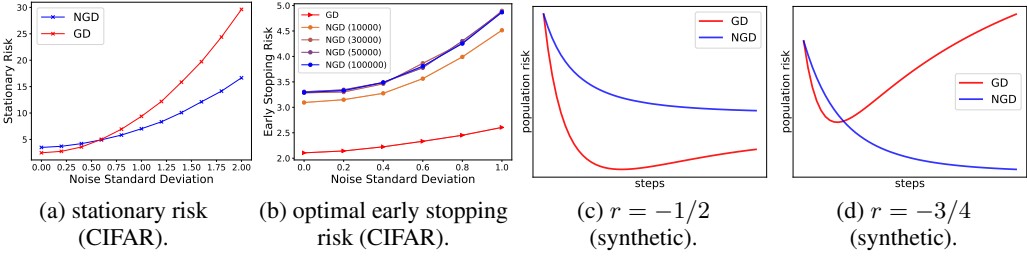

(a) stationary risk    (b) optimal early stopping    (c) $r = -1/2$    (d) $r = -3/4$
(CIFAR).        risk (CIFAR).       (synthetic).      (synthetic).

Figure 21: (a)(b) Additional label noise experiment on CIFAR-10. (c)(d) Population risk of two-layer neural networks in the misalignment setup with synthetic Gaussian data. We set $n = 200$, $d = 50$, the damping coefficient $\lambda = 10^{-6}$, and both the student and the teacher are two-layer ReLU networks with 50 hidden units. The x-axis and the learning rate have been rescaled for each curve (i.e., optimization speed not comparable). When $r$ is sufficiently small, NGD achieves lower early stopping risk, whereas GD is beneficial for larger $r$.

## D  PROOFS AND DERIVATIONS

### D.1  MISSING DERIVATIONS IN SECTION 3

**Gradient Flow of Preconditioned Updates.**  Given positive definite $P$ and $\gamma > 1$, one may check that the gradient flow solution at time $t$ can be written as

$$\boldsymbol{\theta}_{\boldsymbol{P}}(t) = \boldsymbol{P}\boldsymbol{X}^\top\left[\boldsymbol{I}_n - \exp\left(-\frac{t}{n}\boldsymbol{X}\boldsymbol{P}\boldsymbol{X}^\top\right)\right](\boldsymbol{X}\boldsymbol{P}\boldsymbol{X}^\top)^{-1}\boldsymbol{y}.$$

Taking $t \to \infty$ yields the stationary solution $\hat{\boldsymbol{\theta}}_{\boldsymbol{P}} = \boldsymbol{P}\boldsymbol{X}^\top(\boldsymbol{X}\boldsymbol{P}\boldsymbol{X}^\top)^{-1}\boldsymbol{y}$. We remark that the damped inverse of the sample Fisher $\boldsymbol{P} = (\boldsymbol{X}\boldsymbol{X}^\top + \lambda\boldsymbol{I}_d)^{-1}$ leads to the same minimum-norm solution as GD $\hat{\boldsymbol{\theta}}_{\boldsymbol{I}} = \boldsymbol{X}^\top(\boldsymbol{X}\boldsymbol{X}^\top)^{-1}\boldsymbol{y}$ since $\boldsymbol{P}\boldsymbol{X}^\top$ and $\boldsymbol{X}$ share the same eigenvectors. On the other hand, when $\boldsymbol{P}$ is the pseudo-inverse of the sample Fisher $(\boldsymbol{X}\boldsymbol{X}^\top)^\dagger$ which is not full-rank, the trajectory can be obtained via the variation of constants formula:

$$\boldsymbol{\theta}(t) = \left[\frac{t}{n}\sum_{k=0}^{\infty}\frac{1}{(k+1)!}\left(-\frac{t}{n}\boldsymbol{X}^\top(\boldsymbol{X}\boldsymbol{X}^\top)^{-1}\boldsymbol{X}\right)^k\right]\boldsymbol{X}^\top(\boldsymbol{X}\boldsymbol{X}^\top)^{-1}\boldsymbol{y},$$

for which taking the large $t$ limit also yields the minimum-norm solution $\boldsymbol{X}^\top(\boldsymbol{X}\boldsymbol{X}^\top)^{-1}\boldsymbol{y}$.

**Minimum $\|\boldsymbol{\theta}\|_{\boldsymbol{P}^{-1}}$ Norm Interpolant.**  For positive definite $\boldsymbol{P}$ and the corresponding stationary solution $\hat{\boldsymbol{\theta}}_{\boldsymbol{P}} = \boldsymbol{P}\boldsymbol{X}^\top(\boldsymbol{X}\boldsymbol{P}\boldsymbol{X}^\top)^{-1}\boldsymbol{y}$, note that given any other interpolant $\hat{\boldsymbol{\theta}}'$, we have $(\hat{\boldsymbol{\theta}}_{\boldsymbol{P}} - \hat{\boldsymbol{\theta}}')\boldsymbol{P}^{-1}\hat{\boldsymbol{\theta}}_{\boldsymbol{P}} = 0$ because both $\hat{\boldsymbol{\theta}}_{\boldsymbol{P}}$ and $\hat{\boldsymbol{\theta}}'$ achieves zero empirical risk. Hence $\|\hat{\boldsymbol{\theta}}'\|_{\boldsymbol{P}^{-1}}^2 - \|\hat{\boldsymbol{\theta}}_{\boldsymbol{P}}\|_{\boldsymbol{P}^{-1}}^2 = \|\hat{\boldsymbol{\theta}}' - \hat{\boldsymbol{\theta}}_{\boldsymbol{P}}\|_{\boldsymbol{P}^{-1}}^2 \geq 0$. This confirms that $\hat{\boldsymbol{\theta}}_{\boldsymbol{P}}$ is the unique minimum $\|\boldsymbol{\theta}\|_{\boldsymbol{P}^{-1}}$ norm solution.

### D.2  PROOF OF THEOREM 1

**Proof.**  By the definition of the variance term and the stationary $\hat{\boldsymbol{\theta}}$,

$$V(\hat{\boldsymbol{\theta}}) = \mathrm{tr}\left(\mathrm{Cov}(\hat{\boldsymbol{\theta}})\boldsymbol{\Sigma}_{\boldsymbol{X}}\right) = \sigma^2\mathrm{tr}\left(\boldsymbol{P}\boldsymbol{X}^\top(\boldsymbol{X}\boldsymbol{P}\boldsymbol{X}^\top)^{-2}\boldsymbol{X}\boldsymbol{P}\boldsymbol{\Sigma}_{\boldsymbol{X}}\right).$$

Write $\bar{\boldsymbol{X}} = \boldsymbol{X}\boldsymbol{P}^{1/2}$. Similarly, we define $\boldsymbol{\Sigma}_{\boldsymbol{X}\boldsymbol{P}} = \boldsymbol{P}^{1/2}\boldsymbol{\Sigma}_{\boldsymbol{X}}\boldsymbol{P}^{1/2}$. It is clear that the equation above simplifies to

$$V(\hat{\boldsymbol{\theta}}_{\boldsymbol{P}}) = \sigma^2\mathrm{tr}\left(\bar{\boldsymbol{X}}^\top(\bar{\boldsymbol{X}}\bar{\boldsymbol{X}}^\top)^{-2}\bar{\boldsymbol{X}}\boldsymbol{\Sigma}_{\boldsymbol{X}\boldsymbol{P}}\right).$$

The analytic expression of the variance term follows from a direct application of (Hastie et al., 2019, Thorem 4), in which conditions on the population covariance are satisfied by (A2).

Taking the derivative of $m(-\lambda)$ yields

$$m'(-\lambda) = \left(\frac{1}{m^2(-\lambda)} - \gamma\int\frac{\tau^2}{(1+\tau m(-\lambda))^2}\mathrm{d}\boldsymbol{H}_{\boldsymbol{X}\boldsymbol{P}}(\tau)\right)^{-1}.$$

Plugging the quantity into the expression of the variance (omitting the scaling $\sigma^2$ and constant shift),

$$\frac{m'(-\lambda)}{m^2(-\lambda)} = \left(1 - \gamma m^2(-\lambda)\int\frac{\tau^2}{(1+\tau m(-\lambda))^2}\mathrm{d}\boldsymbol{H}_{\boldsymbol{X}\boldsymbol{P}}(\tau)\right)^{-1}.$$

From the monotonicity of $\frac{x}{1+x}$ on $x > 0$ or the Jensen's inequality we know that

$$1 - \gamma\int\left(\frac{\tau m(-\lambda)}{1+\tau m(-\lambda)}\right)^2\mathrm{d}\boldsymbol{H}_{\boldsymbol{X}\boldsymbol{P}}(\tau) \leq 1 - \gamma\left(\int\frac{\tau m(-\lambda)}{1+\tau m(-\lambda)}\mathrm{d}\boldsymbol{H}_{\boldsymbol{X}\boldsymbol{P}}(\tau)\right)^2.$$

Taking $\lambda \to 0$ and omitting the scalar $\sigma^2$, the RHS evaluates to $1 - 1/\gamma$. We thus arrive at the lower bound $V \geq (\gamma - 1)^{-1}$. Note that the equality is only achieved when $\boldsymbol{H}_{\boldsymbol{X}\boldsymbol{P}}$ is a point mass, i.e. $\boldsymbol{P} = \boldsymbol{\Sigma}_{\boldsymbol{X}}^{-1}$. In other words, the minimum variance is achieved by NGD. As a verification, the variance of the NGD solution $\hat{\boldsymbol{\theta}}_{\boldsymbol{F}^{-1}}$ agrees with the calculation for the case were the features are isotropic (Hastie et al., 2019, A.3). □

## D.3 PROOF OF COROLLARY 2

**Proof.** Via calculation similar to (Hastie et al., 2019, Section 5), the bias can be decomposed as

$$
\mathbb{E}\left[B(\hat{\boldsymbol{\theta}}_{\boldsymbol{P}})\right] = \mathbb{E}_{\boldsymbol{x},\hat{\boldsymbol{x}},\boldsymbol{\theta}^*,\boldsymbol{\theta}^c}\left[\left(\boldsymbol{x}^\top \boldsymbol{P}\boldsymbol{X}^\top\left(\boldsymbol{X}\boldsymbol{P}\boldsymbol{X}^\top\right)^{-1}(\boldsymbol{X}\boldsymbol{\theta}^* + \boldsymbol{X}^c\boldsymbol{\theta}^c) - (\boldsymbol{x}^\top\boldsymbol{\theta}^* + \hat{\boldsymbol{x}}^\top\boldsymbol{\theta}^c)\right)^2\right]
$$

$$
\overset{(i)}{=}\mathbb{E}_{\boldsymbol{x},\boldsymbol{\theta}^*}\left[\left(\boldsymbol{x}^\top \boldsymbol{P}\boldsymbol{X}^\top\left(\boldsymbol{X}\boldsymbol{P}\boldsymbol{X}^\top\right)^{-1}\boldsymbol{X}\boldsymbol{\theta}^* - \boldsymbol{x}^\top\boldsymbol{\theta}^*\right)^2\right] + \mathbb{E}_{\boldsymbol{x}^c,\boldsymbol{\theta}^x}\left[(\hat{\boldsymbol{x}}^\top\boldsymbol{\theta}^c)^2\right]
$$

$$
+ \mathbb{E}_{\boldsymbol{x},\boldsymbol{\theta}^c}\left[\left(\boldsymbol{x}^\top \boldsymbol{P}\boldsymbol{X}^\top\left(\boldsymbol{X}\boldsymbol{P}\boldsymbol{X}^\top\right)^{-1}\boldsymbol{X}^c\boldsymbol{\theta}^c\boldsymbol{\theta}^{c\top}\boldsymbol{X}^{c\top}\left(\boldsymbol{X}\boldsymbol{P}\boldsymbol{X}^\top\right)^{-1}\boldsymbol{X}\boldsymbol{P}\boldsymbol{x}\right)^2\right]
$$

$$
\overset{(ii)}{\to} B_{\boldsymbol{\theta}}(\hat{\boldsymbol{\theta}}_{\boldsymbol{P}}) + \frac{1}{d^c}\mathrm{tr}(\boldsymbol{\Sigma}_{\boldsymbol{X}}^c\boldsymbol{\Sigma}_{\boldsymbol{\theta}}^c)(1 + V(\hat{\boldsymbol{\theta}}_{\boldsymbol{P}})),
$$

where we used the independence of $\boldsymbol{x}, \hat{\boldsymbol{x}}$ and $\boldsymbol{\theta}^*, \boldsymbol{\theta}^c$ in (i), and (A1-3) as well as the definition of the well-specified bias $B_{\boldsymbol{\theta}}(\hat{\boldsymbol{\theta}}_{\boldsymbol{P}})$ and variance $V(\hat{\boldsymbol{\theta}}_{\boldsymbol{P}})$ in (ii). $\qquad\square$

## D.4 PROOF OF THEOREM 3

**Proof.** By the definition of the bias term (note that $\boldsymbol{\Sigma}_{\boldsymbol{X}}, \boldsymbol{\Sigma}_{\boldsymbol{\theta}}, \boldsymbol{P}$ are all positive semi-definite),

$$
B(\hat{\boldsymbol{\theta}}_{\boldsymbol{P}}) = \mathbb{E}_{\boldsymbol{\theta}^*}\left[\left\|\boldsymbol{P}\boldsymbol{X}^\top(\boldsymbol{X}\boldsymbol{P}\boldsymbol{X}^\top)^{-1}\boldsymbol{X}\boldsymbol{\theta}_* - \boldsymbol{\theta}^*\right\|_{\boldsymbol{\Sigma}_{\boldsymbol{X}}}^2\right]
$$

$$
= \frac{1}{d}\mathrm{tr}\left(\boldsymbol{\Sigma}_{\boldsymbol{\theta}}\left(\boldsymbol{I}_d - \boldsymbol{P}\boldsymbol{X}^\top(\boldsymbol{X}\boldsymbol{P}\boldsymbol{X}^\top)^{-1}\boldsymbol{X}\right)^\top\boldsymbol{\Sigma}_{\boldsymbol{X}}\left(\boldsymbol{I}_d - \boldsymbol{P}\boldsymbol{X}^\top(\boldsymbol{X}\boldsymbol{P}\boldsymbol{X}^\top)^{-1}\boldsymbol{X}\right)\right)
$$

$$
\overset{(i)}{=} \frac{1}{d}\mathrm{tr}\left(\boldsymbol{\Sigma}_{\boldsymbol{\theta}/\boldsymbol{P}}\left(\boldsymbol{I}_d - \bar{\boldsymbol{X}}^\top(\bar{\boldsymbol{X}}\bar{\boldsymbol{X}}^\top)^{-1}\bar{\boldsymbol{X}}\right)^\top\boldsymbol{\Sigma}_{\boldsymbol{X}\boldsymbol{P}}\left(\boldsymbol{I}_d - \bar{\boldsymbol{X}}^\top(\bar{\boldsymbol{X}}\bar{\boldsymbol{X}}^\top)^{-1}\bar{\boldsymbol{X}}\right)\right)
$$

$$
\overset{(ii)}{=} \lim_{\lambda\to 0_+}\frac{\lambda^2}{d}\mathrm{tr}\left(\boldsymbol{\Sigma}_{\boldsymbol{\theta}/\boldsymbol{P}}\left(\frac{1}{n}\bar{\boldsymbol{X}}^\top\bar{\boldsymbol{X}} + \lambda\boldsymbol{I}_d\right)^{-1}\boldsymbol{\Sigma}_{\boldsymbol{X}\boldsymbol{P}}\left(\frac{1}{n}\bar{\boldsymbol{X}}^\top\bar{\boldsymbol{X}} + \lambda\boldsymbol{I}_d\right)^{-1}\right)
$$

$$
\overset{(iii)}{=} \lim_{\lambda\to 0_+}\frac{\lambda^2}{d}\mathrm{tr}\left(\left(\frac{1}{n}\hat{\boldsymbol{X}}^\top\hat{\boldsymbol{X}} + \lambda\boldsymbol{\Sigma}_{\boldsymbol{\theta}/\boldsymbol{P}}^{-1}\right)^{-2}\boldsymbol{\Sigma}_{\boldsymbol{\theta}/\boldsymbol{P}}^{-1/2}\boldsymbol{\Sigma}_{\boldsymbol{X}\boldsymbol{P}}\boldsymbol{\Sigma}_{\boldsymbol{\theta}/\boldsymbol{P}}^{-1/2}\right),
$$

where we utilized (A3) and defined $\bar{\boldsymbol{X}} = \boldsymbol{X}\boldsymbol{P}^{1/2}$, $\boldsymbol{\Sigma}_{\boldsymbol{X}\boldsymbol{P}} = \boldsymbol{P}^{1/2}\boldsymbol{\Sigma}_{\boldsymbol{X}}\boldsymbol{P}^{1/2}$, $\boldsymbol{\Sigma}_{\boldsymbol{\theta}/\boldsymbol{P}} = \boldsymbol{P}^{-1/2}\boldsymbol{\Sigma}_{\boldsymbol{\theta}}\boldsymbol{P}^{-1/2}$ in (i), applied the equality $(\boldsymbol{A}\boldsymbol{A}^\top)^\dagger\boldsymbol{A} = \lim_{\lambda\to 0}(\boldsymbol{A}^\top\boldsymbol{A} + \lambda\boldsymbol{I})^{-1}\boldsymbol{A}$ in (ii), and defined $\hat{\boldsymbol{X}} = \boldsymbol{X}\boldsymbol{P}^{1/2}\boldsymbol{\Sigma}_{\boldsymbol{\theta}}^{-1/2}$ in (iii). To proceed, we first assume that $\boldsymbol{\Sigma}_{\boldsymbol{\theta}/\boldsymbol{P}}$ is invertible (i.e. $\lambda_{\min}(\boldsymbol{\Sigma}_{\boldsymbol{\theta}/\boldsymbol{P}})$ is bounded away from 0) and observe the following relation via a leave-one-out argument similar to that in Xu & Hsu (2019),

$$
\frac{1}{d}\mathrm{tr}\left(\frac{1}{n}\hat{\boldsymbol{X}}^\top\hat{\boldsymbol{X}}\left(\frac{1}{n}\hat{\boldsymbol{X}}^\top\hat{\boldsymbol{X}} + \lambda\boldsymbol{\Sigma}_{\boldsymbol{\theta}/\boldsymbol{P}}^{-1}\right)^{-2}\right) \tag{D.1}
$$

$$
\overset{(i)}{=} \frac{1}{d}\sum_{i=1}^n\frac{\frac{1}{n}\hat{\boldsymbol{x}}_i^\top\left(\frac{1}{n}\hat{\boldsymbol{X}}^\top\hat{\boldsymbol{X}} + \lambda\boldsymbol{\Sigma}_{\boldsymbol{\theta}/\boldsymbol{P}}^{-1}\right)_{\neg i}^{-2}\hat{\boldsymbol{x}}_i}{\left(1 + \frac{1}{n}\hat{\boldsymbol{x}}_i^\top\left(\frac{1}{n}\hat{\boldsymbol{X}}^\top\hat{\boldsymbol{X}} + \lambda\boldsymbol{\Sigma}_{\boldsymbol{\theta}/\boldsymbol{P}}^{-1}\right)_{\neg i}^{-1}\hat{\boldsymbol{x}}_i\right)^2}
$$

$$
\overset{(ii)}{\underset{p}{\to}} \frac{\frac{1}{d}\mathrm{tr}\left(\left(\frac{1}{n}\hat{\boldsymbol{X}}^\top\hat{\boldsymbol{X}} + \lambda\boldsymbol{\Sigma}_{\boldsymbol{\theta}/\boldsymbol{P}}^{-1}\right)^{-2}\boldsymbol{\Sigma}_{\boldsymbol{\theta}/\boldsymbol{P}}^{-1/2}\boldsymbol{\Sigma}_{\boldsymbol{X}\boldsymbol{P}}\boldsymbol{\Sigma}_{\boldsymbol{\theta}/\boldsymbol{P}}^{-1/2}\right)}{\left(1 + \frac{1}{n}\mathrm{tr}\left(\left(\frac{1}{n}\bar{\boldsymbol{X}}^\top\bar{\boldsymbol{X}} + \lambda\boldsymbol{I}_d\right)^{-1}\boldsymbol{\Sigma}_{\boldsymbol{X}\boldsymbol{P}}\right)\right)^2}, \tag{D.2}
$$

where (i) is due to the Woodbury identity and we defined $\left(\frac{1}{n}\hat{\boldsymbol{X}}^\top\hat{\boldsymbol{X}} + \lambda\boldsymbol{\Sigma}_{\boldsymbol{\theta}/\boldsymbol{P}}^{-1}\right)_{\neg i} = \frac{1}{n}\hat{\boldsymbol{X}}^\top\hat{\boldsymbol{X}} - \frac{1}{n}\hat{\boldsymbol{x}}_i\hat{\boldsymbol{x}}_i^\top + \lambda\boldsymbol{\Sigma}_{\boldsymbol{\theta}/\boldsymbol{P}}^{-1}$ which is independent to $\hat{\boldsymbol{x}}_i$ (see (Xu & Hsu, 2019, Eq. 58) for details), and in

(ii) we used (A3), the convergence to trace (Ledoit & Péché, 2011, Lemma 2.1) and its stability under low-rank perturbation (e.g., see (Ledoit & Péché, 2011, Eq. 18)) which we elaborate below. In particular, denote $\hat{\boldsymbol{\Sigma}} = \frac{1}{n}\hat{\boldsymbol{X}}^{\top}\hat{\boldsymbol{X}} + \lambda\boldsymbol{\Sigma}_{\boldsymbol{\theta}/\boldsymbol{P}}^{-1}$, for the denominator we have

$$
\sup_i\left|\frac{\lambda}{n}\mathrm{tr}\Big(\hat{\boldsymbol{\Sigma}}^{-1}\boldsymbol{\Sigma}_{\boldsymbol{\theta}/\boldsymbol{P}}^{-1/2}\boldsymbol{\Sigma}_{\boldsymbol{XP}}\boldsymbol{\Sigma}_{\boldsymbol{\theta}/\boldsymbol{P}}^{-1/2}\Big) - \frac{\lambda}{n}\mathrm{tr}\Big(\hat{\boldsymbol{\Sigma}}_{\neg i}^{-1}\boldsymbol{\Sigma}_{\boldsymbol{\theta}/\boldsymbol{P}}^{-1/2}\boldsymbol{\Sigma}_{\boldsymbol{XP}}\boldsymbol{\Sigma}_{\boldsymbol{\theta}/\boldsymbol{P}}^{-1/2}\Big)\right|
$$

$$
\leq\frac{\lambda}{n}\Big\|\boldsymbol{\Sigma}_{\boldsymbol{\theta}/\boldsymbol{P}}^{-1/2}\boldsymbol{\Sigma}_{\boldsymbol{XP}}\boldsymbol{\Sigma}_{\boldsymbol{\theta}/\boldsymbol{P}}^{-1/2}\Big\|_2\sup_i\Big|\mathrm{tr}\Big(\hat{\boldsymbol{\Sigma}}^{-1}\Big(\hat{\boldsymbol{\Sigma}} - \hat{\boldsymbol{\Sigma}}_{\neg i}\Big)\hat{\boldsymbol{\Sigma}}_{\neg i}^{-1}\Big)\Big|
$$

$$
\leq\frac{\lambda}{n}\Big\|\boldsymbol{\Sigma}_{\boldsymbol{\theta}/\boldsymbol{P}}^{-1/2}\boldsymbol{\Sigma}_{\boldsymbol{XP}}\boldsymbol{\Sigma}_{\boldsymbol{\theta}/\boldsymbol{P}}^{-1/2}\Big\|_2\Big\|\hat{\boldsymbol{\Sigma}}^{-1}\Big\|_2\sup_i\Big\|\hat{\boldsymbol{\Sigma}}_{\neg i}^{-1}\Big\|_2\mathrm{tr}\Big(\hat{\boldsymbol{\Sigma}} - \hat{\boldsymbol{\Sigma}}_{\neg i}\Big) \overset{(i)}{\to} O_p\Big(\frac{1}{n}\Big),
$$

where (i) is due to the definition of $\hat{\boldsymbol{\Sigma}}_{\neg i}$ and (A1)(A3). The result on the numerator can be obtained via a similar calculation, the details of which we omit.

We first note that the denominator can be evaluated by previous results (e.g. (Dobriban et al., 2018, Theorem 2.1)) as follows,

$$
\frac{1}{n}\mathrm{tr}\left(\left(\frac{1}{n}\bar{\boldsymbol{X}}^{\top}\bar{\boldsymbol{X}} + \lambda\boldsymbol{I}_d\right)^{-1}\boldsymbol{\Sigma}_{\boldsymbol{XP}}\right) \overset{a.s.}{\to} \frac{1}{\lambda m(-\lambda)} - 1. \tag{D.3}
$$

On the other hand, following the same derivation as Dobriban et al. (2018); Hastie et al. (2019), (D.1) can be decomposed as

$$
\frac{1}{d}\mathrm{tr}\left(\frac{1}{n}\hat{\boldsymbol{X}}^{\top}\hat{\boldsymbol{X}}\left(\frac{1}{n}\hat{\boldsymbol{X}}^{\top}\hat{\boldsymbol{X}} + \lambda\boldsymbol{\Sigma}_{\boldsymbol{\theta}/\boldsymbol{P}}^{-1}\right)^{-2}\right)
$$

$$
= \frac{1}{d}\mathrm{tr}\left(\left(\frac{1}{n}\bar{\boldsymbol{X}}^{\top}\bar{\boldsymbol{X}} + \lambda\boldsymbol{I}_d\right)^{-1}\boldsymbol{\Sigma}_{\boldsymbol{\theta}/\boldsymbol{P}}\right) - \frac{\lambda}{d}\mathrm{tr}\left(\left(\frac{1}{n}\bar{\boldsymbol{X}}^{\top}\bar{\boldsymbol{X}} + \lambda\boldsymbol{I}_d\right)^{-2}\boldsymbol{\Sigma}_{\boldsymbol{\theta}/\boldsymbol{P}}\right)
$$

$$
= \frac{1}{d}\mathrm{tr}\left(\left(\frac{1}{n}\bar{\boldsymbol{X}}^{\top}\bar{\boldsymbol{X}} + \lambda\boldsymbol{I}_d\right)^{-1}\boldsymbol{\Sigma}_{\boldsymbol{\theta}/\boldsymbol{P}}\right) + \frac{\lambda}{d}\frac{\mathrm{d}}{\mathrm{d}\lambda}\mathrm{tr}\left(\left(\frac{1}{n}\bar{\boldsymbol{X}}^{\top}\bar{\boldsymbol{X}} + \lambda\boldsymbol{I}_d\right)^{-1}\boldsymbol{\Sigma}_{\boldsymbol{\theta}/\boldsymbol{P}}\right). \tag{D.4}
$$

We employ (Rubio & Mestre, 2011, Theorem 1) to characterize (D.4). In particular, For any deterministic sequence of matrices $\boldsymbol{\Theta}_n \in \mathbb{R}^{d\times d}$ with finite trace norm, as $n, d \to \infty$ we have

$$
\mathrm{tr}\left(\boldsymbol{\Theta}_n\left(\frac{1}{n}\bar{\boldsymbol{X}}^{\top}\bar{\boldsymbol{X}} - z\boldsymbol{I}_d\right)^{-1} - \boldsymbol{\Theta}_n(c_n(z)\boldsymbol{\Sigma}_{\boldsymbol{XP}} - z\boldsymbol{I}_d)^{-1}\right) \overset{a.s.}{\to} 0,
$$

in which $c_n(z) \to -zm(z)$ for $z \in \mathbb{C}\backslash\mathbb{R}^+$ and $m(z)$ is defined in Theorem 1 due to the dominated convergence theorem. By (A3) we are allowed to take $\boldsymbol{\Theta}_n = \frac{1}{d}\boldsymbol{\Sigma}_{\boldsymbol{\theta}/\boldsymbol{P}}$. Thus we have

$$
\frac{\lambda}{d}\mathrm{tr}\left(\boldsymbol{\Sigma}_{\boldsymbol{\theta}/\boldsymbol{P}}\left(\frac{1}{n}\bar{\boldsymbol{X}}^{\top}\bar{\boldsymbol{X}} + \lambda\boldsymbol{I}_d\right)^{-1}\right) \to \frac{\lambda}{d}\mathrm{tr}\left(\boldsymbol{\Sigma}_{\boldsymbol{\theta}/\boldsymbol{P}}(\lambda m(-\lambda)\boldsymbol{\Sigma}_{\boldsymbol{XP}} + \lambda\boldsymbol{I}_d)^{-1}\right)
$$

$$
\overset{(i)}{=}\mathbb{E}\left[\frac{v_x v_\theta v_{xp}^{-1}}{1 + m(-\lambda)v_{xp}}\right], \quad \forall\lambda > -c_l, \tag{D.5}
$$

in which (i) is due to (A3), the fact that the LHS is almost surely bounded for $\lambda > -c_l$, where $c_l$ is the lowest non-zero eigenvalue of $\frac{1}{n}\bar{\boldsymbol{X}}^{\top}\bar{\boldsymbol{X}}$, and the application of the dominated convergence theorem. Differentiating (D.5) (note that the derivative is also bounded on $\lambda > -c_l$) yields

$$
\frac{\lambda}{d}\frac{\mathrm{d}}{\mathrm{d}\lambda}\mathrm{tr}\left(\left(\frac{1}{n}\bar{\boldsymbol{X}}^{\top}\bar{\boldsymbol{X}} + \lambda\boldsymbol{I}_d\right)^{-1}\boldsymbol{\Sigma}_{\boldsymbol{\theta}/\boldsymbol{P}}\right) \to \mathbb{E}\left[\frac{v_x v_\theta v_{xp}^{-1}}{\lambda(1 + m(-\lambda)v_{xp})} - \frac{m'(-\lambda)v_x v_\theta}{(1 + m(-\lambda)v_{xp})^2}\right] \tag{D.6}
$$

Note that the numerator of (D.2) is the quantity of interest. Combining (D.1) (D.2) (D.3) (D.4) (D.5) (D.6) and taking $\lambda \to 0$ yields the formula of the bias term. Finally, when $\boldsymbol{\Sigma}_{\boldsymbol{\theta}/\boldsymbol{P}}$ is not invertible, observe that if we increment all eigenvalues by some small $\epsilon > 0$ to ensure invertibility

$\Sigma_{\theta/P}^{\epsilon} = \Sigma_{\theta/P} + \epsilon I_d$, (3.3) is bounded and also decreasing w.r.t. $\epsilon$. Thus by the dominated convergence theorem we take $\epsilon \to 0$ and obtain the desired result. We remark that similar (but less general) characterization can also be derived based on (Ledoit & Péché, 2011, Theorem 1.2) when the eigenvalues of $\Sigma_{XP}$ and $\Sigma_{\theta/P}$ exhibit certain relations.

To show that $P = U \operatorname{diag}(e_\theta) U^\top$ achieves the lowest bias, first note that under the definition of random variables in (A3), our claimed optimal preconditioner is equivalent to $v_{xp} \stackrel{a.s.}{=} v_x v_\theta$. We therefore define an interpolation $v_\alpha = \alpha v_x v_\theta + (1-\alpha)\bar{v}$ for some $\bar{v}$ and write the corresponding Stieltjes transform as $m_\alpha(-\lambda)$ and the bias term as $B_\alpha$. We aim to show that $\operatorname{argmin}_{\alpha \in [0,1]} B_\alpha = 1$.

For notational convenience define $g_\alpha \triangleq m_\alpha(0) v_x v_\theta$ and $h_\alpha \triangleq m_\alpha(0) v_\alpha$. One can check that

$$B_\alpha = \mathbb{E}\left[\frac{v_x v_\theta}{(1+h_\alpha)^2}\right]\mathbb{E}\left[\frac{h_\alpha}{(1+h_\alpha)^2}\right]^{-1}; \quad \frac{dm_\alpha(-\lambda)}{d\alpha}\bigg|_{\lambda \to 0} = \frac{m_\alpha(0)\mathbb{E}\left[\frac{h_\alpha - g_\alpha}{(1+h_\alpha)^2}\right]}{(1-\alpha)\mathbb{E}\left[\frac{h_\alpha}{(1+h_\alpha)^2}\right]}.$$

We now verify that the derivative of $B_\alpha$ w.r.t. $\alpha$ is non-positive for $\alpha \in [0,1]$. A standard simplification of the derivative yields

$$\frac{dB_\alpha}{d\alpha} \propto -2\mathbb{E}\left[\frac{(g_\alpha - h_\alpha)^2}{(1+h_\alpha)^3}\right]\left(\mathbb{E}\left[\frac{h_\alpha}{(1+h_\alpha)^2}\right]\right)^2 - 2\left(\mathbb{E}\left[\frac{g_\alpha - h_\alpha}{(1+h_\alpha)^2}\right]\right)^2 \mathbb{E}\left[\frac{h_\alpha^2}{(1+h_\alpha)^3}\right]$$

$$+ 4\mathbb{E}\left[\frac{h_\alpha(g_\alpha - h_\alpha)}{(1+h_\alpha)^3}\right]\mathbb{E}\left[\frac{g_\alpha - h_\alpha}{(1+h_\alpha)^2}\right]\mathbb{E}\left[\frac{h_\alpha}{(1+h_\alpha)^2}\right]$$

$$\stackrel{(i)}{\leq} -4\sqrt{\mathbb{E}\left[\frac{(g_\alpha - h_\alpha)^2}{(1+h_\alpha)^3}\right]\mathbb{E}\left[\frac{h_\alpha^2}{(1+h_\alpha)^3}\right]}\left(\mathbb{E}\left[\frac{g_\alpha - h_\alpha}{(1+h_\alpha)^2}\right]\right)^2\left(\mathbb{E}\left[\frac{h_\alpha}{(1+h_\alpha)^2}\right]\right)^2$$

$$+ 4\mathbb{E}\left[\frac{h_\alpha(g_\alpha - h_\alpha)}{(1+h_\alpha)^3}\right]\mathbb{E}\left[\frac{g_\alpha - h_\alpha}{(1+h_\alpha)^2}\right]\mathbb{E}\left[\frac{h_\alpha}{(1+h_\alpha)^2}\right] \stackrel{(ii)}{\leq} 0,$$

where (i) is due to AM-GM and (ii) due to Cauchy-Schwarz on the first term. Note that the two equalities hold when $g_\alpha = h_\alpha$, from which one can easily deduce that the optimum is achieved when $v_{xp} \stackrel{a.s.}{=} v_x v_\theta$, and thus we know that the proposed $P$ is the optimal preconditioner that is codiagonazable with $\Sigma_X$. $\qquad\square$

### D.5 PROOF OF PROPOSITION 4

**Proof.** Since $\Sigma_\theta = \Sigma_X^r$, we can simplify the expressions by defining $v_x \triangleq h$ and thus $v_\theta = h^r$. From Theorem 3 we have the following derivation of the GD bias under (A1)(A3),

$$B(\hat{\theta}_I) \to \frac{m_1'}{m_1^2}\mathbb{E}\frac{h \cdot h^r}{(1+h \cdot m_1)^2} = \frac{\mathbb{E}\frac{h^{1+r}}{(1+h \cdot m_1)^2}}{1 - \gamma\mathbb{E}\frac{(h \cdot m_1)^2}{(1+h \cdot m_1)^2}}, \tag{D.7}$$

where $m_1 = \lim_{\lambda \to 0_+} m(-\lambda)$, and $m$ satisfies

$$\frac{1}{m(-\lambda)} = \lambda + \gamma\mathbb{E}\left[\frac{h}{1+h \cdot m(-\lambda)}\right].$$

Similarly, for NGD ($P = \Sigma_X^{-1}$) we have

$$B(\hat{\theta}_{F^{-1}}) \to \frac{m_2'}{m_2^2}\mathbb{E}\frac{h \cdot h^r}{(1+m_2)^2} = \frac{\mathbb{E}\frac{h^{1+r}}{(1+m_2)^2}}{1 - \gamma\mathbb{E}\frac{m_2^2}{(1+m_2)^2}} = \frac{\mathbb{E}h^{1+r}}{(1+m_2)^2 - \gamma m_2^2}, \tag{D.8}$$

where standard calculation yields $m_2 = (\gamma - 1)^{-1}$, and thus $B(\hat{\theta}_{F^{-1}}) \to (1 - \gamma^{-1})\mathbb{E}h^{1+r}$.

To compare the magnitude of (D.7) and (D.8), observe the following equivalence.

$$B(\hat{\theta}_I) \lessgtr B(\hat{\theta}_{F^{-1}})$$

$$\Leftrightarrow \; \mathbb{E}\frac{h^{1+r}}{(1+h\cdot m_1)^2}\cdot\frac{\gamma}{\gamma-1} \lessgtr \left(1-\gamma\mathbb{E}\frac{(h\cdot m_1)^2}{(1+h\cdot m_1)^2}\right)\mathbb{E}h^{1+r}.$$

$$\overset{(i)}{\Leftrightarrow} \; \mathbb{E}\frac{\zeta^{1+r}}{(1+\zeta)^2}\mathbb{E}\frac{\zeta}{1+\zeta} \lessgtr \mathbb{E}\frac{\zeta}{(1+\zeta)^2}\mathbb{E}\zeta^{1+r}\mathbb{E}\frac{1}{1+\zeta}. \tag{D.9}$$

where (i) follows from the definition of $m_1$ and we defined $\zeta \triangleq h\cdot m_1$. Note that when $r \le -1$ and $h$ is not a point mass, we have

$$\mathbb{E}\frac{\zeta^{1+r}}{(1+\zeta)^2}\mathbb{E}\frac{\zeta}{1+\zeta} > \mathbb{E}\frac{\zeta}{(1+\zeta)^2}\mathbb{E}\frac{\zeta^{1+r}}{1+\zeta} > \mathbb{E}\frac{\zeta}{(1+\zeta)^2}\mathbb{E}\zeta^{1+r}\mathbb{E}\frac{1}{1+\zeta}.$$

On the other hand, when $r \ge 0$, following the exact same procedure we get

$$\mathbb{E}\frac{\zeta^{1+r}}{(1+\zeta)^2}\mathbb{E}\frac{\zeta}{1+\zeta} < \mathbb{E}\frac{\zeta}{(1+\zeta)^2}\mathbb{E}\zeta^{1+r}\mathbb{E}\frac{1}{1+\zeta}.$$

Combining the two cases completes the proof.

$\square$

## D.6 PROOF OF PROPOSITION 5

**Proof.** We first outline a more general setup where $\boldsymbol{P}_\alpha = f(\boldsymbol{\Sigma}_{\boldsymbol{X}};\alpha)$ for continuous and differentiable function of $\alpha$ and $f$ applied to eigenvalues of $\boldsymbol{\Sigma}_{\boldsymbol{x}}$. For any interval $\mathcal{I} \subseteq [0,1]$, we claim

(a) Suppose all four functions $\frac{1}{xf(x;\alpha)}$, $f(x;\alpha)$, $\frac{\partial f(x;\alpha)}{\partial\alpha}/f(x;\alpha)$ and $x\frac{\partial f(x;\alpha)}{\partial\alpha}$ are decreasing functions of $x$ on the support of $v_x$ for all $\alpha \in \mathcal{I}$. In addition, $\frac{\partial f(x;\alpha)}{\partial\alpha} \ge 0$ on the support of $v_x$ for all $\alpha \in \mathcal{I}$. Then the stationary bias is an increasing function of $\alpha$ on $\mathcal{I}$.

(b) For all $\alpha \in \mathcal{I}$, suppose $xf(x;\alpha)$ is a monotonic function of $x$ on the support of $v_x$ and $\frac{\partial f(x;\alpha)}{\partial\alpha}/f(x;\alpha)$ is a decreasing function of $x$ on the support of $v_x$. Then the stationary variance is a decreasing function of $\alpha$ on $\mathcal{I}$.

Let us verify the three choices of $\boldsymbol{P}_\alpha$ in Proposition 5 one by one.

- When $\boldsymbol{P}_\alpha = (1-\alpha)\boldsymbol{I}_d + \alpha(\boldsymbol{\Sigma}_{\boldsymbol{X}})^{-1}$, the corresponding $f(x;\alpha)$ is $(1-\alpha)+\alpha x$. This satisfies all conditions in (a) and (b) for all $\alpha \in [0,1]$. Hence, the stationary variance is a decreasing function and the stationary bias is an increasing function of $\alpha \in [0,1]$.

- When $\boldsymbol{P}_\alpha = (\boldsymbol{\Sigma}_{\boldsymbol{X}})^{-\alpha}$, the corresponding $f(x;\alpha)$ is $x^{-\alpha}$. It is clear that it satisfies all conditions in (a) and (b) for all $\alpha \in [0,1]$ except for the condition that $x\frac{\partial f(x;\alpha)}{\partial\alpha} = -x^{1-\alpha}\ln x$ is a decreasing function of $x$. Note that $x\frac{\partial f(x;\alpha)}{\partial\alpha} = -x^{1-\alpha}\ln x$ is a decreasing function of $x$ on the support of $v_x$ only for $\alpha \ge \frac{\ln(\kappa)-1}{\ln(\kappa)}$ where $\kappa = \sup v_x/\inf v_x$. Hence, the stationary variance is a decreasing function of $\alpha \in [0,1]$ and the stationary bias is an increasing function of $\alpha \in [\max(0,\frac{\ln(\kappa)-1}{\ln(\kappa)}),1]$.

- When $\boldsymbol{P}_\alpha = (\alpha\boldsymbol{\Sigma}_{\boldsymbol{X}}+(1-\alpha)\boldsymbol{I}_d)^{-1}$, the corresponding $f(x;\alpha)$ is $1/(\alpha x + (1-\alpha))$, which satisfies all conditions in (a) and (b) for all $\alpha \in [0,1]$ except for the condition that $x\frac{\partial f(x;\alpha)}{\partial\alpha} = \frac{x(1-x)}{(\alpha x+(1-\alpha))^2}$ is a decreasing function of $x$. Note that $x\frac{\partial f(x;\alpha)}{\partial\alpha} = \frac{x(1-x)}{(\alpha x+(1-\alpha))^2}$ is a decreasing function of $x$ on the support of $v_x$ only for $\alpha \ge \frac{\kappa-2}{\kappa-1}$. Hence, the stationary variance is a decreasing function of $\alpha \in [0,1]$ and the stationary bias is an increasing function of $\alpha \in [\max(0,\frac{\kappa-2}{\kappa-1}),1]$.

To show (a) and (b), note that under the conditions on $\boldsymbol{\Sigma}_{\boldsymbol{x}}$ and $\boldsymbol{\Sigma}_{\boldsymbol{\theta}}$ assumed in Proposition 5, the stationary bias $B(\hat{\boldsymbol{\theta}}_{\boldsymbol{P}_\alpha})$ and the stationary variance $V(\hat{\boldsymbol{\theta}}_{\boldsymbol{P}_\alpha})$ can be simplified to

$$B(\hat{\boldsymbol{\theta}}_{\boldsymbol{P}_\alpha}) = \frac{m'_\alpha(0)}{m_\alpha^2(0)}\mathbb{E}\frac{v_x}{(1+v_xf(v_x;\alpha)m_\alpha(0))^2} \quad\text{and}\quad V(\hat{\boldsymbol{\theta}}_{\boldsymbol{P}_\alpha}) = \sigma^2\cdot\left(\frac{m'_\alpha(0)}{m_\alpha^2(0)}-1\right),$$

where $m_\alpha(z)$ and $m'_\alpha(z)$ satisfy

$$1 = -zm_\alpha(z) + \gamma\mathbb{E}\frac{v_xf(v_x;\alpha)m_\alpha(z)}{1+v_xf(v_x;\alpha)m_\alpha(z)} \tag{D.10}$$

$$\frac{m_\alpha'(z)}{m_\alpha^2(z)} = \frac{1}{1 - \gamma\mathbb{E}\left(\frac{f(v_x;\alpha)m_\alpha(z)}{1+f(v_x;\alpha)m_\alpha(z)}\right)^2}. \tag{D.11}$$

For notation convenience, let $f_\alpha := v_x f(v_x; \alpha)$. From (D.11), we have the following equivalences.

$$B(\hat{\boldsymbol{\theta}}_{\boldsymbol{P}_\alpha}) = \frac{\mathbb{E}\frac{v_x}{(1+f_\alpha m_\alpha(0))^2}}{1 - \gamma\mathbb{E}\left(\frac{f_\alpha m_\alpha(0)}{1+f_\alpha m_\alpha(0)}\right)^2}, \tag{D.12}$$

$$V(\hat{\boldsymbol{\theta}}_{\boldsymbol{P}_\alpha}) = \sigma^2\left(\frac{1}{1 - \gamma\mathbb{E}\left(\frac{f_\alpha m_\alpha(0)}{1+f_\alpha m_\alpha(0)}\right)^2} - 1\right). \tag{D.13}$$

We first show that (b) holds. Note that from (D.13), we have

$$\frac{\partial V(\hat{\boldsymbol{\theta}}_{\boldsymbol{P}_\alpha})}{\partial\alpha} = \gamma\sigma^2\left(\frac{1}{1-\gamma\mathbb{E}\left(\frac{f_\alpha m_\alpha(0)}{1+f_\alpha m_\alpha(0)}\right)^2}\right)^2\mathbb{E}\left[\frac{2f_\alpha m_\alpha(0)}{(1+f_\alpha m_\alpha(0))^3}\left(f_\alpha\frac{\partial m_\alpha(z)}{\partial\alpha}\Big|_{z=0} + \frac{\partial f_\alpha}{\partial\alpha}m_\alpha(0)\right)\right]. \tag{D.14}$$

To calculate $\frac{\partial m_\alpha(z)}{\partial\alpha}\Big|_{z=0}$, we take derivatives with respect to $\alpha$ on both sides of (D.10),

$$0 = \gamma\mathbb{E}\left[\frac{1}{(1+f_\alpha m_\alpha(0))^2}\cdot\left(f_\alpha\frac{\partial m_\alpha(z)}{\partial\alpha}\Big|_{z=0} + \frac{\partial f_\alpha}{\partial\alpha}m_\alpha(0)\right)\right]. \tag{D.15}$$

Therefore, plugging (D.15) into (D.14) yields

$$\frac{\partial V(\hat{\boldsymbol{\theta}}_{\boldsymbol{P}_\alpha})}{\partial\alpha} = 2\gamma\sigma^2\left(\frac{m_\alpha(0)}{1-\gamma\mathbb{E}\left(\frac{f_\alpha m_\alpha(0)}{1+f_\alpha m_\alpha(0)}\right)^2}\right)^2\left(\mathbb{E}\frac{f_\alpha}{(1+f_\alpha m_\alpha(0))^2}\right)^{-1}$$

$$\times\left(\mathbb{E}\frac{f_\alpha\frac{\partial f_\alpha}{\partial\alpha}}{(1+f_\alpha m_\alpha(0))^3}\mathbb{E}\frac{f_\alpha}{(1+f_\alpha m_\alpha(0))^2} - \mathbb{E}\frac{f_\alpha^2}{(1+f_\alpha m_\alpha(0))^3}\mathbb{E}\frac{\frac{\partial f_\alpha}{\partial\alpha}}{(1+f_\alpha m_\alpha(0))^2}\right)$$

Thus showing $V(\hat{\boldsymbol{\theta}}_{\boldsymbol{P}_\alpha})$ is a decreasing function of $\alpha$ is equivalent to showing that

$$\mathbb{E}\frac{f_\alpha^2}{(1+f_\alpha m_\alpha(0))^3}\mathbb{E}\frac{\frac{\partial f_\alpha}{\partial\alpha}}{(1+f_\alpha m_\alpha(0))^2} \geq \mathbb{E}\frac{f_\alpha\frac{\partial f_\alpha}{\partial\alpha}}{(1+f_\alpha m_\alpha(0))^3}\mathbb{E}\frac{f_\alpha}{(1+f_\alpha m_\alpha(0))^2}. \tag{D.16}$$

Let $\mu_x$ be the probability measure of $v_x$. We define a new measure $\tilde{\mu}_x = \frac{f_\alpha\mu_x}{(1+f_\alpha m_\alpha(0))^2}$, and let $\tilde{v}_x$ follow the new measure. Since $\frac{\partial f(x;\alpha)}{\partial\alpha}/f(x;\alpha)$ is a decreasing function of $x$ and $xf(x;\alpha)$ is a monotonic function of $x$,

$$\mathbb{E}\frac{\tilde{v}_x f(\tilde{v}_x;\alpha)}{1+\tilde{v}_x f(\tilde{v}_x;\alpha)m_\alpha(0)}\mathbb{E}\frac{\frac{\partial\tilde{v}_x f(\tilde{v}_x;\alpha)}{\partial\alpha}}{\tilde{v}_x f(\tilde{v}_x;\alpha)} \geq \mathbb{E}\frac{\frac{\partial\tilde{v}_x f(\tilde{v}_x;\alpha)}{\partial\alpha}}{1+\tilde{v}_x f(\tilde{v}_x;\alpha)m_\alpha(0)}.$$

Changing $\tilde{v}_x$ back to $v_x$, we arrive at (D.16) and thus (b).

For the bias term $B(\hat{\boldsymbol{\theta}}_{\boldsymbol{P}_\alpha})$, note that from (D.10) and (D.12), we have

$$\frac{\partial B(\hat{\boldsymbol{\theta}}_{\boldsymbol{P}_\alpha})}{\partial\alpha} = \frac{1}{\gamma}\left(\frac{1}{\gamma} - \mathbb{E}\left(\frac{f_\alpha m_\alpha(0)}{1+f_\alpha m_\alpha(0)}\right)^2\right)^{-2}$$

$$\times\left(-\mathbb{E}\left[2\frac{v_x}{(1+f_\alpha m_\alpha(0))^3}\cdot\left(f_\alpha\frac{\partial m_\alpha(z)}{\partial\alpha}\Big|_{z=0} + \frac{\partial f_\alpha}{\partial\alpha}m_\alpha(0)\right)\right]\mathbb{E}\frac{f_\alpha m_\alpha(0)}{(1+f_\alpha m_\alpha(0))^2}\right.$$

$$\left. + \mathbb{E}\frac{v_x}{(1+f_\alpha m_\alpha(0))^2}\mathbb{E}\left[2\frac{f_\alpha m_\alpha(0)}{(1+f_\alpha m_\alpha(0))^3}\cdot\left(f_\alpha\frac{\partial m_\alpha(z)}{\partial\alpha}\Big|_{z=0} + \frac{\partial f_\alpha}{\partial\alpha}m_\alpha(0)\right)\right]\right). \tag{D.17}$$

Similarly, we combine (D.15) and (D.17) and simplify the expression. To verify $B(\hat{\boldsymbol{\theta}}_{\boldsymbol{P}_\alpha})$ is an increasing function of $\alpha$, we need to show that

$$
\begin{aligned}
0 \leq & \left( \mathbb{E}\frac{v_x f_\alpha m_\alpha(0)}{(1+f_\alpha m_\alpha(0))^3} \mathbb{E}\frac{\frac{\partial f_\alpha}{\partial\alpha}}{(1+f_\alpha m_\alpha(0))^2} - \mathbb{E}\frac{v_x\frac{\partial f_\alpha}{\partial\alpha}}{(1+f_\alpha m_\alpha(0))^3} \mathbb{E}\frac{f_\alpha m_\alpha(0)}{(1+f_\alpha m_\alpha(0))^2} \right) \mathbb{E}\frac{f_\alpha m_\alpha(0)}{(1+f_\alpha m_\alpha(0))^2} \\
& - \mathbb{E}\frac{v_x}{(1+f_\alpha m_\alpha(0))^2} \left( \mathbb{E}\frac{(f_\alpha m_\alpha(0))^2}{(1+f_\alpha m_\alpha(0))^3} \mathbb{E}\frac{\frac{\partial f_\alpha}{\partial\alpha}}{(1+f_\alpha m_\alpha(0))^2} - \mathbb{E}\frac{f_\alpha m_\alpha(0)\frac{\partial f_\alpha}{\partial\alpha}}{(1+f_\alpha m_\alpha(0))^3} \mathbb{E}\frac{f_\alpha m_\alpha(0)}{(1+f_\alpha m_\alpha(0))^2} \right),
\end{aligned}
\tag{D.18}
$$

Let $h_\alpha \triangleq f_\alpha m_\alpha(0) = v_x f(v_x;\alpha)m_\alpha(0)$ and $g_\alpha \triangleq \frac{\partial f_\alpha}{\partial\alpha} = v_x \frac{\partial f(v_x;\alpha)}{\partial\alpha}$. Then (D.18) can be further simplified to the following equation

$$
\begin{aligned}
0 \leq & \underbrace{\mathbb{E}\frac{v_x h_\alpha}{(1+h_\alpha)^3}\mathbb{E}\frac{g_\alpha}{(1+h_\alpha)^3}\mathbb{E}\frac{h_\alpha}{(1+h_\alpha)^3} - \mathbb{E}\frac{v_x}{(1+h_\alpha)^3}\mathbb{E}\frac{g_\alpha}{(1+h_\alpha)^3}\mathbb{E}\frac{h_\alpha^2}{(1+h_\alpha)^3}}_{\text{part 1}} \\
& + \underbrace{\mathbb{E}\frac{v_x}{(1+h_\alpha)^3}\mathbb{E}\frac{g_\alpha h_\alpha}{(1+h_\alpha)^3}\mathbb{E}\frac{h_\alpha}{(1+h_\alpha)^3} - \mathbb{E}\frac{v_x g_\alpha}{(1+h_\alpha)^3}\mathbb{E}\frac{h_\alpha}{(1+h_\alpha)^3}\mathbb{E}\frac{h_\alpha}{(1+h_\alpha)^3}}_{\text{part 2}} \\
& + \underbrace{2\mathbb{E}\frac{v_x h_\alpha}{(1+h_\alpha)^3}\mathbb{E}\frac{g_\alpha h_\alpha}{(1+h_\alpha)^3}\mathbb{E}\frac{h_\alpha}{(1+h_\alpha)^3} - 2\mathbb{E}\frac{v_x g_\alpha}{(1+h_\alpha)^3}\mathbb{E}\frac{h_\alpha^2}{(1+h_\alpha)^3}\mathbb{E}\frac{h_\alpha}{(1+h_\alpha)^3}}_{\text{part 3}} \\
& + \underbrace{\mathbb{E}\frac{v_x h_\alpha}{(1+h_\alpha)^3}\mathbb{E}\frac{g_\alpha h_\alpha}{(1+h_\alpha)^3}\mathbb{E}\frac{h_\alpha^2}{(1+h_\alpha)^3} - \mathbb{E}\frac{v_x g_\alpha}{(1+h_\alpha)^3}\mathbb{E}\frac{h_\alpha^2}{(1+h_\alpha)^3}\mathbb{E}\frac{h_\alpha^2}{(1+h_\alpha)^3}}_{\text{part 4}} \cdot
\end{aligned}
\tag{D.19}
$$

Note that under condition of (a), we know that both $h_\alpha$ and $v_x/h_\alpha$ are increasing functions of $v_x$; and both $g_\alpha/h_\alpha$ and $g_\alpha$ are decreasing functions of $v_x$. Hence, with calculation similar to (D.16), we know part 1,2,3,4 in (D.19) are all non-negative, and therefore (D.19) holds. $\square$

**Remark.** *The above characterization provides sufficient but not necessary conditions for the monotonicity of the bias term. In general, the expression of the bias is rather opaque, and determining the sign of its derivative can be tedious, except for certain special cases (e.g., $\gamma = 2$ and the eigenvalues of $\boldsymbol{\Sigma}_{\boldsymbol{X}}$ are two equally weighted point masses). We conjecture that the bias is monotone for $\alpha \in [0, 1]$ for a much wider class of $\boldsymbol{\Sigma}_{\boldsymbol{X}}$, as shown in Figure 22.*

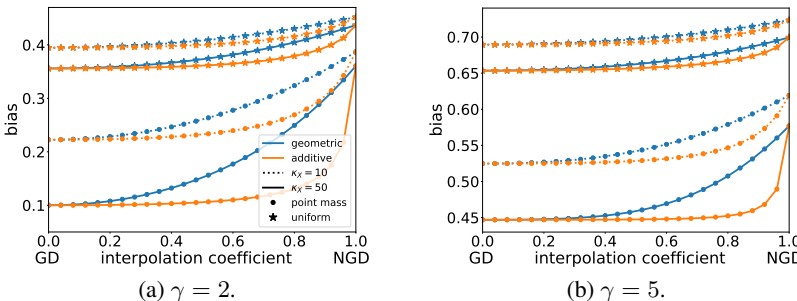

Figure 22: Illustration of the monotonicity of the bias term under $\boldsymbol{\Sigma}_{\boldsymbol{\theta}} = \boldsymbol{I}_d$. We consider two distributions of eigenvalues for $\boldsymbol{\Sigma}_{\boldsymbol{X}}$: two equally weighted point masses (circle) and a uniform distribution (star), and vary the condition number $\kappa_X$ and overparameterization level $\gamma$. In all cases the bias in monotone in $\alpha \in [0, 1]$.

## D.7 PROOF OF PROPOSITION 6

**Proof.** Taking the derivative of $V(\boldsymbol{\theta}_{\boldsymbol{P}}(t))$ w.r.t. time yields (omitting the scalar $\sigma^2$),

$$
\frac{\mathrm{d}V(\boldsymbol{\theta}_{\boldsymbol{P}}(t))}{\mathrm{d}t} = \frac{\mathrm{d}}{\mathrm{d}t}\left\| \boldsymbol{\Sigma}_{\boldsymbol{X}}^{1/2}\boldsymbol{P}\boldsymbol{X}^\top\left(\boldsymbol{I}_n - \exp\left(-\frac{t}{n}\boldsymbol{X}\boldsymbol{P}\boldsymbol{X}^\top\right)\right)\left(\boldsymbol{X}\boldsymbol{P}\boldsymbol{X}^\top\right)^{-1} \right\|_F^2
$$

$$\overset{(i)}{=}\frac{1}{n}\mathrm{tr}\left(\mathbf{\Sigma}_{\boldsymbol{XP}}\,\underbrace{\bar{\boldsymbol{X}}^{\top}\boldsymbol{S}_{\boldsymbol{P}}\exp\left(-\frac{t}{n}\boldsymbol{S}_{\boldsymbol{P}}\right)\boldsymbol{S}_{\boldsymbol{P}}^{-2}\left(\boldsymbol{I}_n-\exp\left(-\frac{t}{n}\boldsymbol{S}_{\boldsymbol{P}}\right)\right)\bar{\boldsymbol{X}}}_{p.s.d.}\right)\overset{(ii)}{>}0,$$

where we defined $\bar{\boldsymbol{X}}=\boldsymbol{XP}^{1/2}$ and $\boldsymbol{S}_{\boldsymbol{P}}=\boldsymbol{XPX}^{\top}$ in (i), and (ii) is due to (A2-3) the inequality $\mathrm{tr}(\boldsymbol{AB})\geq\lambda_{\min}(\boldsymbol{A})\mathrm{tr}(\boldsymbol{B})$ for positive semi-definite $\boldsymbol{A}$ and $\boldsymbol{B}$. $\qquad\square$

### D.8 PROOF OF PROPOSITION 7

**Proof.** Recall the definition of the bias (well-specified) of $\hat{\boldsymbol{\theta}}_{\boldsymbol{P}}(t)$,

$$B(\boldsymbol{\theta}_{\boldsymbol{P}}(t))\overset{(i)}{=}\frac{1}{d}\mathrm{tr}\left(\mathbf{\Sigma}_{\boldsymbol{\theta}}\left(\boldsymbol{I}_d-\boldsymbol{PX}^{\top}\boldsymbol{W}_{\boldsymbol{P}}(t)\boldsymbol{S}_{\boldsymbol{P}}^{-1}\boldsymbol{X}\right)^{\top}\mathbf{\Sigma}_{\boldsymbol{X}}\left(\boldsymbol{I}_d-\boldsymbol{PX}^{\top}\boldsymbol{W}_{\boldsymbol{P}}(t)\boldsymbol{S}_{\boldsymbol{P}}^{-1}\boldsymbol{X}\right)\right)$$

$$\overset{(ii)}{=}\frac{1}{d}\mathrm{tr}\left(\mathbf{\Sigma}_{\boldsymbol{\theta}/\boldsymbol{P}}\left(\boldsymbol{I}_d-\bar{\boldsymbol{X}}^{\top}\boldsymbol{W}_{\boldsymbol{P}}(t)\boldsymbol{S}_{\boldsymbol{P}}^{-1}\bar{\boldsymbol{X}}\right)^{\top}\mathbf{\Sigma}_{\boldsymbol{XP}}\left(\boldsymbol{I}_d-\bar{\boldsymbol{X}}^{\top}\boldsymbol{W}_{\boldsymbol{P}}(t)\boldsymbol{S}_{\boldsymbol{P}}^{-1}\bar{\boldsymbol{X}}\right)\right)$$

$$\overset{(iii)}{\geq}\frac{1}{d}\mathrm{tr}\left(\left(\mathbf{\Sigma}_{\boldsymbol{XP}}^{1/2}\left(\boldsymbol{I}_d-\bar{\boldsymbol{X}}^{\top}\boldsymbol{W}_{\boldsymbol{P}}(t)\boldsymbol{S}_{\boldsymbol{P}}^{-1}\bar{\boldsymbol{X}}\right)\mathbf{\Sigma}_{\boldsymbol{\theta}/\boldsymbol{P}}^{1/2}\right)^{2}\right),\tag{D.20}$$

where we defined $\boldsymbol{S}_{\boldsymbol{P}}=\boldsymbol{XPX}^{\top}$, $\boldsymbol{W}_{\boldsymbol{P}}(t)=\boldsymbol{I}_n-\exp(-\frac{t}{n}\boldsymbol{S}_{\boldsymbol{P}})$ in (i), $\bar{\boldsymbol{X}}=\boldsymbol{XP}^{1/2}$ in (ii), and (iii) is due to the inequality $\mathrm{tr}\left(\boldsymbol{A}^{\top}\boldsymbol{A}\right)\geq\mathrm{tr}\left(\boldsymbol{A}^{2}\right)$.

When $\mathbf{\Sigma}_{\boldsymbol{X}}=\mathbf{\Sigma}_{\boldsymbol{\theta}}^{-1}$, i.e. NGD achieves lowest stationary bias, (D.20) simplifies to

$$B(\boldsymbol{\theta}_{\boldsymbol{P}}(t))\geq\frac{1}{d}\mathrm{tr}\left(\left(\boldsymbol{I}_d-\bar{\boldsymbol{X}}^{\top}\boldsymbol{W}_{\boldsymbol{P}}(t)\boldsymbol{S}_{\boldsymbol{P}}^{-1}\bar{\boldsymbol{X}}\right)^{2}\right)=\left(1-\frac{1}{\gamma}\right)+\frac{1}{d}\sum_{i=1}^{n}\exp\left(-\frac{t}{n}\bar{\lambda}_i\right)^{2},\tag{D.21}$$

where $\bar{\lambda}$ is the eigenvalue of $\boldsymbol{S}_{\boldsymbol{P}}$. On the other hand, since $\boldsymbol{F}=\mathbf{\Sigma}_{\boldsymbol{X}}$, for the NGD iterate $\hat{\boldsymbol{\theta}}_{\boldsymbol{F}^{-1}}(t)$,

$$B(\boldsymbol{\theta}_{\boldsymbol{F}^{-1}}(t))=\frac{1}{d}\mathrm{tr}\left(\left(\boldsymbol{I}_d-\hat{\boldsymbol{X}}^{\top}\boldsymbol{W}_{\boldsymbol{F}^{-1}}(t)\boldsymbol{S}_{\boldsymbol{F}^{-1}}^{-1}\hat{\boldsymbol{X}}\right)^{2}\right)=\left(1-\frac{1}{\gamma}\right)+\frac{1}{d}\sum_{i=1}^{n}\exp\left(-\frac{t}{n}\hat{\lambda}_i\right)^{2}\tag{D.22}$$

where $\hat{\boldsymbol{X}}=\boldsymbol{X}\mathbf{\Sigma}_{\boldsymbol{X}}^{-1/2}$ and $\bar{\lambda}$ is the eigenvalue of $\boldsymbol{S}_{\boldsymbol{F}^{-1}}=\hat{\boldsymbol{X}}\hat{\boldsymbol{X}}^{\top}$. Comparing (D.21)(D.22), we see that given $\hat{\boldsymbol{\theta}}_{\boldsymbol{P}}(t)$ at a fixed t, if we run NGD for time $T>\frac{\bar{\lambda}_{\max}}{\hat{\lambda}_{\min}}t$ (note that $T/t=O(1)$ by (A2-3)), then we have $B(\boldsymbol{\theta}_{\boldsymbol{P}}(t))\geq B(\boldsymbol{\theta}_{\boldsymbol{F}^{-1}}(T))$ for any $\boldsymbol{P}$ satisfying (A3). This thus implies that $B^{\mathrm{opt}}(\boldsymbol{\theta}_{\boldsymbol{P}})\geq B^{\mathrm{opt}}(\boldsymbol{\theta}_{\boldsymbol{F}^{-1}})$.

On the other hand, when $\mathbf{\Sigma}_{\boldsymbol{\theta}}=\boldsymbol{I}_d$, we can show that the bias term of GD is monotonically decreasing through time by taking its derivative,

$$\frac{\mathrm{d}}{\mathrm{d}t}B(\boldsymbol{\theta}_{\boldsymbol{I}}(t))=\frac{1}{d}\frac{\mathrm{d}}{\mathrm{d}t}\mathrm{tr}\left(\left(\boldsymbol{I}_d-\boldsymbol{X}^{\top}\boldsymbol{W}_{\boldsymbol{I}}(t)\boldsymbol{S}_{\boldsymbol{I}}^{-1}\boldsymbol{X}\right)^{\top}\mathbf{\Sigma}_{\boldsymbol{X}}\left(\boldsymbol{I}_d-\boldsymbol{X}^{\top}\boldsymbol{W}_{\boldsymbol{I}}(t)\boldsymbol{S}_{\boldsymbol{I}}^{-1}\boldsymbol{X}\right)\right)$$

$$=-\frac{1}{nd}\mathrm{tr}\left(\mathbf{\Sigma}_{\boldsymbol{X}}\,\underbrace{\boldsymbol{X}^{\top}\boldsymbol{S}\exp\left(-\frac{t}{n}\boldsymbol{S}\right)\boldsymbol{S}^{-1}\boldsymbol{X}\left(\boldsymbol{I}_d-\boldsymbol{X}^{\top}\boldsymbol{W}_{\boldsymbol{I}}(t)\boldsymbol{S}_{\boldsymbol{I}}^{-1}\boldsymbol{X}\right)}_{p.s.d.}\right)<0.\tag{D.23}$$

Similarly, one can verify that the expected bias of NGD is monotonically decreasing for all choices of $\mathbf{\Sigma}_{\boldsymbol{X}}$ and $\mathbf{\Sigma}_{\boldsymbol{\theta}}$ satisfying (A2-4),

$$\frac{\mathrm{d}}{\mathrm{d}t}\mathrm{tr}\left(\mathbf{\Sigma}_{\boldsymbol{\theta}}\left(\boldsymbol{I}_d-\boldsymbol{F}^{-1}\boldsymbol{X}^{\top}\boldsymbol{W}_{\boldsymbol{F}^{-1}}(t)\boldsymbol{S}_{\boldsymbol{F}^{-1}}^{-1}\boldsymbol{X}\right)^{\top}\mathbf{\Sigma}_{\boldsymbol{X}}\left(\boldsymbol{I}_d-\boldsymbol{F}^{-1}\boldsymbol{X}^{\top}\boldsymbol{W}_{\boldsymbol{F}^{-1}}(t)\boldsymbol{S}_{\boldsymbol{F}^{-1}}^{-1}\boldsymbol{X}\right)\right)$$

$$=\frac{\mathrm{d}}{\mathrm{d}t}\mathrm{tr}\left(\mathbf{\Sigma}_{\boldsymbol{X}\boldsymbol{\theta}}\left(\boldsymbol{I}_d-\hat{\boldsymbol{X}}^{\top}\boldsymbol{W}_{\boldsymbol{F}^{-1}}(t)\boldsymbol{S}_{\boldsymbol{F}^{-1}}^{-1}\hat{\boldsymbol{X}}\right)^{\top}\left(\boldsymbol{I}_d-\hat{\boldsymbol{X}}^{\top}\boldsymbol{W}_{\boldsymbol{F}^{-1}}(t)\boldsymbol{S}_{\boldsymbol{F}^{-1}}^{-1}\hat{\boldsymbol{X}}\right)\right)\overset{(i)}{<}0,$$

where (i) follows from calculation similar to (D.23). Since the expected bias is decreasing through time for both GD and NGD when $\mathbf{\Sigma_\theta} = \mathbf{I}_d$, and from Theorem 3 we know that $B(\hat{\boldsymbol{\theta}}_I) \le B(\hat{\boldsymbol{\theta}}_{\mathbf{F}^{-1}})$, we conclude that $B^{\mathrm{opt}}(\boldsymbol{\theta}_I) \le B^{\mathrm{opt}}(\boldsymbol{\theta}_{\mathbf{F}^{-1}})$. □

## D.9 PROOF OF THEOREM 8

### D.9.1 SETUP AND MAIN RESULT

We restate the setting and assumptions. $\mathcal{H}$ is an RKHS included in $L_2(P_X)$ equipped with a bounded kernel function $k$ satisfying $\sup_{\mathrm{supp}(P_X)} k(\boldsymbol{x}, \boldsymbol{x}) \le 1$. $K_{\boldsymbol{x}} \in \mathcal{H}$ is the Riesz representation of the kernel function $k(x, \cdot)$, that is, $k(\boldsymbol{x}, \boldsymbol{y}) = \langle K_{\boldsymbol{x}}, K_{\boldsymbol{y}}\rangle_{\mathcal{H}}$. $S$ is the canonical embedding operator from $\mathcal{H}$ to $L_2(P_X)$. We write $\Sigma = S^*S : \mathcal{H} \to \mathcal{H}$ and $L = SS^*$. Note that the boundedness of the kernel gives $\|Sf\|_{L_2(P_X)} \le \sup_{\boldsymbol{x}} |f(\boldsymbol{x})| = \sup_{\boldsymbol{x}} |\langle K_{\boldsymbol{x}}, f\rangle| \le \|K_{\boldsymbol{x}}\|_{\mathcal{H}}\|f\|_{\mathcal{H}} \le \|f\|_{\mathcal{H}}$. Hence we know $\|\Sigma\| \le 1$ and $\|L\| \le 1$. Our analysis will be made under the following regularity assumptions.

- **(A4)**. There exist $r \in (0, \infty)$ and $M > 0$ s.t. $f^* = L^r h^*$ for some $h^* \in L_2(P_X)$ and $\|f^*\|_\infty \le M$.

- **(A5)**. There exists $s > 1$ s.t. $\mathrm{tr}(\Sigma^{1/s}) < \infty$ and $2r + s^{-1} > 1$.

- **(A6)**. There exist $\mu \in [s^{-1}, 1]$ and $C_\mu > 0$ s.t. $\sup_{\boldsymbol{x} \in \mathrm{supp}(P_X)} \|\Sigma^{1/2-1/\mu} K_{\boldsymbol{x}}\|_{\mathcal{H}} \le C_\mu$.

(A5)(A6) are standard regularity assumptions in the literature that provide capacity control of the RKHS (e.g., see Caponnetto & De Vito (2007); Pillaud-Vivien et al. (2018)). For (A4), it is worth noting that previous works mostly consider $r \ge 1/2$ which implies that $f^* \in \mathcal{H}$.

The training data is generated as $y_i = f^*(\boldsymbol{x}_i) + \varepsilon_i$, where $\varepsilon_i$ is an i.i.d. noise satisfying $|\varepsilon_i| \le \sigma$ almost surely. Let $\boldsymbol{y} \in \mathbb{R}^n$ be the label vector. We identify $\mathbb{R}^n$ with $L_2(P_n)$ and define

$$\hat{\Sigma} = \frac{1}{n}\sum_{i=1}^n K_{\boldsymbol{x}_i} \otimes K_{\boldsymbol{x}_i} : \mathcal{H} \to \mathcal{H}, \quad \hat{S}^*Y = \frac{1}{n}\sum_{i=1}^n Y_i K_{\boldsymbol{x}_i}, \ (Y \in L_2(P_n)).$$

We consider the following preconditioned update on $f_t \in \mathcal{H}$:

$$f_t = f_{t-1} - \eta(\Sigma + \lambda I)^{-1}(\hat{\Sigma} f_{t-1} - \hat{S}^*Y), \quad f_0 = 0.$$

We briefly comment on how our analysis differs from Rudi et al. (2017), which showed that a preconditioned update (the FALKON algorithm) for kernel ridge regression can also achieve accelerated convergence in the population risk. We emphasize the following differences.

- The two algorithms optimize different objectives, as highlighted by the different role of the "ridge" coefficient $\lambda$. In FALKON, $\lambda$ turns the objective into kernel ridge regression; whereas in our (4.1), $\lambda$ controls the interpolation between GD and NGD. As we aim to study how the preconditioner affects generalization, it is important that we look at the objective in its original (instead of regularized) form.

- To elaborate on the first point, since FALKON minimizes a regularized objective, it would not overfit even after large number of gradient steps, but it is unclear how preconditioning impacts generalization (i.e., any preconditioner may generalize well with proper regularization). In contrast, we consider the unregularized objective, and thus early stopping plays a crucial role in avoiding overfitting – this differs from most standard analysis of GD.

- Algorithm-wise, the two updates employ different preconditioners. FALKON involves inverting the kernel matrix $K$ defined on the training points, whereas we consider the population covariance operator $\Sigma$, which is consistent with our earlier discussion on the population Fisher in Section 3.

- In terms of the theoretical setup, our analysis allows for $r < 1/2$, whereas Rudi et al. (2017) and many other previous works assumed $r \in [1/2, 1]$, as commented in Section 4.3.

We aim to show the following theorem:

**Theorem** (Formal Statement of Theorem 8). *Given (A4-6), if the sample size $n$ is sufficiently large so that $1/(n\lambda) \ll 1$, then for $\eta < \|\Sigma\|$ with $\eta t \ge 1$ and $0 < \delta < 1$ and $0 < \lambda < 1$, it holds that*

$$\|Sf_t - f^*\|_{L_2(P_X)}^2 \le C(B(t) + V(t)),$$

*with probability $1 - 3\delta$, where $C$ is a constant and*

$$B(t) := \exp(-\eta t) \vee \left(\frac{\lambda}{\eta t}\right)^{2r},$$

$$V(t) := V_1(t) + (1 + \eta t)\left(\frac{\lambda^{-1}B(t) + \sigma^2 \mathrm{tr}\left(\Sigma^{\frac{1}{s}}\right)\lambda^{-\frac{1}{s}}}{n} + \frac{\lambda^{-1}(\sigma + M + (1 + t\eta)\lambda^{-(\frac{1}{2}-r)_+})^2}{n^2}\right)\log(1/\delta)^2,$$

*in which*

$$V_1(t) := \left[\exp(-\eta t) \vee \left(\frac{\lambda}{\eta t}\right)^{2r} + (t\eta)^2\left(\frac{\beta'(1 \vee \lambda^{2r-\mu})\mathrm{tr}\left(\Sigma^{\frac{1}{s}}\right)\lambda^{-\frac{1}{s}}}{n} + \frac{\beta'^2(1 + \lambda^{-\mu}(1 \vee \lambda^{2r-\mu}))}{n^2}\right)\right](1 + t\eta)^2,$$

*for $\beta' = \log\left(\frac{28C_\mu^2(2^{2r-\mu}\vee\lambda^{-\mu+2r})\mathrm{tr}(\Sigma^{1/s})\lambda^{-1/s}}{\delta}\right)$. When $r \geq 1/2$, if we set $\lambda = n^{-\frac{s}{2rs+1}} =: \lambda^*$ and $t = \Theta(\log(n))$, then the overall convergence rate becomes*

$$\|Sg_t - f^*\|_{L_2(P_X)}^2 = \widetilde{O}_p\left(n^{-\frac{2rs}{2rs+1}}\right),$$

*which is the minimax optimal rate ($\widetilde{O}_p(\cdot)$ hides a poly-$\log(n)$ factor). On the other hand, when $r < 1/2$, the bound is also $\widetilde{O}_p\left(n^{-\frac{2rs}{2rs+1}}\right)$ except the term $V_1(t)$. In this case, if $2r \geq \mu$ holds additionally, we have $V_t(t) = \widetilde{O}_p\left(n^{-\frac{2rs}{2rs+1}}\right)$, which again recovers the optimal rate.*

Note that if the GD (with iterates $\tilde{f}_t$) is employed, from previous work Lin & Rosasco (2017) we know that the *bias term* $\left(\frac{\lambda}{\eta t}\right)^{2r}$ is replaced by $\left(\frac{1}{\eta t}\right)^{2r}$, and therefore the upper bound translates to

$$\|S\tilde{f}_t - f^*\|_{L_2(P_X)}^2 \leq C\left\{(\eta t)^{-2r} + \frac{1}{n}\left(\mathrm{tr}\left(\Sigma^{1/s}\right)(\eta t)^{1/s} + \frac{\eta t}{n}\right)\left(\sigma^2 + \left(\frac{1}{\eta t}\right)^{2r} + \frac{M^2 + (\eta t)^{-(2r-1)}}{n}\right)\right\},$$

with high probability. In other words, by the condition $\eta = O(1)$, we need $t = \Theta(n^{\frac{2rs}{2rs+1}})$ steps to sufficiently diminish the bias term. In contrast, the preconditioned update that interpolates between GD and NGD (4.1) only require $t = O(\log(n))$ steps to make the bias term negligible. This is because the NGD amplifies the high frequency component and rapidly captures the detailed "shape" of the target function $f^*$.

### D.9.2 PROOF OF MAIN RESULT

**Proof.** We follow the proof strategy of Lin & Rosasco (2017). First we define a reference optimization problem with iterates $\bar{f}_t$ that directly minimize the population risk:

$$\bar{f}_t = \bar{f}_{t-1} - \eta(\Sigma + \lambda I)^{-1}(\Sigma\bar{f}_{t-1} - S^*f^*), \quad \bar{f}_0 = 0.$$

Note that $\mathbb{E}[f_t] = \bar{f}_t$. In addition, we define the degrees of freedom and its related quantity as

$$\mathcal{N}_\infty(\lambda) := \mathbb{E}_{\boldsymbol{x}}[\langle K_{\boldsymbol{x}}, \Sigma_\lambda^{-1}K_{\boldsymbol{x}}\rangle_{\mathcal{H}}] = \mathrm{tr}\left(\Sigma\Sigma_\lambda^{-1}\right), \quad \mathcal{F}_\infty(\lambda) := \sup_{\boldsymbol{x}\in\mathrm{supp}(P_X)}\|\Sigma_\lambda^{-1/2}K_{\boldsymbol{x}}\|_{\mathcal{H}}^2.$$

We can see that the risk admits the following bias-variance decomposition

$$\|Sf_t - f^*\|_{L_2(P_X)}^2 \leq 2(\underbrace{\|Sf_t - S\bar{f}_t\|_{L_2(P_X)}^2}_{V(t),\text{ variance}} + \underbrace{\|\bar{f}_t - f^*\|_{L_2(P_X)}^2}_{B(t),\text{ bias}}).$$

We upper bound the bias and variance separately.

**Bounding the bias term $B(t)$:** Note that by the update rule (4.1), it holds that

$$S\bar{f}_t - f^* = S\bar{f}_{t-1} - f^* - \eta S(\Sigma + \lambda I)^{-1}(\Sigma \bar{f}_{t-1} - S^* f^*)$$
$$\Leftrightarrow \quad S\bar{f}_t - f^* = (I - \eta S(\Sigma + \lambda I)^{-1}S^*)(S\bar{f}_{t-1} - f^*).$$

Hence, unrolling the recursion gives $S\bar{f}_t - f^* = (I - \eta S(\Sigma+\lambda I)^{-1}S^*)^t(S\bar{f}_0 - f^*) = (I - \eta S(\Sigma + \lambda I)^{-1}S^*)^t(-f^*) = -(I - \eta S(\Sigma + \lambda I)^{-1}S^*)^t L^r h^*$. Write the spectral decomposition of $L$ as $L = \sum_{j=1}^\infty \sigma_j \phi_j \phi_j^*$ for $\phi_j \in L_2(P_X)$ for $\sigma_j \geq 0$. We have $\|(I - \eta S(\Sigma+\lambda I)^{-1}S^*)^t L^r h^*\|_{L_2(P_X)} = \sum_{j=1}^\infty(1 - \eta\frac{\sigma_j}{\sigma_j+\lambda})^{2t}\sigma_j^{2r}h_j^2$, where $h = \sum_{j=1}^\infty h_j\phi_j$. We then apply Lemma 11 to obtain

$$B(t) \leq \exp(-\eta t)\sum_{j:\sigma_j\geq\lambda} h_j^2 + \left(\frac{2r}{e}\frac{\lambda}{\eta t}\right)^{2r}\sum_{j:\sigma_j<\lambda} h_j^2 \leq C\left[\exp(-\eta t) \vee \left(\frac{\lambda}{\eta t}\right)^{2r}\right]\|h^*\|_{L_2(P_X)}^2,$$

where $C$ is a constant depending only on $r$.

**Bounding the variance term $V(t)$:** We now handle the variance term $V(t)$. For notational convenience, we write $A_\lambda := A + \lambda I$ for a linear operator $A$ from a Hilbert space $H$ to $H$. By the definition of $f_t$, we know

$$f_t = (I - \eta(\Sigma + \lambda I)^{-1}\hat{\Sigma})f_{t-1} + \eta(\Sigma + \lambda I)^{-1}\hat{S}^*Y$$
$$= \sum_{j=0}^{t-1}(I - \eta(\Sigma + \lambda I)^{-1}\hat{\Sigma})^j\eta(\Sigma + \lambda I)^{-1}\hat{S}^*Y$$
$$= \Sigma_\lambda^{-1/2}\eta\left[\sum_{j=0}^{t-1}(I - \eta\Sigma_\lambda^{-1/2}\hat{\Sigma}\Sigma_\lambda^{-1/2})^j\right]\Sigma_\lambda^{-1/2}\hat{S}^*Y =: \Sigma_\lambda^{-1/2}G_t\Sigma_\lambda^{-1/2}\hat{S}^*Y,$$

where we defined $G_t := \eta\left[\sum_{j=0}^{t-1}(I - \eta\Sigma_\lambda^{-1/2}\hat{\Sigma}\Sigma_\lambda^{-1/2})^j\right]$. Accordingly, we decompose $V(t)$ as

$$\|Sf_t - S\bar{f}_t\|_{L_2(P_X)}^2 \leq 2(\underbrace{\|S(f_t - \Sigma_\lambda^{-1/2}G_t\Sigma_\lambda^{-1/2}\hat{\Sigma}\bar{f}_t)\|_{L_2(P_X)}^2}_{(a)}$$
$$+ \underbrace{\|S(\Sigma_\lambda^{-1/2}G_t\Sigma_\lambda^{-1/2}\hat{\Sigma}\bar{f}_t - \bar{f}_t)\|_{L_2(P_X)}^2}_{(b)}).$$

We bound $(a)$ and $(b)$ separately.

**Step 1. Bounding $(a)$.** Decompose $(a)$ as

$$\|S(f_t - \Sigma_\lambda^{-1/2}G_t\Sigma_\lambda^{-1/2}\hat{\Sigma}\bar{f}_t)\|_{L_2(P_X)}^2 = \|S\Sigma_\lambda^{-1/2}G_t\Sigma_\lambda^{-1/2}(\hat{S}^*Y - \hat{\Sigma}\bar{f}_t)\|_{L_2(P_X)}^2$$
$$\leq \|S\Sigma_\lambda^{-1/2}\|^2\|G_t\Sigma_\lambda^{-1/2}\hat{\Sigma}_\lambda\Sigma_\lambda^{-1/2}\|^2\|\Sigma_\lambda^{1/2}\hat{\Sigma}_\lambda^{-1}\Sigma_\lambda^{1/2}\|^2\|\Sigma_\lambda^{-1/2}(\hat{S}^*Y - \hat{\Sigma}\bar{f}_t)\|_{\mathcal{H}}^2.$$

We bound the terms in the RHS individually.

**(i)** $\|S\Sigma_\lambda^{-1/2}\|^2 = \|\Sigma_\lambda^{-1/2}\Sigma\Sigma_\lambda^{-1/2}\| \leq 1$.

**(ii)** Note that $\Sigma_\lambda^{-1/2}\hat{\Sigma}_\lambda\Sigma_\lambda^{-1/2} = I - \Sigma_\lambda^{-1/2}(\Sigma - \hat{\Sigma})\Sigma_\lambda^{-1/2} \succeq (1 - \|\Sigma_\lambda^{-1/2}(\Sigma - \hat{\Sigma})\Sigma_\lambda^{-1/2}\|)I$.

Proposition 6 of Rudi & Rosasco (2017) and its proof implies that for $\lambda \leq \|\Sigma\|$ and $0 < \delta < 1$,

$$\|\Sigma_\lambda^{-1/2}(\Sigma - \hat{\Sigma})\Sigma_\lambda^{-1/2}\| \leq \sqrt{\frac{2\beta\mathcal{F}_\infty(\lambda)}{n}} + \frac{2\beta(1 + \mathcal{F}_\infty(\lambda))}{3n} =: \Xi_n, \qquad (D.24)$$

with probability $1 - \delta$, where $\beta = \log\left(\frac{4\text{tr}(\Sigma\Sigma_\lambda^{-1})}{\delta}\right) = \log\left(\frac{4\mathcal{N}_\infty(\lambda)}{\delta}\right)$. By Lemma 14, $\beta \leq \log\left(\frac{4\text{tr}(\Sigma^{1/s})\lambda^{-1/s}}{\delta}\right)$ and $\mathcal{F}_\infty(\lambda) \leq \lambda^{-1}$. Therefore, if $\lambda = o(n^{-1}\log(n))$ and $\lambda = \Omega(n^{-1/s})$, the RHS can be smaller than $1/2$ for sufficiently large $n$, i.e. $\Xi_n = O(\sqrt{\log(n)/(n\lambda)}) \leq 1/2$; thus,

$$\Sigma_\lambda^{-1/2}\hat{\Sigma}_\lambda\Sigma_\lambda^{-1/2} \succeq \frac{1}{2}I.$$

We denote this event as $\mathcal{E}_1$.

**(iii)** Note that

$$
\begin{aligned}
G_t \Sigma_\lambda^{-1/2} \hat{\Sigma}_\lambda \Sigma_\lambda^{-1/2} &= \eta \left[ \sum_{j=0}^{t-1} (I - \eta \Sigma_\lambda^{-1/2} \hat{\Sigma} \Sigma_\lambda^{-1/2})^j \right] \Sigma_\lambda^{-1/2} \hat{\Sigma}_\lambda \Sigma_\lambda^{-1/2} \\
&= \eta \left[ \sum_{j=0}^{t-1} (I - \eta \Sigma_\lambda^{-1/2} \hat{\Sigma} \Sigma_\lambda^{-1/2})^j \right] (\Sigma_\lambda^{-1/2} \hat{\Sigma} \Sigma_\lambda^{-1/2} + \lambda \Sigma_\lambda^{-1}).
\end{aligned}
$$

Thus, by Lemma 12 we have

$$
\begin{aligned}
& \| G_t \Sigma_\lambda^{-1/2} \hat{\Sigma}_\lambda \Sigma_\lambda^{-1/2} \| \\
& \leq \underbrace{\left\| \eta \left[ \sum_{j=0}^{t-1} (I - \eta \Sigma_\lambda^{-1/2} \hat{\Sigma} \Sigma_\lambda^{-1/2})^j \right] \Sigma_\lambda^{-1/2} \hat{\Sigma} \Sigma_\lambda^{-1/2} \right\|}_{\leq 1 \text{ (due to Lemma 12)}} + \left\| \eta \left[ \sum_{j=0}^{t-1} (I - \eta \Sigma_\lambda^{-1/2} \hat{\Sigma} \Sigma_\lambda^{-1/2})^j \right] \lambda \Sigma_\lambda^{-1} \right\| \\
& \leq 1 + \eta \sum_{j=0}^{t-1} \| (I - \eta \Sigma_\lambda^{-1/2} \hat{\Sigma} \Sigma_\lambda^{-1/2})^j \| \| \lambda \Sigma_\lambda^{-1} \| \leq 1 + \eta t.
\end{aligned}
$$

**(iv)** Note that

$$
\| \Sigma_\lambda^{-1/2} (\hat{S}^* Y - \hat{\Sigma} \bar{f}_t) \|_{\mathcal{H}}^2 \leq 2 ( \| \Sigma_\lambda^{-1/2} [ (\hat{S}^* Y - \hat{\Sigma} \bar{f}_t) - (S^* f^* - \Sigma \bar{f}_t) ] \|_{\mathcal{H}}^2 + \| \Sigma_\lambda^{-1/2} (S^* f^* - \Sigma \bar{f}_t) \|_{\mathcal{H}}^2 ).
$$

First we bound the first term of the RHS. Let $\xi_i = \Sigma_\lambda^{-1/2} [ K_{\boldsymbol{x}_i} y_i - K_{\boldsymbol{x}_i} \bar{f}_t(\boldsymbol{x}_i) - (S^* f^* - \Sigma \bar{f}_t) ]$. Then, $\{\xi_i\}_{i=1}^n$ is an i.i.d. sequence of zero-centered random variables taking value in $\mathcal{H}$ and thus

$$
\| \Sigma_\lambda^{-1/2} [ (\hat{S}^* Y - \hat{\Sigma} \bar{f}_t) - (S^* f^* - \Sigma \bar{f}_t) ] \|_{\mathcal{H}}^2 = \left\| \frac{1}{n} \sum_{i=1}^n \xi_i \right\|_{\mathcal{H}}^2.
$$

The RHS can be bounded by using Bernstein's inequality in Hilbert space Caponnetto & De Vito (2007). To apply the inequality, we need to bound the variance and sup-norm of the random variable. The variance can be bounded as

$$
\begin{aligned}
\mathbb{E}[\| \xi_i \|_{\mathcal{H}}^2] &\leq \mathbb{E}_{(\boldsymbol{x},y)} \left[ \| \Sigma_\lambda^{-1/2} (K_{\boldsymbol{x}} (f^*(\boldsymbol{x}) - \bar{f}_t(\boldsymbol{x})) + K_{\boldsymbol{x}} \epsilon) \|_{\mathcal{H}}^2 \right] \\
&\leq 2 \left\{ \mathbb{E}_{(\boldsymbol{x},y)} \left[ \| \Sigma_\lambda^{-1/2} (K_{\boldsymbol{x}} (f^*(\boldsymbol{x}) - \bar{f}_t(x))) \|_{\mathcal{H}}^2 + \| \Sigma_\lambda^{-1/2} (K_{\boldsymbol{x}} \epsilon) \|_{\mathcal{H}}^2 \right] \right\} \\
&\leq 2 \left\{ \sup_{\boldsymbol{x} \in \mathrm{supp}(P_X)} \| \Sigma_\lambda^{-1/2} K_{\boldsymbol{x}} \|^2 \| f^* - S \bar{f}_t \|_{L_2(P_X)}^2 + \sigma^2 \mathrm{tr}(\Sigma_\lambda^{-1} \Sigma) \right\} \\
&\leq 2 \{ \mathcal{F}_\infty(\lambda) B(t) + \sigma^2 \mathrm{tr}(\Sigma_\lambda^{-1} \Sigma) \} \\
&\leq 2 \{ \lambda^{-1} B(t) + \sigma^2 \mathrm{tr}(\Sigma_\lambda^{-1} \Sigma) \},
\end{aligned}
$$

The sup-norm can be bounded as follows. Observe that $\| \bar{f}_t \|_\infty \leq \| \bar{f}_t \|_{\mathcal{H}}$, and thus by Lemma 13,

$$
\begin{aligned}
\| \xi_i \|_{\mathcal{H}} &\leq 2 \sup_{\boldsymbol{x} \in \mathrm{supp}(P_X)} \| \Sigma_\lambda^{-1/2} K_{\boldsymbol{x}} \|_{\mathcal{H}} (\sigma + \| f^* \|_\infty + \| \bar{f}_t \|_\infty) \\
&\lesssim \mathcal{F}_\infty^{1/2}(\lambda) (\sigma + M + (1 + t\eta) \lambda^{-(1/2-r)_+}) \\
&\lesssim \lambda^{-1/2} (\sigma + M + (1 + t\eta) \lambda^{-(1/2-r)_+}).
\end{aligned}
$$

Hence, for $0 < \delta < 1$, Bernstein's inequality (Proposition 2 of Caponnetto & De Vito (2007)) gives

$$
\left\| \frac{1}{n} \sum_{i=1}^n \xi_i \right\|_{\mathcal{H}}^2 \leq C \left( \sqrt{\frac{\lambda^{-1} B(t) + \sigma^2 \mathrm{tr}(\Sigma_\lambda^{-1} \Sigma)}{n}} + \frac{\lambda^{-1/2} (\sigma + M + (1 + t\eta) \lambda^{-(1/2-r)_+})}{n} \right)^2 \log(1/\delta)^2
$$

with probability $1 - \delta$ where $C$ is a universal constant. We define this event as $\mathcal{E}_2$.

For the second term $\|\Sigma_\lambda^{-1/2}(S^* f^* - \Sigma \bar{f}_t)\|_{\mathcal{H}}^2$ we have

$$\|\Sigma_\lambda^{-1/2}(S^* f^* - \Sigma \bar{f}_t)\|_{\mathcal{H}}^2 \leq \|\Sigma_\lambda^{1/2}(f^* - S\bar{f}_t)\|_{\mathcal{H}}^2 = \|f^* - S\bar{f}_t\|_{L_2(P_X)}^2 \leq B(t).$$

Combining these evaluations, on the event $\mathcal{E}_2$ where $P(\mathcal{E}_2) \geq 1 - \delta$ for $0 < \delta < 1$ we have

$$\|\Sigma_\lambda^{-1/2}(\hat{S}^* Y - \hat{\Sigma}\bar{f}_t)\|_{\mathcal{H}}^2$$

$$\overset{(i)}{\leq} C \left( \sqrt{\frac{\lambda^{-1} B(t) + \sigma^2 \mathrm{tr}\left(\Sigma_\lambda^{-1}\Sigma\right)}{n}} + \frac{\lambda^{-1/2}(\sigma + M + (1 + t\eta)\lambda^{-(1/2-r)_+})}{n} \right)^2 \log(1/\delta)^2 + B(t).$$

where we used Lemma 14 in (i).

**Step 2. Bounding** $(b)$. On the event $\mathcal{E}_1$, the term $(b)$ can be evaluated as

$$\|S(\Sigma_\lambda^{-1/2} G_t \Sigma_\lambda^{-1/2} \hat{\Sigma} \bar{f}_t - \bar{f}_t)\|_{L_2(P_X)}^2$$

$$\leq \|\Sigma^{1/2}(\Sigma_\lambda^{-1/2} G_t \Sigma_\lambda^{-1/2} \hat{\Sigma} \bar{f}_t - \bar{f}_t)\|_{\mathcal{H}}^2$$

$$\leq \|\Sigma^{1/2} \Sigma_\lambda^{-1/2}(G_t \Sigma_\lambda^{-1/2} \hat{\Sigma} \Sigma_\lambda^{-1/2} - I)\Sigma_\lambda^{1/2} \bar{f}_t\|_{\mathcal{H}}^2$$

$$\leq \|\Sigma^{1/2} \Sigma_\lambda^{-1/2}\| \|(G_t \Sigma_\lambda^{-1/2} \hat{\Sigma} \Sigma_\lambda^{-1/2} - I)\Sigma_\lambda^{1/2} \bar{f}_t\|_{\mathcal{H}}^2$$

$$\leq \|(G_t \Sigma_\lambda^{-1/2} \hat{\Sigma} \Sigma_\lambda^{-1/2} - I)\Sigma_\lambda^{1/2} \bar{f}_t\|_{\mathcal{H}}^2. \tag{D.25}$$

where we used Lemma 13 in the last inequality. The term $\|(G_t \Sigma_\lambda^{-1/2} \hat{\Sigma} \Sigma_\lambda^{-1/2} - I)\Sigma_\lambda^{1/2} f_t\|_{\mathcal{H}}$ can be bounded as follows. First, note that

$$(G_t \Sigma_\lambda^{-1/2} \hat{\Sigma} \Sigma_\lambda^{-1/2} - I)\Sigma_\lambda^{1/2} = \left\{ \eta \left[ \sum_{j=0}^{t-1}(I - \eta \Sigma_\lambda^{-1/2} \hat{\Sigma} \Sigma_\lambda^{-1/2})^j \right] \Sigma_\lambda^{-1/2} \hat{\Sigma} \Sigma_\lambda^{-1/2} - I \right\} \Sigma_\lambda^{1/2}$$

$$= (I - \eta \Sigma_\lambda^{-1/2} \hat{\Sigma} \Sigma_\lambda^{-1/2})^t \Sigma_\lambda^{1/2}.$$

Therefore, the RHS of (D.25) can be further bounded by

$$\|(I - \eta \Sigma_\lambda^{-1/2} \hat{\Sigma} \Sigma_\lambda^{-1/2})^t \Sigma_\lambda^{1/2} \bar{f}_t\|_{\mathcal{H}}$$

$$= \|(I - \eta \Sigma_\lambda^{-1/2} \Sigma \Sigma_\lambda^{-1/2} + \eta \Sigma_\lambda^{-1/2}(\Sigma - \hat{\Sigma})\Sigma_\lambda^{-1/2})^t \Sigma_\lambda^{1/2} \bar{f}_t\|_{\mathcal{H}}$$

$$= \| \sum_{k=0}^{t-1}(I - \eta \Sigma_\lambda^{-1/2} \hat{\Sigma} \Sigma_\lambda^{-1/2})^k (\eta \Sigma_\lambda^{-1/2}(\Sigma - \hat{\Sigma})\Sigma_\lambda^{-1/2})(I - \eta \Sigma_\lambda^{-1}\Sigma)^{t-k-1} \Sigma_\lambda^{1/2} \bar{f}_t - (I - \eta \Sigma_\lambda^{-1}\Sigma)^t \Sigma_\lambda^{1/2} \bar{f}_t\|_{\mathcal{H}}$$

$$\overset{(i)}{\leq} \|(I - \eta \Sigma_\lambda^{-1}\Sigma)^t \Sigma_\lambda^{1/2} \bar{f}_t\|_{\mathcal{H}}$$

$$+ \eta \sum_{k=0}^{t-1} \|(I - \eta \Sigma_\lambda^{-1/2} \hat{\Sigma} \Sigma_\lambda^{-1/2})^k \Sigma_\lambda^{-1/2}(\Sigma - \hat{\Sigma})\Sigma_\lambda^{-1/2+r}(I - \eta \Sigma_\lambda^{-1}\Sigma)^{t-k-1} \Sigma_\lambda^{1/2-r} \bar{f}_t\|_{\mathcal{H}}$$

$$\leq \|(I - \eta \Sigma_\lambda^{-1}\Sigma)^t \Sigma_\lambda^{1/2} \bar{f}_t\|_{\mathcal{H}} + t\eta \|\Sigma_\lambda^{-1/2}(\Sigma - \hat{\Sigma})\Sigma_\lambda^{-1/2+r}\| \|\Sigma_\lambda^{1/2-r} \bar{f}_t\|_{\mathcal{H}}$$

$$= \|(I - \eta \Sigma_\lambda^{-1}\Sigma)^t \Sigma_\lambda^r\| \|\Sigma_\lambda^{1/2-r} \bar{f}_t\|_{\mathcal{H}} + t\eta \|\Sigma_\lambda^{-1/2}(\Sigma - \hat{\Sigma})\Sigma_\lambda^{-1/2+r}\| \|\Sigma_\lambda^{1/2-r} \bar{f}_t\|_{\mathcal{H}}$$

$$\lesssim \|(I - \eta \Sigma_\lambda^{-1}\Sigma)^t \Sigma_\lambda^r\| + t\eta \|\Sigma_\lambda^{-1/2}(\Sigma - \hat{\Sigma})\Sigma_\lambda^{-1/2+r}\|(1 + t\eta)\|h^*\|_{L_2(P_X)}, \tag{D.26}$$

where (i) is due to exchangeability of $\Sigma_\lambda$ and $\Sigma$. By Lemma 11, for the RHS we have

$$\|(I - \eta \Sigma_\lambda^{-1}\Sigma)^t \Sigma_\lambda^r\| \leq \exp(-\eta t/2) \vee \left( \frac{1}{e} \frac{\lambda}{\eta t} \right)^r.$$

Next, as in the (D.24), by applying the Bernstein inequality for asymmetric operators (Corollary 3.1 of Minsker (2017) with the argument in its Section 3.2), it holds that

$$\|\Sigma_\lambda^{-1/2}(\Sigma - \hat{\Sigma})\Sigma_\lambda^{-1/2+r}\|$$

$$\leq C'\left(\sqrt{\frac{\beta'C_\mu^2(2^{2r-\mu}\vee\lambda^{2r-\mu})\mathcal{N}_\infty(\lambda)}{n}}+\frac{\beta'((1+\lambda)^r+C_\mu^2\lambda^{-\mu/2}(2^{2r-\mu}\vee\lambda^{r-\mu/2}))}{n}\right)=:\Xi'_n,$$

with probability $1-\delta$, where $C'$ is a universal constant and $\beta' \leq$ $\log\left(\frac{28C_\mu^2(2^{2r-\mu}\vee\lambda^{-\mu+2r})\mathrm{tr}(\Sigma^{1/s})\lambda^{-1/s}}{\delta}\right)$. We also used the following bounds on the sup-norm and the second order moments:

$$\begin{aligned}
\text{(sup-norm)}\quad &\|\Sigma_\lambda^{-1/2}(K_{\boldsymbol{x}}K_{\boldsymbol{x}}^*-\Sigma)\Sigma_\lambda^{-1/2+r}\|\\
&\leq\|\Sigma_\lambda^{-1/2}K_{\boldsymbol{x}}K_{\boldsymbol{x}}^*\Sigma_\lambda^{-1/2+r}\|+\|\Sigma_\lambda^r\|\\
&\leq\|\Sigma_\lambda^{-\mu/2}\Sigma_\lambda^{\mu/2-1/2}K_{\boldsymbol{x}}K_{\boldsymbol{x}}^*\Sigma_\lambda^{-1/2+\mu/2}\Sigma_\lambda^{r-\mu/2}\|+\|\Sigma_\lambda^r\|\\
&\leq C_\mu^2\lambda^{-\mu/2}(2^{r-\mu/2}\vee\lambda^{r-\mu/2})+(1+\lambda)^r\quad\text{(a.s.)},
\end{aligned}$$

$$\begin{aligned}
\text{(2nd order moment 1)}\quad &\|\mathbb{E}_{\boldsymbol{x}}[\Sigma_\lambda^{-1/2}(K_{\boldsymbol{x}}K_{\boldsymbol{x}}^*-\Sigma)\Sigma_\lambda^{-1+2r}(K_{\boldsymbol{x}}K_{\boldsymbol{x}}^*-\Sigma)\Sigma_\lambda^{-1/2}]\|\\
&\leq\|\Sigma_\lambda^{-1/2}\Sigma\Sigma_\lambda^{-1/2}\|\sup_{\boldsymbol{x}\in\mathrm{supp}(P_X)}[K_{\boldsymbol{x}}^*\Sigma_\lambda^{-1/2+\mu/2}\Sigma_\lambda^{-\mu+2r}\Sigma_\lambda^{-1/2+\mu/2}K_{\boldsymbol{x}}]\\
&\leq C_\mu^2(2^{2r-\mu}\vee\lambda^{2r-\mu}),
\end{aligned}$$

$$\begin{aligned}
\text{(2nd order moment 2)}\quad &\|\mathbb{E}_{\boldsymbol{x}}[\Sigma_\lambda^{-1/2+r}(K_{\boldsymbol{x}}K_{\boldsymbol{x}}^*-\Sigma)\Sigma_\lambda^{-1/2}\Sigma_\lambda^{-1/2}(K_{\boldsymbol{x}}K_{\boldsymbol{x}}^*-\Sigma)\Sigma_\lambda^{-1/2+r}]\|\\
&\leq\|\mathbb{E}_{\boldsymbol{x}}[\Sigma_\lambda^{-1/2+r}K_{\boldsymbol{x}}K_{\boldsymbol{x}}^*\Sigma_\lambda^{-1}K_{\boldsymbol{x}}K_{\boldsymbol{x}}^*\Sigma_\lambda^{-1/2+r}]\|\\
&\leq C_\mu^2(2^{2r-\mu}\vee\lambda^{2r-\mu})\mathbb{E}_{\boldsymbol{x}}[K_{\boldsymbol{x}}^*\Sigma_\lambda^{-1}K_{\boldsymbol{x}}]\\
&= C_\mu^2(2^{2r-\mu}\vee\lambda^{2r-\mu})\mathrm{tr}(\Sigma\Sigma_\lambda^{-1})\\
&= C_\mu^2(2^{2r-\mu}\vee\lambda^{2r-\mu})\mathcal{N}_\infty(\lambda).
\end{aligned}$$

We define this event as $\mathcal{E}_3$. Therefore, the RHS of (D.26) can be further bounded by

$$[\|(I-\eta\Sigma_\lambda^{-1}\Sigma)^t\Sigma_\lambda^r\|+Ct\eta\|\Sigma_\lambda^{-1/2}(\Sigma-\hat{\Sigma})\Sigma_\lambda^{-1/2+r}\|](1+t\eta)\|h^*\|_{L_2(P_X)}$$
$$\leq\left[\exp(-\eta t/2)\vee\left(\frac{1}{e}\frac{\lambda}{\eta t}\right)^r+t\eta\Xi'_n\right](1+t\eta)\|h^*\|_{L_2(P_X)}.$$

Finally, note that when $\lambda=\lambda^*$ and $2r\geq\mu$,

$$\Xi_n'^2=\tilde{O}\left(\frac{\lambda^{*2r-\mu-1/s}}{n}+\frac{\lambda^{*2(r-\mu)}}{n^2}\right)\leq\tilde{O}(n^{-\frac{s(4r-\mu)}{2rs+1}}+n^{-\frac{s(4r-2\mu)+2}{2rs+1}})\leq\tilde{O}(n^{-\frac{2rs}{2rs+1}}).$$

**Step 3.** Combining the calculations in Step 1 and 2 leads to the desired result. $\qquad\square$

### D.9.3 AUXILIARY LEMMAS

**Lemma 11.** *For $t\in\mathbb{N}$, $0<\eta<1$, $0<\sigma\leq 1$ and $0\leq\lambda$, it holds that*

$$\left(1-\eta\frac{\sigma}{\sigma+\lambda}\right)^t\sigma^r\leq\begin{cases}\exp(-\eta t/2)&(\sigma\geq\lambda)\\\left(\frac{2r}{e}\frac{\lambda}{\eta t}\right)^r&(\sigma<\lambda)\end{cases}.$$

**Proof.** When $\sigma\geq\lambda$, we have

$$\left(1-\eta\frac{\sigma}{\sigma+\lambda}\right)^t\sigma^r\leq\left(1-\eta\frac{\sigma}{2\sigma}\right)^t\sigma^r=(1-\eta/2)^t\sigma^r\leq\exp(-t\eta/2)\sigma^r\leq\exp(-t\eta/2)$$

due to $\sigma\leq 1$. On the other hand, note that

$$\left(1-\eta\frac{\sigma}{\sigma+\lambda}\right)^t\sigma^r\leq\exp\left(-\eta t\frac{\sigma}{\sigma+\lambda}\right)\times\left(\frac{\sigma\eta t}{\sigma+\lambda}\right)^r\left(\frac{\sigma+\lambda}{\eta t}\right)^r$$

$$\leq \sup_{x>0} \exp(-x)x^r \left(\frac{\sigma + \lambda}{\eta t}\right)^r \leq \left(\frac{(\sigma + \lambda)r}{\eta te}\right)^r,$$

where we used $\sup_{x>0} \exp(-x)x^r = (r/e)^r$. $\qquad\square$

**Lemma 12.** *For $t = \mathbb{N}$, $0 < \eta$ and $0 \leq \sigma$ such that $\eta\sigma < 1$, it holds that $\eta \sum_{j=0}^{t-1}(1 - \eta\sigma)^j \sigma \leq 1$.*

**Proof.** If $\sigma = 0$, then the statement is obvious. Assume that $\sigma > 0$, then

$$\sum_{j=0}^{t-1}(1 - \eta\sigma)^j \sigma = \frac{1 - (1 - \eta\sigma)^t}{1 - (1 - \eta\sigma)}\sigma = \frac{1}{\eta}[1 - (1 - \eta\sigma)^t] \leq \eta^{-1}.$$

This yields the desired claim. $\qquad\square$

**Lemma 13.** *Under (A4-6), for any $0 < \lambda < 1$ and $q \leq r$, it holds that*
$$\|\Sigma_\lambda^{-s}\bar{f}_t\|_{\mathcal{H}} \lesssim (1 + \lambda^{-(1/2+(q-r))_+} + \lambda t\eta\lambda^{-(3/2+(q-r))_+})\|h^*\|_{L_2(P_X)}.$$

**Proof.** Recall that

$$\bar{f}_t = (I - \eta(\Sigma + \lambda I)^{-1}\Sigma)\bar{f}_{t-1} + \eta(\Sigma + \lambda I)^{-1}S^*f^* = \sum_{j=0}^{t-1}(I - \eta(\Sigma + \lambda I)^{-1}\Sigma)^j\eta(\Sigma + \lambda I)^{-1}S^*f^*.$$

Therefore, we obtain the following

$$\|\Sigma_\lambda^{-q}\bar{f}_t\|_{\mathcal{H}} = \eta\|\sum_{j=0}^{t-1}(I - \eta\Sigma_\lambda^{-1}\Sigma)^j\Sigma_\lambda^{-1-q}S^*L^rh^*\|_{\mathcal{H}}$$

$$=\eta\|\sum_{j=0}^{t-1}(I - \eta\Sigma_\lambda^{-1}\Sigma)^j\Sigma_\lambda^{-1}(\Sigma + \lambda I)\Sigma_\lambda^{-q-1}S^*L^rh^*\|_{\mathcal{H}}$$

$$\leq\eta\|\sum_{j=0}^{t-1}(I - \eta\Sigma_\lambda^{-1}\Sigma)^j\Sigma_\lambda^{-1}\Sigma\Sigma_\lambda^{-q-1}S^*L^rh^*\|_{\mathcal{H}} + \lambda\eta\|\sum_{j=0}^{t-1}(I - \eta\Sigma_\lambda^{-1}\Sigma)^j\Sigma_\lambda^{-1}\Sigma_\lambda^{-q-1}S^*L^rh^*\|_{\mathcal{H}}$$

$$\leq\eta\|\sum_{j=0}^{t-1}(I - \eta\Sigma_\lambda^{-1}\Sigma)^j\Sigma_\lambda^{-1}\Sigma\|\|\Sigma_\lambda^{-q-1}S^*L^rh^*\|_{\mathcal{H}} + \lambda\eta\|\sum_{j=0}^{t-1}(I - \eta\Sigma_\lambda^{-1}\Sigma)^j\Sigma_\lambda^{-1}\Sigma_\lambda^{-q-1}S^*L^rh^*\|_{\mathcal{H}}$$

$$\leq\|\Sigma_\lambda^{-q-1}S^*L^rh^*\|_{\mathcal{H}} + \lambda t\eta\|\Sigma_\lambda^{-1}\Sigma_\lambda^{-q-1}S^*L^rh^*\|_{\mathcal{H}}$$

$$\leq\|S^*L_\lambda^{-q-1+r}h^*\|_{\mathcal{H}} + \lambda t\eta\|S^*L_\lambda^{-q-2+r}h^*\|_{\mathcal{H}}$$

$$\leq\sqrt{\langle h^*, L_\lambda^{-q-1+r}SS^*L_\lambda^{-q-1+r}h^*\rangle_{L_2(P_X)}} + \lambda t\eta\sqrt{\langle h^*, L_\lambda^{-q-2+r}SS^*L_\lambda^{-q-2+r}h^*\rangle_{L_2(P_X)}}$$

$$=\sqrt{\langle h^*, L_\lambda^{-q-1+r}LL_\lambda^{-q-1+r}h^*\rangle_{L_2(P_X)}} + \lambda t\eta\sqrt{\langle h^*, L_\lambda^{-q-2+r}LL_\lambda^{-q-2+r}h^*\rangle_{L_2(P_X)}}$$

$$\leq(\lambda^{-1/2-(q-r)} + \lambda t\eta\lambda^{-3/2-(q-r)})\|h^*\|_{L_2(P_X)} \leq (1 + t\eta)\lambda^{-1/2-(q-r)}\|h^*\|_{L_2(P_X)}.$$
$\qquad\square$

**Lemma 14.** *Given (A4-6) and $\lambda \in (0, 1)$, we have $\mathcal{N}_\infty(\lambda) \leq \mathrm{tr}(\Sigma^{1/s})\lambda^{-1/s}$, and $\mathcal{F}_\infty(\lambda) \leq 1/\lambda$.*

**Proof.** For the first inequality, we have

$$\mathcal{N}_\infty(\lambda) = \mathrm{tr}(\Sigma\Sigma_\lambda^{-1}) = \mathrm{tr}\left(\Sigma^{1/s}\Sigma^{1-1/s}\Sigma_\lambda^{-(1-1/s)}\Sigma_\lambda^{-1/s}\right)$$

$$\leq\mathrm{tr}\left(\Sigma^{1/s}\Sigma^{1-1/s}\Sigma_\lambda^{-(1-1/s)}\right)\lambda^{-1/s} \leq \mathrm{tr}\left(\Sigma^{1/s}\right)\lambda^{-1/s}.$$

As for the second inequality, note that

$$\mathcal{F}_\infty(\lambda) = \sup_{\boldsymbol{x}}\langle K_{\boldsymbol{x}}, \Sigma_\lambda^{-1}K_{\boldsymbol{x}}\rangle_{\mathcal{H}} \leq \sup_{\boldsymbol{x}}\lambda^{-1}\langle K_{\boldsymbol{x}}, K_{\boldsymbol{x}}\rangle_{\mathcal{H}} \leq \lambda^{-1}\sup_{\boldsymbol{x}}k(\boldsymbol{x}, \boldsymbol{x}) \leq \lambda^{-1}.$$

$\qquad\square$

## D.10   PROOF OF COROLLARY 9

**Proof.** Note that in this setting $v_x$ takes value of $\frac{2}{1+\kappa}$ and $\frac{2\kappa}{1+\kappa}$ with probability 1/2 each. From (D.8) in the proof of Proposition 4 one can easily verify that for NGD,

$$B(\hat{\boldsymbol{\theta}}_{\boldsymbol{F}^{-1}}) \to \frac{2^r(1+\kappa^{1+r})}{(1+\kappa)^{1+r}}\left(1 - \frac{1}{\gamma}\right).$$

For GD, the bias formula (D.7) can be simplified as

$$B(\hat{\boldsymbol{\theta}}_{\boldsymbol{I}}) \to \frac{1}{\gamma} \cdot \left(\frac{\left(\frac{2}{1+\kappa}\right)^r}{(1+\kappa+2m_1)^2} + \frac{\kappa\left(\frac{2\kappa}{1+\kappa}\right)^r}{(1+\kappa+2\kappa m_1)^2}\right) \cdot \left(\frac{m_1}{(1+\kappa+2m_1)^2} + \frac{\kappa m_1}{(1+\kappa+2\kappa m_1)^2}\right)^{-1},$$

where $m_1$ is the Stieltjes transform defined after Equation (D.7). From standard numerical calculation one can show that when $\gamma > 1$, $\kappa \geq 1$,

$$m_1 = \frac{(\kappa+1)\sqrt{\gamma^2(\kappa+1)^2 + 4(1-\gamma)(\kappa-1)^2} + (2-\gamma)(\kappa+1)^2}{8(\gamma-1)\kappa}.$$

Setting $B(\hat{\boldsymbol{\theta}}_{\boldsymbol{I}}) = B(\hat{\boldsymbol{\theta}}_{\boldsymbol{F}^{-1}})$ and solve for $r$, we have

$$r^* = -\ln c_{\kappa,\gamma}/\ln\kappa, \quad c_{\kappa,\gamma} = \frac{c_4 - c_2}{c_1 - c_3}, \tag{D.27}$$

where we defined

$$c_1 = \left(1 - \frac{1}{\gamma}\right)\frac{1}{\kappa+1},$$

$$c_2 = \left(1 - \frac{1}{\gamma}\right)\frac{\kappa}{\kappa+1},$$

$$c_3 = \frac{1}{\gamma} \cdot \frac{(1+\kappa+2\kappa m_1)^2}{m_1(1+\kappa+2\kappa m_1)^2 + \kappa m_1(1+\kappa+2m_1)^2},$$

$$c_4 = \frac{1}{\gamma} \cdot \frac{\kappa(1+\kappa+2m_1)^2}{m_1(1+\kappa+2\kappa m_1)^2 + \kappa m_1(1+\kappa+2m_1)^2}.$$

Hence, Proposition 4 (from which we know $r^* \in (-1, 0)$) and the uniqueness of (D.27) implies that when $r \geq r^*$, $B(\hat{\boldsymbol{\theta}}_{\boldsymbol{I}}) \leq B(\hat{\boldsymbol{\theta}}_{\boldsymbol{F}^{-1}})$, and vice versa. Finally, observe that in the special case of $\gamma = 2$, $m_1 = \frac{\kappa+1}{2\sqrt{\kappa}}$. Therefore, one can check that constants in (D.27) simplify to

$$c_1 - c_3 = \frac{1-\sqrt{\kappa}}{2(\kappa+1)}, \quad c_2 - c_4 = \frac{\sqrt{\kappa}(\sqrt{\kappa}-1)}{2(\kappa+1)},$$

which implies that $r^* = -1/2$.

$\square$

## D.11   PROOF OF PROPOSITION 10

**Proof.** Part (a) is a simple combination of (Bai & Yin, 2008, Theorem 2) and assumption (A3), which implies $\|\boldsymbol{\Sigma}_{\boldsymbol{X}}\|_2$ and $\|\boldsymbol{\Sigma}_{\boldsymbol{X}}^{-1}\|_2$ are both finite. For part (b), the substitution error for the variance term (ignoring the scalar $\sigma^2$) can be bounded as

$$|V^* - \hat{V}| = \left|\text{tr}\left(\boldsymbol{F}^{-1}\boldsymbol{X}^\top(\boldsymbol{X}\boldsymbol{F}^{-1}\boldsymbol{X}^\top)^{-2}\boldsymbol{X}\boldsymbol{F}^{-1}\boldsymbol{\Sigma}_{\boldsymbol{X}}\right) - \text{tr}\left(\hat{\boldsymbol{F}}^{-1}\boldsymbol{X}^\top(\boldsymbol{X}\hat{\boldsymbol{F}}^{-1}\boldsymbol{X}^\top)^{-2}\boldsymbol{X}\hat{\boldsymbol{F}}^{-1}\boldsymbol{\Sigma}_{\boldsymbol{X}}\right)\right|$$

$$\overset{(i)}{\leq} O(1)\left\|\boldsymbol{F}^{-1} - \hat{\boldsymbol{F}}^{-1}\right\|_2\left(\text{tr}\left(\boldsymbol{X}^\top\boldsymbol{S}^{-2}\boldsymbol{X}\boldsymbol{F}\boldsymbol{\Sigma}_{\boldsymbol{X}}\right) + \sqrt{d}\left\|\boldsymbol{X}^\top\boldsymbol{S}^{-2}\boldsymbol{X}\right\|_2\left\|\boldsymbol{\Sigma}_{\boldsymbol{X}}\hat{\boldsymbol{F}}^{-1}\right\|_F\right)$$

$$+ \text{tr}\left(\boldsymbol{X}\hat{\boldsymbol{F}}^{-1}\boldsymbol{\Sigma}_{\boldsymbol{X}}\boldsymbol{F}^{-1}\boldsymbol{X}^\top\right)\left\|\boldsymbol{S}^{-2} - \hat{\boldsymbol{S}}^{-2}\right\|_2 \overset{(ii)}{=} O(\epsilon).$$

where we defined $\boldsymbol{S} = \boldsymbol{X}\boldsymbol{F}^{-1}\boldsymbol{X}^{\top}$ and $\hat{\boldsymbol{S}} = \boldsymbol{X}\hat{\boldsymbol{F}}^{-1}\boldsymbol{X}^{\top}$ in (i) and applied $\operatorname{tr}(\boldsymbol{A}\boldsymbol{B}) \leq \lambda_{\max}(\boldsymbol{A} + \boldsymbol{A}^{\top})\operatorname{tr}(\boldsymbol{B})$ for positive semi-definite $\boldsymbol{B}$, as well as $\operatorname{tr}(\boldsymbol{A}\boldsymbol{B}) \leq \sqrt{d}\|\boldsymbol{A}\|_2\|\boldsymbol{B}\|_F$, and (ii) is due to (A3), $\psi > 1$, (Wedin, 1973, Theorem 4.1) and the following estimate,

$$
n_u^2 \left\|\boldsymbol{S}^{-2} - \hat{\boldsymbol{S}}^{-2}\right\|_2 \leq \left\|n_u\boldsymbol{S}^{-1} - n_u\hat{\boldsymbol{S}}^{-1}\right\|_2 \left(n_u\|\boldsymbol{S}^{-1}\|_2 + n_u\left\|\hat{\boldsymbol{S}}^{-1}\right\|_2\right)
$$
$$
\overset{(i)}{=} O(1)\left\|n_u\boldsymbol{S}^{-1}\right\|_2\left\|n_u\hat{\boldsymbol{S}}^{-1}\right\|_2\left\|\boldsymbol{S}/n_u - \hat{\boldsymbol{S}}/n_u\right\|_2 \overset{(ii)}{=} O(\epsilon),
$$

where (i) again follows from (Wedin, 1973, Theorem 4.1), and (ii) is due to (A1)(A3) and $\psi > 1$ (which implies $\|n_u\boldsymbol{S}^{-1}\|_2$ and $\|n_u\hat{\boldsymbol{S}}^{-1}\|_2$ are bounded a.s.). Finally, from part (a) we know that $\psi = \Theta(\epsilon^{-2})$ suffices to achieve $\epsilon$-accurate approximation of $\boldsymbol{F}$ in spectral norm. The substitution error for the bias term can be derived from similar calculation, the details of which we omit. $\quad\square$

## E  EXPERIMENT SETUP

### E.1  PROCESSING THE DATASETS

To obtain extra unlabeled data to estimate the Fisher, we zero pad pixels on the boarders of each image before randomly cropping; a random horizontal flip is also applied for CIFAR10 images (Krizhevsky et al., 2009). We preprocess all images by dividing pixel values by 255 before centering them to be located within $[-0.5, 0.5]$ with the subtraction by $1/2$. For CIFAR10, we downsample the original images using a max pooling layer with kernel size 2 and stride 2.

### E.2  SETUP AND IMPLEMENTATION FOR OPTIMIZERS

In all settings, GD uses a learning rate of 0.01 that is exponentially decayed every 1k updates with the parameter value 0.999. For NGD, we use a fixed learning rate of 0.03. Since inverting a parameter-by-parameter-sized Fisher estimate per iteration would be costly, we adopt the Hessian free approach (Martens, 2010) which computes approximate matrix-inverse-vector products using the conjugate gradient (CG) method (Hestenes et al., 1952). For each approximate inversion, we run CG for 200 iterations starting from the solution returned by the previous CG run. The precise number of CG iterations and the initialization heuristic roughly follow Martens & Sutskever (2012). For the first run of CG, we initialize the vector from a standard Gaussian, and run CG for 5k iterations. To ensure invertibility, we apply a very small amount of damping (0.00001) in most scenarios. For geometric interpolation experiments between GD and NGD, we use the singular value decomposition to compute the minus $\alpha$ power of the Fisher, as CG is not applicable in this scenario.

### E.3  OTHER DETAILS

For experiments in the label noise and misspecification sections, we pretrain the teacher using the Adam optimizer (Kingma & Ba, 2014) with its default hyperparameters and a learning rate of 0.001.

For experiments in the misalignment section, we downsample all images twice using max pooling with kernel size 2 and stride 2. Moreover, only for experiments in this section, we implement natural gradient descent by exactly computing the Fisher on a large batch of unlabeled data and inverting the matrix by calling PyTorch's `torch.inverse` before right multiplying the gradient.

