# OpenReview forum: "When does preconditioning help or hurt generalization?"
_ICLR.cc/2021/Conference — ICLR 2021 Poster_

### Official Review · AnonReviewer4 · 2020-10-23
**Insights for Natural Gradient Descent In Some Specific Cases**

**Rating:** 7
**Confidence:** 3

**Review:**

Summary:

The authors theoretically study the prediction performance of pre-conditioned gradient descent/flow with linear models and squared loss aligning in the setting of least squares regression and non-parametric regression. For parametric least squares, the predication performance of the limiting solution for preconditioned gradient flow i.e. time goes to infinity, is studied in an asymptotic regime where both the number of samples and dimension go to infinity in proportion to one another. Meanwhile for non-parametric regression, source and capacity assumptions are leveraged to achieve finite sample guarantees. Experiments are also conducted on neural networks in a student and teacher setup.

Summary of main Contributions:

A1) In the case of parametric least squares, an asymptotic characterisation of the test risk is utilised to study the limiting solution of preconditioned gradient flow. Preconditioning with the inverse Fisher information matrix (covariates population covariance) is shown to achieve the optimal variance among preconditioned updates (Theorem 1).  Meanwhile for the asymptotic bias, the optimal pre-conditioner depends upon the covariance of the ground truth parameter. In a mis-aligned case, where the ground truth covariance is equal to the inverse of the population covariates covariance, the optimal pre-conditioner for the bias aligns with the inverse Fisher information matrix (Theorem 2).

A2) In the case of an Isotropic covariance for the ground truth parameter, it is found that the Bias and Variance can be traded-off by interpolating between the two aforementioned pre-conditioners (Proposition 4).

A3) For non-parametric regression, gradient descent pre-conditioned with the inverse regularised population covariates covariance is considered. Mini-max optimal statistical rates are achieved with a number of iterations that grows logarithmically in the data set size i.e. linear convergence (Theorem 7).

A4) Experiments for neural networks are conducted in support of A1). Specifically, gradient descent pre-conditioned with the Fisher information matrix achieves better generalisation performance when the noise is large or the model is misaligned (Section 5).

A5) For parametric least squares with a mis-aligned ground truth parameter, it is shown that early stopping with NGD achieves lower Bias than any other pre-conditioned gradient descent (Proposition 6).

Pros:
B1) I feel contribution A1) in conjunction with A5) is novel and offers a precise interpretation of when pre-conditioning with the inverse Fisher information matrix can yield an improvement in performance.

B2) Contribution A2) is also interesting and can point towards understanding and controlling the implicit bias of gradient descent through the pre-conditioner i.e. taking a linear combination of two pre-conditioners.

B3) Contribution A4) supports the findings in A1) in a setting beyond least squares.



Concerns:

C1) The authors do not compare their theoretical results for non-parametric regression (contribution A3) ) to prior work within the literature. Specifically, reference [1] where the generalisation performance of a pre-conditioned gradient method is considered. To remedy this, I feel the authors should discuss how their theoretical results and proof method differ from [1] as well as the novelty of their approach.

C2) The theoretical results and discussion focus on a particular type of pre-conditioner: the inverse population covariates covariance and transforms thereof. This limits the applicability of the insights as this quantity is often not known in practice. Similarly, the experiments are in a setting where Fisher information is estimated accurately using 100,000 samples while training uses 1024 samples. In contrast, prior work for non-parametric regression considers pre-conditioners involving estimates of the population covariance [1]. To remedy this, I feel the authors should include a discussion on how their insights i.e. A1), A2) are impacted when the population covariance is swapped for an estimate (using unlabelled data).

C3) The manuscript can be difficult to read. For instance, the authors start with a time varying pre-conditioner while all pre-conditioners considered are constant in time. Tools from random matrix theory and regularity assumptions for non-parametric regression are introduced with little discussion. Section 3.3 "Misspecficiation \approx Label Noise" considers misspecification that is independent and gets interpreted as additional noise. It is not clear what this brings to the manuscript in terms of insights and introduces another layer of complexity.

 C4) For parametric least squares regression the results focus on three cases for the ground truth covariance: well-aligned (where it equals the covariates population covariance), mis-aligned (where it equals the inverse covariance population covariance) and Isotropic. Whereas the theoretical results allow for a more general ground truth covariance to be considered. It would be natural to follow the source conditions from non-parametric regression and investigate natural gradient descent when the ground truth covariance is not fully well- or mis-aligned.


General Comments:

-Remark on page 4 states "we demonstrate generalisation properties only possed by the population Fisher", clarify which properties are /only/ held the population Fisher versus Sample Fisher.

-In Proposition 4 possibly change the description "interpolating preconditioners" as all the preconditioned methods are interpolating the data, and thus, can be confusing.

-Proposition 4 states for pre-conditioners (ii) and (iii) the bias is monotone for $\alpha$ in some range depending upon the covariates population covariance. What is the range of $\alpha$ and is the risk increasing or decreasing? What conclusions are we to draw from this part of the result?

-In Figure 6 and Figure 23 how is "geometric" and "additive" interpolation defined ?

-More discussion around Proposition 6 would be helpful. For instance, in the statement of the result what is choice of P ? The analysis is described as difficult, although no details are provided into how this result was obtained. Within the proof why is the ratio of Eigenvalues \overline{\lambda}_{min}/\widehat{\lambda}_{min} is bounded, and how many iterations are required until NGD is below, say, standard gradient descent?

-In Section 5, the misalignment experiment in Figure 7 is conducted for MNIST but not CIFAR-10, with no discussion in the main body of the manuscript for why this is. Although, paragraph "Misalignment" in Appendix C.3 states the phenomena of NGD outperforming GD in the misaligned case is "... difficult to observe in practical neural network training on real-world data". The authors then go on to state that, in short, this is due to (see Appendix A) NGD moving parameters further from initialisation, and thus, no longer well described by a linear model i.e. NTK.  Is there a link between this discussion within the Appendix and the experiments?

-A Summary at the start of Appendix A to describe contents of A1-A4 would improve readability.  Similarly, for Appendix C and D.

-In the proof of Theorem 2 (Appendix D.2) some details on how to get from (ii) to (iii).


[1] - Rudi, A., Carratino, L., and Rosasco, L. "Falkon: An optimal large scale kernel method",  Advances in Neural Information Processing System 2017.


POST REBUTTAL EDIT:

I thank the authors for providing detailed answers regarding my concerns. I have updated my score in light these comments. Below are some additional comments in response.

Response to comments regarding C1) and C2):
While early stopping with pre-conditioned updates differentiates this work from (A. Rudi et. al 2019), the analysis still requires the knowledge of the population covariance. Indeed, while the authors have included a section (Appendix A.3) showing that the operator norm of the population and the inverse regularised empirical covariance can be controlled, it would be insightful to discuss to what extent this allows the analysis for the pre-conditioned gradient descent to be extended to an approximated population covariance.

Response to comment regarding C3):
I am inclined to agree with reviewer 3, in that the manuscript is difficult to read due to the larger number of fragmented results. In this regard, I feel the authors should focus on a single phenomenon that is supported by both the parametric and non-parametric aspects of the paper, for instance, how pre-conditioning helps against misalignment.

Response to comment regarding different prior on ground truth (point 4. third bullet point):
Note that some concurrent works have studied the case of different priors on the ground truth [2,3], which are likely relevant in this case.

Minor Comment: The pre-conditioned updates for non-parametric regression (4.1) use notation $\alpha$ where as Appendix D.8.1 uses notation $\lambda$, with the discussion then switching back to using $\alpha$ and $\lambda$ being used in reference to the regularisation used within FALKON. The switching of notation is possibly confusing here.

[2] - D. Richards, J. Mourtada, L. Rosasco "Asymptotics of Ridge (less) Regression under General Source Condition", arXiv:2006.06386 (2020)

[3] - Wu, D. and Xu, J. "On the Optimal Weighted $\ell_2 $ Regularization in Overparameterized Linear Regression" NeurIPS 2020

---

> ### Author Response · Authors · 2020-11-17
> **Reply to Reviewer 4: Thank you for providing detailed and thoughtful comments.**
>
> Thank you for the comments and suggestions. We will correct typos, clarify the confusing points you mentioned, provide additional discussion on our assumptions, and add a summary of results in the Appendix in our revision. The technical comments are addressed below:
>
> 1. Comparison with prior works (FALKON) (C1).
> Thank you for mentioning this paper which is highly relevant to our Section 4.3; we will discuss and compare the results in Section 4.3 as well as Appendix D.8 of our revision. We agree that both FALKON and our Equation 4.1 are preconditioned updates that achieve the minimax optimal rate more efficiently. But we emphasize the following differences between [1] and our analysis in Section 4.3.
>     + The two algorithms optimize different objectives, which is highlighted by the different role of the “ridge” coefficient ($\alpha$ in our update, and $\lambda$ in FALKON). In FALKON, $\lambda$ turns the objective into kernel ridge regression; whereas in our update Equation 4.1, $\alpha$ controls the interpolation between GD and NGD. As we aim to study *how preconditioning affects generalization*, it is important that we look at the objective in its original (instead of regularized) form.
>     + To elaborate on the first point, since FALKON minimizes a regularized objective, it may not "overfit" even after infinite number of gradient steps, yet it is unclear how preconditioning impacts the generalization error (i.e., any preconditioner may generalize well under appropriate regularization). In contrast, we consider the "unregularized" objective, and thus early stopping plays a crucial role to avoid overfitting; this is different from many standard analysis on gradient descent.
>     + Algorithm-wise, the two updates employ different preconditinoers. The FALKON algorithm involves inverting the kernel matrix $K$ defined on the training points (or random projections), whereas we consider the population covariance operator $\Sigma$, which is consistent with our earlier discussion on the population Fisher in Section 3.
>     + In terms of the theoretical setup, our analysis allows for $r<1/2$, whereas [1] and many other previous works assumed $r\in[1/2,1]$ (as commented in Appendix D.8.1).
>
> 2. Estimating the population covariance (C2).
> Thank you for bringing this up. For simplicity and conciseness of results, our current theoretical analysis considers the idealized setup with access to the exact population covariance, which can be estimated from additional unlabeled data. Following your suggestion, we will include a discussion on swapping the exact quantity with the sample-based estimate in the new Appendix A.3.
> In summary, standard estimate implies that to achieve a stationary bias and variance that is $\epsilon$-close to that of the exact population Fisher, roughly $\Theta(\epsilon^{-2}d)$ samples is required in the estimation (thus to have $\epsilon\to 0$, additional logarithmic oversampling is required).
> We remark that this substitution error does not impose any structural assumptions on the estimated matrix, and it is known that under certain structures (e.g. Kronecker factorization), estimation of the Fisher can be more sample-efficient.
> Additionally, in certain cases if we only need to obtain the population spectrum (e.g., in the codiagonalizable setting), then accurate approximation is possible using less samples, for instance by inverting the Marchenko-Pastur equation [El Karoui 2008].
> Finally, we note that in Section 5.1 we plotted the population risk of trained neural networks under varying amount of unlabeled data in Fisher estimation. Observe that when the number of unlabeled data is small (preconditioner is more similar to the sample Fisher), the trend resembles that of ordinary GD, and vice versa; this is consistent with our findings in Section 3.
>
> 3. Comment on model misspecification (C3).
> As discussed in Section 3, the population risk can be decomposed into the variance, the well-specified bias and the misspecified bias. The well-specified bias can be interpreted as the difficulty of learning within the function class of the student, whereas the misspecified bias captures what's beyond the capacity of the student. We analyze the model misspecification setting for two reasons.
>     + In certain special case the misspecified bias behaves the same as the variance term, and thus we can easily determine the corresponding optimal preconditioner.
>     + While Proposition 3 only considers a rather limited setting, we empirically observe similar trends in neural network optimization (see figures in Section 5) by creating a mismatch between the student and teacher model.
> We believe that this scenario may be relevant in real-world problems.

---

> > ### Author Response · Authors · 2020-11-17
> > **Reply to Reviewer 4 (continued)**
> >
> > 4. Beyond three cases for ground truth covariance (C4).
> > We make the following remarks.
> >     + The (mis-)alignment conditions are constructed to discuss the advantage/disadvantage of NGD, whereas the characterization of optimal preconditioner (Theorem 2) holds regardless of such conditions.
> >     + Precise quantitative comparison between GD and NGD beyond the aforementioned cases would depend on the specific structure of the covariances and may vary case-by-case. To provide a qualitative intuition, if $\theta^*$ is dominated by isotropic components, then we expect GD to have lower stationary bias (even though it would not give the optimal solution) compared to NGD. In contrast, if $\Sigma_\theta$ is closer to fully misaligned ($\Sigma_X^{-1}$), then NGD should be beneficial.
> > We will comment on this in the main text in the revision, and try to come up with a more quantitative statement in the next few days.
> >     + Lastly, it is worth noting that most previous works on asymptotic analysis of the ridgeless interpolant (e.g., [Dobriban and Wager 2018], [Hastie et al. 2019]) only handles the isotropic case, and our extension of such analysis to more general target covariances is technically non-trivial.
> >
> > 5. Population vs. sample Fisher (Remark on page 4).
> > The paragraphs above the Remark on page 4 indicates that sample Fisher-based preconditioned updates converge to the same minimum Euclidean norm solution as ordinary GD.
> > Therefore, in Section 3.1-3.3, by comparing the stationary solution of GD and NGD (population Fisher), we also reveal the different properties of the sample vs. population Fisher.
> >
> > 6. Monotonicity in Proposition 4.
> > We make the following clarifications.
> >     + The monotonicity in Proposition 4 suggests that as we increase $\alpha$, the stationary variance increases and the stationary bias decreases. This suggests that one can trade in one of bias/variance for the other by varying $\alpha$, and thus at certain signal-to-noise ratio, update that interpolates between GD and NGD can be beneficial -- this is also empirically supported in Appendix A.
> >     + The monotonicity holds for all $\alpha$ between 0 and 1 for the variance term and certain range of $\alpha$ depending on the condition number of $\Sigma_X$ as specified by the points (a)(b) in the proof of Proposition 4. This designated range is likely an artifact of our current proof, as we empirically verify such monotonicity in the bias for a wide range of distributions and for all $\alpha\in[0,1]$ in Appendix D.5.
> >
> > 7. Definition of "geometric" and "additive" interpolation.
> > As indicated in the Remark in Section 4.1, the additive interpolation refers to choice (ii) in Proposition 4, and geometric interpolation refers to choice (iii). We will make this more explicit in the revision.
> >
> > 8. Clarification on Proposition 6.
> >     + The first part in Proposition 6 holds true for all choices of P satisfying (A3). Note that the ratio of eigenvalues is bounded due to $\gamma>1$ and (A3), which implies that eigenvalues are upper- and lower-bounded (away from 0).
> >     + The current proposition 6 only considers the continuous time case for conciseness of result, and thus we do not characterize the discretized dynamics (i.e. number of iterations).
> >     + What we intend to convey in the paragraph before Proposition 6 is that analyzing the early stopping bias beyond our considered special case can be difficult; one reason is that the bias can be non-monotone w.r.t. time, as empirically shown in Appendix A.2; this presents a challenge in determining the optimal early stopping point.
> >
> > 9. Neural network experiments.
> > To clarify, all the neural network experiments in the main text are conducted on either MNIST or CIFAR-10, with no particular preference in one over the other (the major difference is that CIFAR is more time-consuming).
> > We do however acknowledge that it seems difficult to construct a "misalignment" setup in neural network that robustly manifests the desired trend (as in linear regression), which we partially attribute to the discrepancy between linear model (the characterization of which is the primary focus of our work) and neural networks (especially when trained with NGD). So while we are able to observe the different behaviors of GD and NGD in rather artificial settings, we did not perform extensive experiments (changing dataset, varying the number of unlabeled data, etc.) for misalignment.
> >
> > 10. Comments on assumptions.
> > We will include such comments in our revision.
> > Currently, due to space constraint, we deferred most discussions on our setup to to the Appendix (for parametric regression see D.1 and D.2, and for comments on assumptions for non-parametric regression see D.8.1).
> >
> > 11. Details on proof of Theorem 2.
> > From (ii) to (iii) we absorbed $\Sigma_{\theta/P}$ into the inverse by left- and right-multiplying $\Sigma_{\theta/P}^{-1/2}$, and then merged the inverses.

---

> > > ### Author Response · Authors · 2020-11-20
> > > **Update on "4. Beyond three cases for ground truth covariance"**
> > >
> > > Motivated by your suggestion in (C4), we followed the analogy of source condition in Section 3.2 and considered the setting of $\Sigma_{\theta} = \Sigma_x^{-r}$, where larger $r$ corresponds to more "misaligned" problems. We are able to show that
> > > + GD achieves lower bias than NGD for $r\le 0$, whereas NGD outperforms GD for $r\ge 1$. This is a fairly intuitive result as it suggests that GD is preferred when the teacher model $\theta^*$ is more "aligned" with the features than the isotropic case, and NGD is preferred in settings more misaligned than $\Sigma_{\theta} = \Sigma_x^{-1}$.
> > > + The above characterization also implies a transition from the GD-dominated regime to the NGD-dominated regime for some $r^*$ between 0 and 1, i.e. NGD achieves lower bias than GD when $r>r^*$, and vice versa.
> > > As mentioned in the previous reply, this transition point $r^*$ depends on specific spectral properties of $\Sigma_x$ and varies case-by-case. To give a concrete example, we looked at a special case where $\Sigma_x$ has a simple block structure, for which we are able to give an explicit formula of $r^*$ that relates to $\gamma=d/n$ and the condition number of $\Sigma_x$.
> > >
> > > These results can be found in the new Appendix A.4 of the latest revision.
> > >
> > > We hope our reply and revision addressed most of your concerns for an improved score.
> > > Please let us know if you have any additional suggestions or followup questions.
> > >
> > > Thanks.

---

### Official Review · AnonReviewer3 · 2020-10-27
**The paper studies the effects of preconditioning on generalization properties in deep learning.**

**Rating:** 6
**Confidence:** 3

**Review:**

Summary:

The paper studies the effects of preconditioning on generalization properties in deep learning. By using a bias-variance decomposition of the expected risk, the paper determines optimal precondition matrix $P$ for bias and variance. Then the paper analyzes the generalization performance via the aspects: clean labels, well-specified model and aligned signal. Finally, it extends the analysis to the reproducing kernel Hilbert.

Pros:

The theoretical results provide guidelines of choosing precondition matrix for practical problems. In particular, by decomposing the risk into a sum of a bias and a variance, the paper addresses the following points:

1. The asymptotic result on the variance (Theorem 1) implies that NGD achieves the minimal variance at stationary points, that suggests using NGD in the case where the variance term dominates.

2. Theorem 2, on the other hand, provides the asymptotic result on the bias and the optimal precondition matrix for the bias to reach minimal value at stationary.

3. Based on the results on the variance and the bias, Proposition 4 suggests an interpolating scheme between NGD and GD that aim at achieving better stationary risk than NGD or GD. The efficiency of this scheme is demonstrated in a least squares regression with the regular RKHS, where the interpolating scheme achieves the optimal convergence rate in fewer step than GD.

Cons:

1. The paper contains a number of unclear / undefined terms such as well-specified and aligned signal, that make it difficult to read.

2. The paper uses a lot of vague and unverified claims / statements which are usually the explanations after each theorem / proposition. For example, after theorem 1, it says that "Theorem 1 implies that preconditioning with the inverse population Fisher results in the optimal stationary variance... In other words, when the labels are noisy so that the risk is dominated by the variance term... We emphasize that this advantage is only present when the population Fisher is used, but not its sample-based counterpart". For me, it would be more clear if these statements could be explained in detail.

3. The paper is not well-organized. For me, it is a collection of results that are unconnected. For example, after reading the analyses of bias and variance, I have no idea how they support the study of generalization or why section "3.3 misspecification" is placed along with bias and variance analyses, etc. I am not saying these results are irrelevant, however, there should be a better way of arranging / writing them so that they can support well the ideas of the paper.

---

> ### Author Response · Authors · 2020-11-17
> **Reply to Reviewer 3: Please reconsider your evaluation.**
>
> We appreciate the reviewer's feedback and have put additional effort to highlight the definitions and provide intuitive explanations of our result in the revision. However, we respectfully disagree with some of the reviewer's comments, as discussed below.
>
> 1. "The paper contains a number of unclear / undefined terms..."
> We would like to bring the reviewer's attention to the introduction section and Section 3.2 of the original submission. The intuition of model misspecification and alignment is discussed in the bullet points in the introduction on page 2. The definition of well-specified model ($f^*$ also linear on input features) can be found on the first paragraph of Section 3.2, and similarly, misspecification ($f^*$ contains a residual in additional to the linear model on input features) is discussed in the first paragraph of Section 3.3. The notion of misalignment is explained after Theorem 2, and in Figure 4(c) we consider an example of misalignment $\Sigma_X=\Sigma_\theta^{-1}$ in which NGD achieves the lowest well-specified bias under (A3).
> We will add a separate figure on page 5 in the revision to better illustrate this intuition.
>
> 2. "The paper uses a lot of vague and unverified claims / statements..."
> We respectively disagree. The discussion after Theorem 1 highlights that the inverse of population Fisher results in the lowest stationary variance; in contrast, the sample Fisher converges to the same minimum norm solution and thus does not possess this advantage. This is discussed in the first two paragraphs on page 4 of the original submission, and is also empirically supported by neural network experiments in Section 5; we will emphasize on this distinction in our revision of Section 3.1.
> Lower stationary variance of NGD indicates that when the risk is dominated by the variance term (see the bias-variance decomposition on page 4), then NGD would achieve lower risk. This claim is verified in Figure 4(a), in which we observed that NGD leads to lowest stationary variance across all $\gamma>1$.
>
> 3. "After reading the analyses of bias and variance, I have no idea how they support the study of generalization"
> We would like to bring the reviewer's attention to the equation on page 4 after the remark, which indicates that the population risk, or the generalization error, can be decomposed into the bias and variance -- this decomposition is a very standard result.
> We therefore study the bias and variance of the ridgeless interpolant to understand its generalization performance.
>
> 4. "Why section "3.3 misspecification" is placed along with bias and variance analyses."
> As explained above, the population risk can be decomposed into the variance term, which depends on the label noise, and the bias term which depends on the teacher model $f^*$ (see Equation on page 4). Model misspecification therefore contributes to the bias term in the risk (see paragraph before and after Proposition 3), and Proposition 3 suggests that in certain special cases the misspecified bias is analogous to the variance term in Theorem 1 (thus we introduced the variance result first). This observation is also supported neural networks experiments (see figures in Section 5). We thus think this subsection is an integral part of the bias-variance analysis.
> We will add a more explicit explanation on page 4 in our revision.
>
> Finally, we notice that the reviewer did not provide any comments on our technical contributions (exact risk for ridgeless interpolant, minimax optimal rate in RKHS regression, etc.).
> We hope that our responses so far have cleaned up any confusion for the reviewer to reevaluate and to discuss our contribution.
> But if this is due to the reviewer not being familiar with the topic, we are happy to provide more context in the follow-up discussion and future revision of the paper.
>
> Please let us know if there is anything else we could clarify.

---

> > ### Author Response · Authors · 2020-11-23
> > **Update on the Revised Manuscript**
> >
> > To address the points raised in your review, we’ve made the following modifications in the main text, primarily to clarify and provide more context for our technical results.
> > - In Section 1, we reworded the introduction on misalignment and misspecification to explain how they contribute to the population risk.
> > - On page 4 of Section 3, we included a discussion on the bias-variance decomposition.
> > - In Section 3.1 and the new Appendix A.3, we emphasized the discrepancy between the sample Fisher and the population Fisher.
> > - In Section 3.2, we provided additional comments on our generalized random effects hypothesis, and added a figure on page 5 to illustrate the intuition of misalignment. We also elaborated the connection with the source condition in more details in the new Appendix A.4.
> >
> > We hope that our reply and revision addressed most of your concerns.
> > Since the focus of our work is mainly theoretical, we would appreciate if your review can be updated to evaluate and discuss our *technical contributions*.
> > If you have any additional questions or comments, it would be great if you could let us know before the rebuttal period ends.
> >
> > Thanks.

---

> > > ### Comment · AnonReviewer3 · 2020-11-24
> > > **I will update my review.**
> > >
> > > Dear authors,
> > >
> > > Thank you for your comments. Your comment addressed most of my concerns. Also, after reading the other reviewers' comments, especially Reviewer 4's comment, my confusion about some parts of the paper has been cleared up. I will add some more details in my review and reconsider my score.
> > >
> > > Best.

---

> > > > ### Author Response · Authors · 2020-11-25
> > > > **Thank you for your update.**
> > > >
> > > > Thank you for engaging in the discussion and for spending time to update your review!
> > > > We are glad that we were able to address most of your concerns, which definitely helped us improve our paper.

---

### Official Review · AnonReviewer2 · 2020-10-27
**interesting study of the generalization properties of preconditioned gradient methods**

**Rating:** 8
**Confidence:** 4

**Review:**

The paper studies generalization properties of preconditioned gradient descent on linear/kernel regression problems. The main preconditioner that is studied in addition to vanilla GD is the (population) Fisher matrix (natural gradient descent or NGD), its empirical counterpart, and its interpolation with GD.
The authors first consider the "ridgeless" regression setup in high-dimension, where the estimator corresponds to the limiting gradient flow iterate, and show that NGD leads to a smaller (and optimal) variance term, and can improve the bias term compared to GD particularly in the presence of strong misspecification. Among others, the authors also consider early-stopping in a non-parametric RKHS setup, showing that an appropriate interpolation between NGD and GD achieves optimal rates with a much smaller number of steps compared to GD, a difference which becomes larger for "difficult" problems (which require more weight on the Fisher preconditioner). The findings are further illustrated with simple experiments on neural networks.

Overall, the paper provides a comprehensive study of the impact of preconditioning/second-order methods/natural gradient on generalization by giving a precise analysis in tractable regression settings, which illustrate conditions under which preconditioning is or is not useful for better generalization. This makes the paper a strong contribution, and I am in favor of acceptance.

comments/typos:
- section 3: 'population risk' -> should this be excess risk given the presence of noise? add a reference or some more details on the bias-variance decomposition?
- the last sentence in section 3.2 "in the analogy..." could be clarified
- end of p.5 "lower bias compare to" -> "compared to"
- Prop. 6: first part with theta_P holds for any P? please specify
- theorem 7: specify conditions on eta?
- some comments on computational difficulties of the full preconditioner would be welcome. Would a diagonal preconditioner, as often used in deep learning, provide any (partial) benefits as in the full-matrix case presented here?


### Update after rebuttal
Thank you for the clarifications.
A couple minor comments:
- regarding theorem 7, my comment was that it would be useful to include the conditions on eta in the theorem statement in the main text (though I do not feel strongly about it)
- regarding "misalignment" and the relationship between the random effects model and the source condition, I appreciate the improved explanation of this analogy, but I still find that the last paragraph in section 3.2 could do a better job at providing the right intuition (skimming through the Richards et al. reference pointed out by R4 gave me a better intuition).

---

> ### Author Response · Authors · 2020-11-17
> **Reply to Reviewer 2: Thank you for the thoughtful review.**
>
> Thank you for the comments and suggestions. We will correct the typos that make a few clarifications in our revision of the manuscript.
> The technical comments are addressed below:
>
> 1. "Population risk" in the presence of noise?
> Thank you for bringing this up. We followed the setting of [Hastie et al. 2019], in which the risk is computed on "clean" data (i.e., the risk definition on page 4 involves the true target (teacher model) $f^*$ but not the noised labels $y$). We will refer to prior works on this setting of bias-variance decomposition.
> We remark that if we consider the population risk in the presence of noise, the only difference would be that the variance term shifts by a scalar (the -1 in Equation 3.2 is dropped), which does not affect any of our results.
>
> 2. Analogy in Section 3.2.
> In the linear regression setting, the source condition can be interpreted as an upper bound on the magnitude $\mathbb{E}||\Sigma^{-r/2}\theta_*||$ (analogous to Assumption (A4)). To provide an intuition, in the codiagonalizable case, smaller $r$ entails that the eigenvalues of $\Sigma_x$ and $\Sigma_\theta$ are more "misaligned" (problem is more difficult), and vice versa.  In the RKHS literature, the relation between $r$ and the regularity of the target function $f^*$ is relatively known fact, as briefly discussed in Section 4.3. We will make a comment on this in the main text. In addition, we will include a new subsection in the Appendix (A.4) to further elaborate this analogy.
>
> 3. "Prop. 6: first part with $\theta_P$ holds for any P?"
> Yes we confirm that the first part of the result holds for any P satisfying assumption (A3) -- sorry for the confusion.
>
> 4. "Theorem 7: specify conditions on $\eta$?"
> Theorem 7 in the main text is an informal version of the theorem in Appendix D.8, in which the step size is specified as $\eta<||\Sigma||$.
>
> 5. Comment on diagonal preconditioner.
> The limitation of our current analysis on the stationary risk is that we can only characterize
> (i) time-invariant preconditioners satisfying (A2) for the variance term or (A3) for the bias term;
> (ii) preconditioners that do not alter the span of the gradient (which converges to the same minimum norm solution).
> The former includes any fixed and full-rank diagonal preconditioners (this is however not true for Adagrad or Adam except in special cases), whereas the latter includes full-matrix Adagrad (as noted in the first paragraph on page 4).
> Understanding the performance of general time-varying preconditioner under similar setup is an important future direction.

---

### Author Response · Authors · 2020-11-17
**Reply to All Reviewers: Summary of Revision**

We thank all reviewers for their feedback, which helped us improve the submission in many ways. We have uploaded the revised paper, and we highlight the major changes below.

1. Clarifications and explanations in the main text.
In Section 3 we included a short explanation on the bias-variance decomposition on page 4, and added a figure on page 5 to illustrate the intuition of misalignment. We also provided additional comments on our technical results in Section 3.1 and 3.2.
In Section 4 we clarified a few confusing points raised by the reviewers.

2. Additional result on estimating the population covariance.
We introduced a new subsection in the Appendix (A.3) to discuss how the number of unlabeled data affects the accurate estimation of the population Fisher.

3. Additional result on comparing the well-specified bias.
Following the analogy of source condition, in the new Appendix A.4 we analyzed the setting of $\Sigma_{\theta}=\Sigma_x^{-r}$, in which $r$ controls the extent of "misalignment". We are able to provide a more precise comparison of GD vs. NGD in certain special cases.

4. Comparison with prior work.
In Section 4.3 and Appendix D.8 we discussed a relevant work (FALKON) mentioned in R4's review.

5. Update in paper organization.
We included a table of content for the Appendix to improve readability.
We also moved the neural networks experiments on interpolating between GD and NGD to page 9 in the main text, as one additional page is allowed.

Please let us know if there are any additional suggestions or follow-up questions.

---

### Decision · Program_Chairs · 2021-01-07
**Final Decision**

**Decision:**

Accept (Poster)

**Comment:**

The paper provides a study of the impact of preconditioning/second-order methods on generalization by giving a precise analysis in tractable regression settings.
It illustrates conditions under which preconditioning might be useful for better generalization.
The readability issues raised by the reviewers have been taken into account, as well as some missing references, except

Wu, D. and Xu, J. "On the Optimal Weighted Regularization in Overparameterized Linear Regression" NeurIPS 2020, raised by reviewer (though it is a really recent reference).
Overall the contributions are significant enough to accept the paper for publication.